# Variance Reduced Distributed Nonconvex Optimization Using Matrix Stepsizes

## Abstract

Matrix-stepsized gradient descent algorithms have been shown to have superior performance in non-convex optimization problems compared to their scalar counterparts. The det-CGD algorithm, as introduced by Li et al. (2024b), leverages matrix stepsizes to perform compressed gradient descent for non-convex objectives and matrix-smooth problems in a federated manner. The authors establish the algorithm's convergence to a neighborhood of a weighted stationarity point under a convex condition for the symmetric and positive-definite matrix stepsize. In this paper, we propose two variance-reduced versions of the det-CGD algorithm, incorporating MARINA and DASHA methods. Notably, we establish theoretically and empirically, that det-MARINA and det-DASHA outperform MARINA, DASHA and the distributed det-CGD algorithms in terms of iteration and communication complexities.

## 1 Introduction

We focus on optimizing the finite sum non-convex objective

$$\min_{x \in \mathbb{R}^d} \left\{ f(x) := \frac{1}{n} \sum_{i=1}^{n} f_i(x) \right\}. \tag{1}$$

In this context, each function $f_i : \mathbb{R}^d \to \mathbb{R}$ is differentiable and bounded from below. This type of objective function finds extensive application in various practical machine learning algorithms, which increase not only in terms of the data size but also in the model size and overall complexity as well. As a result, most neural network architectures result in highly non-convex empirical losses, which need to be minimized. In addition, it becomes computationally infeasible to train these models on one device, often excessively large, and one needs to redistribute them amongst different devices/clients. This redistribution results in a high communication overhead, which often becomes the bottleneck in this framework.

In other words, we have the following setting. The data is partitioned into $n$ clients, where the $i$-th client has access to the component function $f_i$ and its derivatives. The clients are connected to each other through a central device, called the server. In this work, we are going to study iterative gradient descent-based algorithms that operate as follows. The clients compute the local gradients in parallel. Then they compress these gradients to reduce the communication cost and send them to the server in parallel. The server then aggregates these vectors and broadcasts the iterate update back to the clients. This meta-algorithm is called federated learning. We refer the readers to Konečný et al. (2016); McMahan et al. (2017); Kairouz et al. (2021) for a more thorough introduction to federated learning.

### 1.1 Contributions

In this paper, we introduce two novel federated learning algorithms named det-MARINA and det-DASHA. These algorithms extend a recent method called det-CGD (Li et al., 2024b), which aims to solve problem (1) using matrix stepsized gradient descent. Under the matrix smoothness assumption proposed by Safaryan et al. (2021), the authors demonstrate that the matrix stepsized

version of the Distributed Compressed Gradient Descent (Khirirat et al., 2018) algorithm enhances communication complexity compared to its scalar counterpart. However, in their analysis, Li et al. (2024b) show stationarity only within a certain neighborhood due to stochastic compressors. Our algorithm addresses this issue by incorporating previously known variance reduction schemes, namely, MARINA (Gorbunov et al., 2021) and DASHA (Tyurin & Richtárik, 2024). We establish theoretically and empirically, that both algorithms outperform their scalar alternatives, as well as the distributed det-CGD algorithms. In addition, we describe specific matrix stepsize choices, for which our algorithms beat MARINA, DASHA and distributed det-CGD both in theory and in practice.

## 2 BACKGROUND AND MOTIVATION

For a given $\varepsilon > 0$, finding an approximately global optimum, that is $x_\varepsilon$ such that $f(x_\varepsilon) - \min_x f(x) < \varepsilon$, is known to be NP-hard (Jain et al., 2017; Danilova et al., 2022). However, gradient descent based methods are still useful in this case. When these methods are applied to non-convex objectives, they treat the function $f$ as locally convex and aim to converge to a local minimum. Despite this simplification, such methods have gained popularity in practice due to their superior performance compared to other approaches for non-convex optimization, such as convex relaxation-based methods (Tibshirani, 1996; Cai et al., 2010).

### 2.1 STOCHASTIC GRADIENT DESCENT

Arguably, one of the most prominent meta-methods for tackling non-convex optimization problems is stochastic gradient descent (SGD). The formulation of SGD is presented as the following iterative algorithm: $x^{k+1} = x^k - \gamma g^k$. Here, $g^k \in \mathbb{R}^d$ serves as a stochastic estimator of the gradient $\nabla f(x^k)$. SGD essentially mimics the classical gradient descent algorithm, and recovers it when $g^k = \nabla f(x^k)$. In this scenario, the method approximates the objective function $f$ using a linear function and takes a step of size $\gamma$ in the direction that maximally reduces this approximation. When the stepsize is sufficiently small, and the function $f$ is suitably smooth, it can be demonstrated that the function value decreases, as discussed in (Bubeck et al., 2015; Gower et al., 2019).

However, computing the full gradient can often be computationally expensive. In such cases, stochastic approximations of the gradient come into play. Stochastic estimators of the gradient can be employed for various purposes, leading to the development of different methods. These include stochastic batch gradient descent (Nemirovski et al., 2009; Johnson & Zhang, 2013; Defazio et al., 2014), randomized coordinate descent (Nesterov, 2012; Wright, 2015), and compressed gradient descent (Alistarh et al., 2017; Khirirat et al., 2018; Mishchenko et al., 2019). The latter, compressed gradient descent, holds particular relevance to this paper, and we will delve into a more detailed discussion of it in subsequent sections.

### 2.2 SECOND ORDER METHODS

The stochastic gradient descent is considered as a first-order method as it uses only the first order derivative information. Although being immensely popular, the first order methods are not always optimal. Not surprisingly, using higher order derivatives in deciding update direction can yield to faster algorithms. A simple instance of such algorithms is the Newton Star algorithm (Islamov et al., 2021):

$$x^{k+1} = x^k - \left(\nabla^2 f(x^\star)\right)^{-1} \nabla f(x^k), \tag{NS}$$

where $x^\star$ is the minimum point of the objective function. The authors establish that under specific conditions, the algorithm's convergence to the unique solution $x^\star$ in the convex scenario occurs at a local quadratic rate. Nonetheless, its practicality is limited since we do not have prior knowledge of the Hessian matrix at the optimal point. Despite being proposed recently, the Newton-Star algorithm gives a deeper insight on the generic Newton method (Gragg & Tapia, 1974; Miel, 1980; Yamamoto, 1987):

$$x^{k+1} = x^k - \gamma \left(\nabla^2 f(x^k)\right)^{-1} \nabla f(x^k). \tag{NM}$$

Here, the unknown Hessian of the Newton-Star algorithm, is estimated progressively along the iterations. The latter causes elevated computational costs, as the inverting a large square matrix is

expensive. As an alternative, quasi-Newton methods replace the inverse of the Hessian at the iterate with a computationally cheaper estimate (Broyden, 1965; Dennis & Moré, 1977; Al-Baali & Khalfan, 2007; Al-Baali et al., 2014).

### 2.3 FIXED MATRIX STEPSIZES

The det-CGD algorithm falls into this framework of the second order methods as well. Proposed by Li et al. (2024b)[1], the algorithm suggests using a uniform "upper bound" on the inverse Hessian matrix. Assuming matrix smoothness of the objective (Safaryan et al., 2021), they replace the scalar stepsize with a positive definite matrix $\boldsymbol{D}$. The algorithm is given as follows:

$$x^{k+1} = x^k - \boldsymbol{D}\boldsymbol{S}^k\nabla f(x^k). \tag{det-CGD}$$

**Matrix $\boldsymbol{D}$.** Here, $\boldsymbol{D}$ plays the role of the stepsize. Essentially, it uniformly lower bounds the inverse Hessian. The standard SGD is a particular case of this method, as the scalar stepsize $\gamma$ can be seen as a matrix $\gamma\boldsymbol{I}_d$, where $\boldsymbol{I}_d$ is the $d$-dimensional identity matrix. An advantage of using a matrix stepsize is more evident if we take the perspective of the second order methods. Indeed, the scalar stepsize $\gamma\boldsymbol{I}_d$ uniformly estimates the largest eigenvalue of the Hessian matrix, while $\boldsymbol{D}$ can capture the Hessian more accurately. The authors show both theoretical and empirical improvement that comes with matrix stepsizes.

**Matrix $\boldsymbol{S}^k$.** We assume that $\boldsymbol{S}^k$ is a positive semi-definite, stochastic sketch matrix. Furthermore, it is unbiased: $\mathbb{E}[\boldsymbol{S}^k] = \boldsymbol{I}_d$. We notice that det-CGD can be seen as a matrix stepsize instance of SGD, with $g^k = \boldsymbol{S}^k\nabla f(x^k)$. The sketch matrix can be seen as a linear compressing operator, hence the name of the algorithm: Compressed Gradient Descent (CGD) (Alistarh et al., 2017; Khirirat et al., 2018). A commonly used example of such a compressor is the Rand-$\tau$ compressor. This compressor randomly selects $\tau$ entries from its input and scales them using a scalar multiplier to ensure an unbiased estimation. By adopting this approach, instead of using all $d$ coordinates of the gradient, only a subset of size $\tau$ is communicated. Formally, Rand-$\tau$ is defined as follows:

$$\boldsymbol{S} = \frac{d}{\tau}\sum_{j=1}^{\tau} e_{i_j}e_{i_j}^\top. \tag{2}$$

Here, $e_{i_j}$ denotes the $i_j$-th standard basis vector in $\mathbb{R}^d$. For a more comprehensive understanding of compression techniques, we refer to Safaryan et al. (2022b).

### 2.4 THE NEIGHBORHOOD OF THE DISTRIBUTED DET-CGD1

The distributed version of det-CGD follows the standard federated learning paradigm (McMahan et al., 2017). The pseudocode of the method, as well as the convergence result of Li et al. (2024b), can be found in Appendix F. Informally, their convergence result can be written as

$$\min_{k=1,...,K} \mathbb{E}\left[\left\|\nabla f(x^k)\right\|_{\boldsymbol{D}}^2\right] \leq \mathcal{O}\left(\frac{(1+\alpha)^K}{K}\right) + \mathcal{O}\left(\alpha\right),$$

where $\alpha > 0$ is a constant that can be controlled. The crucial insight from this result is that the error bound does not diminish as the number of iterations increases. In fact, by controlling $\alpha$ and considering a large $K$, it is impossible to make the second term smaller than $\varepsilon$. This implies that the algorithm converges to a certain neighborhood surrounding the (local) optimum. This phenomenon is a common occurrence in SGD and is primarily attributable to the variance associated with the stochastic gradient estimator. In the case of det-CGD the stochasticity comes from the sketch $\boldsymbol{S}^k$.

### 2.5 VARIANCE REDUCTION

To eliminate this neighborhood, various techniques for reducing variance are employed. One of the simplest techniques applicable to CGD is gradient shifting. By replacing $\boldsymbol{S}^k\nabla f(x^k)$ with

---

[1]In the original paper, the algorithm is referred to as det-CGD, as there is a variant of the same algorithm named det-CGD2. Since we are going to use only the first one and our framework is applicable to both, we will remove the number in the end for the sake of brevity.

$S^k(\nabla f(x^k) - \nabla f(x^\star)) + \nabla f(x^\star)$, the neighborhood effect is removed from the general CGD. This algorithm is an instance of a more commonly known method called SGD$_\star$ (Gower et al., 2020). However, since the exact optimum $x^\star$ is typically unknown, this technique encounters similar challenges as the Newton-Star algorithm mentioned earlier. Fortunately, akin to quasi-Newton methods, one can employ methods that iteratively learn the optimal shift (Shulgin & Richtárik, 2022). A line of research focuses on variance reduction for CGD based algorithms on this insight.

To eliminate the neighborhood in the distributed version of CGD, denoted as det-CGD1, we apply a technique called MARINA (Gorbunov et al., 2021). MARINA cleverly combines the general shifting (Shulgin & Richtárik, 2022) technique with loopless variance reduction techniques (Qian et al., 2021). This approach introduces an alternative gradient estimator specifically designed for the federated learning setting. Thanks to its structure, it allows to establish an upper bound on the stationarity error that diminishes significantly with a large number of iterations. In this paper, we construct the analog of the this algorithm called det-MARINA, using matrix stepsizes and sketch gradient compressors. For this new method, we prove a convergence guarantee similar to the results of Li et al. (2024b) without a neighborhood term.

Furthermore, we also propose det-DASHA, which is the extension of DASHA in the matrix stepsize setting. The latter was proposed by Tyurin & Richtárik (2024) and it combines MARINA with momentum variance reduction techniques (Cutkosky & Orabona, 2019). DASHA offers better practicality compared to MARINA, as it always sends compressed gradients and does not need to synchronize among all the nodes.

### 2.6 Organization of the paper

The rest of the paper is organized as follows. Section 3 discusses the general mathematical framework. Section 4 and Section 5 present the det-MARINA and det-DASHA algorithms, respectively. We show the superior theoretical performance of our algorithms compared to the relevant existing algorithms, that is MARINA, DASHA and det-CGD in Section 6. The experimental results validating our theoretical findings are presented in Section 7, with additional details and setups available in the Appendix. We conclude the paper by outlining several directions of future work in Section 8.

## 3 Mathematical framework

In this section we present the assumptions that we further require in the analysis.

**Assumption 1.** (Lower Bound) *There exists $f^\star \in \mathbb{R}$ such that, $f(x) \geq f^\star$ for all $x \in \mathbb{R}^d$.*

This is a standard assumption in optimization, as otherwise the problem of minimizing the objective would not be correct mathematically. We then need a matrix version of Lipschitz continuity for the gradient.

**Definition 1.** ($L$-Lipschitz Gradient) *Assume that $f : \mathbb{R}^d \to \mathbb{R}$ is a continuously differentiable function and matrix $\boldsymbol{L} \in \mathbb{S}_{++}^d$. We say the gradient of $f$ is $\boldsymbol{L}$-Lipschitz if for all $x, y \in \mathbb{R}^d$*

$$\|\nabla f(x) - \nabla f(y)\|_{\boldsymbol{L}^{-1}} \leq \|x - y\|_{\boldsymbol{L}}. \tag{3}$$

In the following, we will assume that (3) is satisfied for component functions $f_i$.

**Assumption 2.** *Each function $f_i$ is $\boldsymbol{L}_i$-gradient Lipschitz, while $f$ is $\boldsymbol{L}$-gradient Lipschitz.*

In fact, the second half of the assumption is a consequence of the first one. Below, we formalize this claim.

**Proposition 1.** *If $f_i$ is $\boldsymbol{L}_i$-gradient Lipschitz for every $i = 1, \ldots, n$, then function $f$ has $\boldsymbol{L}$-Lipschitz gradient with $\boldsymbol{L} \in \mathbb{S}_{++}^d$ satisfying*

$$\frac{1}{n} \sum_{i=1}^n \lambda_{\max}\left(\boldsymbol{L}^{-1}\right) \cdot \lambda_{\max}\left(\boldsymbol{L}_i\right) \cdot \lambda_{\max}\left(\boldsymbol{L}_i \boldsymbol{L}^{-1}\right) = 1.$$

**Remark 1.** *In the scalar case, where $\boldsymbol{L} = L\boldsymbol{I}_d$, $\boldsymbol{L}_i = L_i\boldsymbol{I}_d$, the relation becomes $L^2 = \frac{1}{n} \sum_{i=1}^n L_i^2$. This corresponds to the statement in Assumption 1.2 in (Gorbunov et al., 2021).*

Nevertheless, the matrix $L$ found according to Proposition 1 is only an estimate. In principle, there might exist a better $L_f \preceq L$ such that $f$ has $L_f$-Lipschitz gradient.

More generally, this condition can be interpreted as follows. The gradient of $f$ naturally belongs to the dual space of $\mathbb{R}^d$, as it is defined as a linear functional on $\mathbb{R}^d$. In the scalar case, $\ell_2$-norm is self-dual, thus (3) reduces to the standard Lipschitz continuity of the gradient. However, with the matrix smoothness assumption, we are using the $L$-norm for the iterates, which naturally induces the $L^{-1}$-matrix norm for the gradients in the dual space. This insight, which is originally presented by Nemirovski & Yudin (1983), plays a key role in our analysis.

See Appendix C for a more thorough discussion on the properties of Assumption 2, as well as its connection to matrix smoothness (Safaryan et al., 2021).

## 4 MARINA-BASED VARIANCE REDUCTION

In this section, we present our algorithm det-MARINA with its convergence result. We construct a sequence of vectors $g^k$ which are stochastic estimators of $\nabla f(x^k)$. At each iteration, the server samples a Bernoulli random variable (coin flip) $c_k$ and broadcasts it in parallel to the clients, along with the current gradient estimate $g^k$. Each client, then, does a det-CGD-type update with the stepsize $D$ and a gradient estimate $g^k$. The next gradient estimate $g^{k+1}$ is then computed. With a low probability, that is when $c_k = 1$, we take the $g^{k+1}$ to be the full gradient $\nabla f(x^{k+1})$. Otherwise, we update it using the compressed gradient differences at each client. See Algorithm 1 for the pseudocode of det-MARINA.

---

**Algorithm 1** det-MARINA

1: **Input:** starting point $x^0$, stepsize matrix $D$, probability $p \in (0, 1]$, number of iterations $K$
2: Initialize $g^0 = \nabla f(x^0)$
3: **for** $k = 0, 1, \ldots, K - 1$ **do**
4:     Sample $c_k \sim \text{Be}(p)$
5:     Broadcast $g^k$ to all workers
6:     **for** $i = 1, 2, \ldots$ in parallel **do**
7:        $x^{k+1} = x^k - D \cdot g^k$
8:        **if** $c_k = 1$ **then**
9:           $g_i^{k+1} = \nabla f_i(x^{k+1})$
10:       **else**
11:          $g_i^{k+1} = g^k + S_i^k \left( \nabla f_i(x^{k+1}) - \nabla f_i(x^k) \right)$
12:       **end if**
13:     **end for**
14:     $g^{k+1} = \frac{1}{n} \sum_{i=1}^{n} g_i^{k+1}$
15: **end for**
16: **Return:** $\tilde{x}^K$ chosen uniformly at random from $\{x^k\}_{k=0}^{K-1}$

---

### 4.1 CONVERGENCE GUARANTEES

In the following theorem, we formulate one of the main results of this paper, which guarantees the convergence of Algorithm 1 under the above-mentioned assumptions.

**Theorem 1.** *Assume that Assumptions 1 and 2 hold, and the following condition on stepsize matrix $D \in \mathbb{S}_{++}^d$ holds,*

$$D^{-1} \succeq \left( \frac{(1-p) \cdot R(D, \mathcal{S})}{np} + 1 \right) L, \tag{4}$$

*where $R(D, \mathcal{S}) := \frac{1}{n} \sum_{i=1}^{n} \lambda_{\max}(L_i) \lambda_{\max}\left( L^{-\frac{1}{2}} L_i L^{-\frac{1}{2}} \right) \times \lambda_{\max}\left( \mathbb{E}\left[ S_i^k D S_i^k \right] - D \right)$. Then, after $K$ iterations of det-MARINA, we have*

$$\mathbb{E}\left[ \left\| \nabla f(\tilde{x}^K) \right\|_{\frac{D}{\det(D)^{1/d}}}^2 \right] \leq \frac{2 \left( f(x^0) - f^\star \right)}{\det(D)^{1/d} \cdot K}. \tag{5}$$

*Here, $\tilde{x}^K$ is chosen uniformly randomly from the first $K$ iterates of the algorithm.*

The criterion $\|\cdot\|^2_{\boldsymbol{D}/\det(\boldsymbol{D})^{1/d}}$ is the same as that used in Li et al. (2024b), known as determinant normalization. The weight matrix of the matrix norm has determinant 1 after normalization, which makes it comparable to the standard Euclidean norm.

**Remark 2.** *We notice that the right-hand side of the algorithm vanishes with the number of iterations, thus solving the neighborhood issue of the distributed det-CGD. Therefore, det-MARINA is indeed the variance reduced version of det-CGD in the distributed setting and has better convergence guarantees.*

**Remark 3.** *Theorem 1 implies the following iteration complexity for the algorithm. In order to get an $\varepsilon^2$ stationarity error[2], the algorithm requires $K$ iterations, with*

$$K \geq \frac{2(f(x^0) - f^\star)}{\det(\boldsymbol{D})^{1/d} \cdot \varepsilon^2}.$$

**Remark 4.** *In the case where no compression is applied, that is we have $\boldsymbol{S}_i^k = \boldsymbol{I}_d$, condition (4) reduces to $\boldsymbol{D} \preceq \boldsymbol{L}^{-1}$. The latter is due to $\mathbb{E}\left[\boldsymbol{S}_i^k \boldsymbol{D} \boldsymbol{S}_i^k\right] = \boldsymbol{D}$, which results in $R(\boldsymbol{D}, \mathcal{S}) = 0$. This is expected, since in the deterministic case det-MARINA reduces to GD with matrix stepsize.*

The convergence condition and rate of matrix stepsize GD can be found in (Li et al., 2024b). Below we do a sanity check to verify that the convergence condition for scalar MARINA can be obtained.

**Remark 5.** *Let us consider the scalar case. That is $\boldsymbol{L}_i = L_i \boldsymbol{I}_d, \boldsymbol{L} = L \boldsymbol{I}_d, \boldsymbol{D} = \gamma \boldsymbol{I}_d$ and $\omega = \lambda_{\max}\left(\mathbb{E}\left[\left(\boldsymbol{S}_i^k\right)^\top \boldsymbol{S}_i^k\right]\right) - 1$. Then, the condition (4) reduces to*

$$\gamma \leq \left[L\left(1 + \sqrt{\frac{(1-p)\omega}{pn}}\right)\right]^{-1}.$$

The latter coincides with the stepsize condition of the convergence result of scalar MARINA.

## 4.2 OPTIMIZING THE MATRIX STEPSIZE

Now let us look at the right-hand side of (5). We notice that it decreases in terms of the determinant of the stepsize matrix. Therefore, one needs to solve the following optimization problem to find the optimal stepsize:

$$\begin{aligned}
\text{minimize} \quad & \log\det(\boldsymbol{D}^{-1}) \\
\text{subject to} \quad & \boldsymbol{D} \text{ satisfying (4).}
\end{aligned}$$

The solution of this constrained minimization problem on $\mathbb{S}^d_{++}$ is not explicit. In theory, one may show that the constraint (4) is convex and attempt to solve the problem numerically. However, as stressed by Li et al. (2024b), the similar stepsize condition for det-CGD is not easily computed using solvers like CVXPY (Diamond & Boyd, 2016). Instead, we may relax the problem to certain linear subspaces of $\mathbb{S}^d_{++}$. In particular, we fix a matrix $\boldsymbol{W} \in \mathbb{S}^d_{++}$, and define $\boldsymbol{D} := \gamma \boldsymbol{W}$. Then, the condition on the matrix $\boldsymbol{D}$ becomes a condition for the scalar $\gamma$, which is given in the following corollary.

**Corollary 1.** *Let $\boldsymbol{W} \in \mathbb{S}^d_{++}$, defining $\boldsymbol{D} := \gamma \cdot \boldsymbol{W}$, where $\gamma \in \mathbb{R}_+$. then the condition in (4) reduces to the following condition on $\gamma$*

$$\gamma \leq \frac{2\lambda_{\boldsymbol{W}}}{1 + \sqrt{1 + 4\alpha\beta \cdot \Lambda_{\boldsymbol{W}, \mathcal{S}} \lambda_{\boldsymbol{W}}}}, \tag{6}$$

*where $\Lambda_{\boldsymbol{W}, \mathcal{S}} = \lambda_{\max}\left(\mathbb{E}\left[\boldsymbol{S}_i^k \boldsymbol{W} \boldsymbol{S}_i^k\right] - \boldsymbol{W}\right)$, $\lambda_{\boldsymbol{W}} = \lambda_{\max}^{-1}\left(\boldsymbol{W}^{\frac{1}{2}} \boldsymbol{L} \boldsymbol{W}^{\frac{1}{2}}\right)$, $\alpha = \frac{1-p}{np}$ and $\beta = \frac{1}{n}\sum_{i=1}^n \lambda_{\max}\left(\boldsymbol{L}_i\right) \cdot \lambda_{\max}\left(\boldsymbol{L}^{-1}\boldsymbol{L}_i\right)$.*

---

[2]We say a (possibly random) vector $x \in \mathbb{R}^d$ is an $\varepsilon$-stationary point of a possibly non-convex function $f : \mathbb{R}^d \mapsto \mathbb{R}$, if $\mathbb{E}\left[\|\nabla f(x)\|^2\right] \leq \varepsilon^2$. The expectation is over the randomness of the algorithm

This means that for every fixed $\boldsymbol{W}$, we can find the optimal scaling coefficient $\gamma$. In section Section 6, we will use this corollary to prove that a suboptimal matrix step size, determined in this efficient way, is already better than the optimal scalar step size.

**Extension to det-CGD2.** A variant of det-CGD, called det-CGD2, was also proposed by Li et al. (2024b). This algorithm, has the same structure as det-CGD with the sketch and stepsize interchanged. It was shown, that this algorithm has explicit stepsize condition in the single node setting. In Appendix G, we propose the variance reduced extension of the distributed det-CGD2 following the MARINA scheme.

## 5 DASHA-BASED VARIANCE REDUCTION

In this section, we present our second algorithm based on DASHA. The latter utilizes a different type of variance reduction based on momentum (MVR). Compared to MARINA, dasha makes simpler optimization steps and does not require periodic synchronization with all the nodes. Notice that one may further simplify the notations here used in the algorithm. However, we keep it this way as it is consistent with (Tyurin & Richtárik, 2024).

---

**Algorithm 2** det-DASHA

---

1: **Input:** starting point $x^0 \in \mathbb{R}^d$, stepsize matrix $\boldsymbol{D} \in \mathbb{S}_{++}^d$, momentum $a \in (0, 1]$, number of iterations $K$
2: Initialize $g_i^0, h_i^0 \in \mathbb{R}^d$ on the nodes and $g^0 = \frac{1}{n} \sum_{i=1}^n g_i^0$ on the server
3: **for** $k = 0, 1, \ldots, K - 1$ **do**
4:     $x^{k+1} = x^k - \boldsymbol{D} \cdot g^k$
5:     Broadcast $x^{k+1}$ to all nodes
6:     **for** $i = 1, 2, \ldots n$ in parallel **do**
7:         $h_i^{k+1} = \nabla f_i(x^{k+1})$
8:         $m_i^{k+1} = \boldsymbol{S}_i^k \left( h_i^{k+1} - h_i^k - a \left( g_i^k - h_i^k \right) \right)$
9:         $g_i^{k+1} = g_i^k + m_i^{k+1}$
10:      Send $m_i^{k+1}$ to the server.
11:     **end for**
12:     $g^{k+1} = g^k + \frac{1}{n} \sum_{i=1}^n m_i^{k+1}$
13: **end for**
14: **Return:** $\tilde{x}^K$ chosen uniformly at random from $\{x^k\}_{k=0}^{K-1}$

---

### 5.1 THEORETICAL GUARANTEES

**Theorem 2.** *Suppose that Assumptions 1 and 2 hold. Let us initialize $g_i^0 = h_i^0 = \nabla f_i(x^0)$ for all $i \in [n]$ in Algorithm 2, and define*

$$\Lambda_{\boldsymbol{D},\mathcal{S}} = \lambda_{\max} \left( \mathbb{E} \left[ \boldsymbol{S}_i^k \boldsymbol{D} \boldsymbol{S}_i^k \right] - \boldsymbol{D} \right), \quad \omega_{\boldsymbol{D}} = \lambda_{\max} \left( \boldsymbol{D}^{-1} \right) \cdot \Lambda_{\boldsymbol{D},\mathcal{S}}.$$

*If $a = \frac{1}{2\omega_{\boldsymbol{D}} + 1}$, and the following condition on stepsize $\boldsymbol{D} \in \mathbb{S}_{++}^d$ is satisfied*

$$\boldsymbol{D}^{-1} \succeq \boldsymbol{L} - \frac{4\lambda_{\max}(\boldsymbol{D}) \, \omega_{\boldsymbol{D}} \, (4\omega_{\boldsymbol{D}} + 1)}{n^2} \sum_{i=1}^n \lambda_{\max}(\boldsymbol{L}_i) \, \boldsymbol{L}_i,$$

*then the following inequality holds for the iterates of Algorithm 2*

$$\mathbb{E} \left[ \left\| \nabla f(\tilde{x}^K) \right\|_{\boldsymbol{D}/(\det(\boldsymbol{D}))^{1/d}}^2 \right] \leq \frac{2(f(x^0) - f^\star)}{\det(\boldsymbol{D})^{1/d} \cdot K}.$$

*Here $\tilde{x}^K$ is chosen uniformly randomly from the first K iterates of the algorithm.*

**Remark 6.** *The term $\Lambda_{\boldsymbol{D},\mathcal{S}}$ can be viewed as the matrix version of $\gamma \cdot \omega$, where $\omega$ is associated with the sketch, and $\gamma$ is the scalar stepsize. On the other hand, the $\omega_{\boldsymbol{D}}$ is the extension of $\omega$ in matrix norm. Similar to Remark 5, plugging in scalar arguments in the algorithm, we recover the result from Tyurin & Richtárik (2024).*

Following the same scheme as in Section 4, we choose $\boldsymbol{D} = \gamma_{\boldsymbol{W}} \cdot \boldsymbol{W}$, where $\boldsymbol{W} \in \mathbb{S}_{++}^d$. Thus, for a fixed $\boldsymbol{W}$, we relax the problem of finding the optimal stepsize to the problem of finding the optimal scaling factor $\gamma_{\boldsymbol{W}} > 0$.

**Corollary 2.** *For a fixed $\boldsymbol{W} \in \mathbb{S}_{++}^d$, the optimal scaling factor $\gamma_{\boldsymbol{W}} \in \mathbb{R}_+$ is given by*

$$\gamma_{\boldsymbol{W}} = \frac{2\lambda_{\boldsymbol{W}}}{1 + \sqrt{1 + 16C_{\boldsymbol{W}}\lambda_{\min}(\boldsymbol{L}) \cdot \lambda_{\boldsymbol{W}}}},$$

*where $C_{\boldsymbol{W}} := \lambda_{\max}(\boldsymbol{W}) \cdot \omega_{\boldsymbol{W}}(4\omega_{\boldsymbol{W}} + 1)/n$, and $\lambda_{\boldsymbol{W}} := \lambda_{\max}^{-1}\left(\boldsymbol{L}^{\frac{1}{2}}\boldsymbol{W}\boldsymbol{L}^{\frac{1}{2}}\right)$.*

We observe that the structure of the optimal scaling factor for obtained above is similar to the one obtained in Corollary 1.

**The availability of $\boldsymbol{L}$:**  For both det-MARINA and det-DASHA, in order to determine the matrix stepsize, the knowledge of $\boldsymbol{L}$ is needed, if $\boldsymbol{L}$ is known, better complexities are guaranteed. When $\boldsymbol{L}$ is unknown, a closed-form solution can be obtained for generalized linear models. In more general cases, $\boldsymbol{L}_i$ can be treated as hyperparameters and estimated using first-order information via a gradient-based method (Wang et al., 2022). One can think of this as some type of preprocessing step, after which the matrices are learnt.

## 6 COMPLEXITIES OF THE ALGORITHMS

### 6.1 DET-MARINA

The following corollary formulates the iteration complexity for det-MARINA for $\boldsymbol{W} = \boldsymbol{L}^{-1}$.

**Corollary 3.** *If we take $\boldsymbol{W} = \boldsymbol{L}^{-1}$, then the condition (6) on $\gamma$ is given by*

$$\gamma \leq 2\left(1 + \sqrt{1 + 4\alpha\beta \cdot \Lambda_{\boldsymbol{L}^{-1},\mathcal{S}}}\right)^{-1}. \tag{7}$$

*In order to satisfy $\varepsilon$-stationarity, that is $\mathbb{E}\left[\left\|\nabla f(\tilde{x}^K)\right\|_{\frac{\boldsymbol{D}}{\det(\boldsymbol{D})^{1/d}}}^2\right] \leq \varepsilon^2$, we require*

$$K \geq \mathcal{O}\left(\frac{\Delta_0 \cdot \det(\boldsymbol{L})^{\frac{1}{d}}}{\varepsilon^2} \cdot \left(1 + \sqrt{1 + 4\alpha\beta \cdot \Lambda_{\boldsymbol{L}^{-1},\mathcal{S}}}\right)\right),$$

*where $\Delta_0 := f(x^0) - f(x^\star)$. Moreover, this iteration complexity is always better than the one of MARINA.*

The proof can be found in the Appendix. In fact, we can show that in cases where we fix $\boldsymbol{W} = \boldsymbol{I}_d$ and $\boldsymbol{W} = \operatorname{diag}^{-1}(\boldsymbol{L})$, the same conclusion also holds, relevant details can be found in Appendix D.3. This essentially means that det-MARINA always has a "larger" stepsize compared to MARINA, even if the stepsize is suboptimal for the sake of efficiency, which leads to a better iteration complexity. In addition, because we are using the same compressor for those two algorithms, the communication complexity of det-MARINA is also provably better than that of MARINA.

In order to compute the communication complexity, we borrow the concept of expected density from Gorbunov et al. (2021).

**Definition 2.** *For a given sketch matrix $\boldsymbol{S} \in \mathbb{S}_+^d$, the expected density is defined as*

$$\zeta_{\boldsymbol{S}} = \sup_{x \in \mathbb{R}^d} \mathbb{E}\left[\|\boldsymbol{S}x\|_0\right],$$

*where $\|x\|_0$ denotes the number of non-zero components of $x \in \mathbb{R}^d$.*

In particular, we have $\zeta_{\mathtt{Rand}-\tau} = \tau$. Below, we state the communication complexity of det-MARINA with $\boldsymbol{W} = \boldsymbol{L}^{-1}$ and the Rand-$\tau$ compressor.

**Corollary 4.** *Assume that we are using sketch $\boldsymbol{S} \sim \mathcal{S}$ with expected density $\zeta_{\mathcal{S}}$. Suppose also we are running det-MARINA with probability $p$ and we use the optimal stepsize matrix with respect to $\boldsymbol{W} = \boldsymbol{L}^{-1}$. Then the overall communication complexity of the algorithm is given by $\mathcal{O}\big((Kp + 1)d + (1-p)K\zeta_{\mathcal{S}}\big)$. Specifically, if we pick $p = \zeta_{\mathcal{S}}/d$, then the communication complexity is given by*

$$\mathcal{O}\left(d + \frac{\Delta_0 \det(\boldsymbol{L})^{\frac{1}{d}}}{\varepsilon^2}\left(\zeta_{\mathcal{S}} + \sqrt{\frac{\beta}{n}\Lambda_{\boldsymbol{L}^{-1},\mathcal{S}}\zeta_{\mathcal{S}}(d - \zeta_{\mathcal{S}})}\right)\right).$$

Notice that in case where no compression is applied, the communication complexity reduces to $\mathcal{O}\big(d\Delta_0 \cdot \det(\boldsymbol{L})^{\frac{1}{d}}/\varepsilon^2\big)$. The latter coincides with the rate of matrix stepsize GD (see (Li et al., 2024b)). Therefore, the dependence on $\varepsilon$ is not possible to improve further since GD is optimal among first order methods (Carmon et al., 2020).

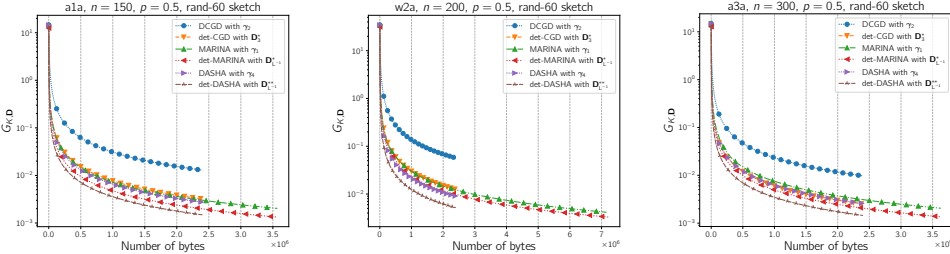

Figure 1: Comparison of DCGD with optimal scalar stepsize, det-CGD with matrix stepsize $\boldsymbol{D}_3^*$, MARINA with optimal scalar stepsize, DASHA with optimal scalar stepsize, det-MARINA with optimal stepsize $\boldsymbol{D}_{\boldsymbol{L}^{-1}}^*$ and det-DASHA with optimal stepsize $\boldsymbol{D}_{\boldsymbol{L}^{-1}}^{**}$. Throughout the experiment, we are using Rand-$\tau$ sketch with $\tau = 60$, and each algorithm is run for a fixed number of iterations $K = 10000$. The $G_{K,\boldsymbol{D}}$ in the y-axis is defined in (63), which is the average squared matrix norm of the gradients.

## 6.2 DET-DASHA

The difference of compression mechanisms, does not allow to have a direct comparison of the complexities of these algorithms. In particular, det-MARINA compresses the gradient difference with some probability $p$, while det-DASHA compresses the gradient difference with momentum in each iteration.

**Corollary 5.** *If we pick $\boldsymbol{D} = \gamma_{\boldsymbol{L}^{-1}} \cdot \boldsymbol{L}^{-1}$, then in order to reach an $\varepsilon^2$ stationary point, det-DASHA needs $K$ iterations with*

$$K \geq \frac{f(x^0) - f^\star}{\det(\boldsymbol{L})^{-\frac{1}{d}}\varepsilon^2}\left(1 + \sqrt{1 + 16C_{\boldsymbol{L}^{-1}}\lambda_{\min}(\boldsymbol{L})}\right).$$

The following corollary compares the complexities of DASHA and det-DASHA. For the sake of brevity, we defer the complexities and other details to the proof of this corollary.

**Corollary 6.** *Suppose that the conditions in Theorem 2 hold, then compared to DASHA, det-DASHA with $\boldsymbol{W} = \boldsymbol{L}^{-1}$ always has a **better** iteration complexity, therefore, communication complexity as well.*

The following corollary suggests that the communication complexity of det-DASHA is better than that of det-MARINA,

**Corollary 7.** *The iteration complexity of det-MARINA with $p = 1/(\omega_{\boldsymbol{L}^{-1}}+1)$ and det-DASHA with momentum $1/(2\omega_{\boldsymbol{L}^{-1}}+1)$ is the same, therefore the communication complexity of det-DASHA is **better than** the communication complexity of det-MARINA.*

This is expected since the same relation occurs between MARINA and DASHA as it is described by Tyurin & Richtárik (2024, Table 1). We refer the readers to Appendix E.2.1.

## 7 EXPERIMENTS

This section contains several plots which confirm our theoretical improvements on the existing methods. Figure 1 shows that the performance in terms of communication complexity of det-DASHA and det-MARINA is better than their scalar counterpart DASHA and MARINA respectively. This validates the efficiency of using a matrix stepsize over a scalar stepsize. Further, we notice that det-DASHA and det-MARINA have better communication complexity in this case, compared to det-CGD. This demonstrates the effectiveness of applying variance reduction. Finally, as expected, det-DASHA has better communication complexity than det-MARINA. We refer the readers to the appendix for more technical details of the experiments.

## 8 FUTURE WORK

i) In this paper, we have only considered (linear) sketches as the compression operator. However, there exists a variety of compressors which are useful in practice that do not fall into this category. Extending det-CGD and det-MARINA for general unbiased compressors is a promising future work direction. ii) Additionally, given recent successes with adaptive stepsizes (e.g., (Loizou et al., 2021; Orvieto et al., 2022; Schaipp et al., 2023)), designing an adaptive matrix stepsize tailored to our case could be viable. iii) Finally, recent advances suggest that server step sizes play a key role in accelerating federated learning algorithms (Jhunjhunwala et al., 2023; Li et al., 2024a). Designing a matrix version of the server step size could also be interesting.

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

CONTENTS

## A   ADDITIONAL DETAILS

### A.1   NOTATIONS

The standard Euclidean norm on $\mathbb{R}^d$ is defined as $\|\cdot\|$. We use $\mathbb{S}^d_{++}$ (resp. $\mathbb{S}^d_+$) to denote the positive definite (resp. semi-definite) cone of dimension $d$. $\mathbb{S}^d$ is used to denote all symmetric matrices of dimension $d$. We use the notation $\boldsymbol{I}_d$ to denote the identity matrix of size $d \times d$, and $\boldsymbol{O}_d$ to denote the zero matrix of size $d \times d$. Given $\boldsymbol{Q} \in \mathbb{S}^d_{++}$ and $x \in \mathbb{R}^d$,

$$\|x\|_{\boldsymbol{Q}} := \sqrt{x^\top \boldsymbol{Q} x} = \sqrt{\langle x, \boldsymbol{Q} x \rangle},$$

where $\langle \cdot, \cdot \rangle$ is the standard Euclidean inner product on $\mathbb{R}^d$. For a matrix $\boldsymbol{A} \in \mathbb{S}^d$, we use $\lambda_{\max}(\boldsymbol{A})$ (resp. $\lambda_{\min}(\boldsymbol{A})$) to denote the largest (resp. smallest) eigenvalue of the matrix $\boldsymbol{A}$. For a function $f : \mathbb{R}^d \mapsto \mathbb{R}$, its gradient and its Hessian at a point $x \in \mathbb{R}^d$ are respectively denoted as $\nabla f(x)$ and $\nabla^2 f(x)$. For the sketch matrices $\boldsymbol{S}_i^k$ used in the algorithm, we use the superscript $k$ to denote the iteration and subscript $i$ to denote the client, the matrix $\boldsymbol{S}_i^k$ is thus sampled for client $i$ in the $k$-th iteration from the same distribution $\mathcal{S}$. For any matrix $\boldsymbol{A} \in \mathbb{S}^d$, we use the notation $\operatorname{diag}(\boldsymbol{A}) \in \mathbb{S}^d$ to denote the diagonal of matrix $\boldsymbol{A}$.

## A.2 ADDITIONAL PRIOR WORK

Numerous effective convex optimization techniques have been adapted for application in non-convex scenarios. Here's a selection of these techniques, although it's not an exhaustive list: adaptivity (Dvinskikh et al., 2019; Zhang et al., 2020b), variance reduction (J Reddi et al., 2016; Li et al., 2021), and acceleration (Guminov et al., 2019). Of particular relevance to our work is the paper by Khaled & Richtárik (2023), which introduces a unified approach for analyzing stochastic gradient descent for non-convex objectives. A comprehensive overview of non-convex optimization can be found in (Jain et al., 2017; Danilova et al., 2022).

An illustrative example of a matrix stepsized method is Newton's method, which has been a long-standing favorite in the optimization community (Gragg & Tapia, 1974; Miel, 1980; Yamamoto, 1987). However, the computational complexity involved in computing the stepsize as the inverse of the Hessian of the current iteration is substantial. Instead, quasi-Newton methods employ a readily computable estimator to replace the inverse Hessian (Broyden, 1965; Dennis & Moré, 1977; Al-Baali & Khalfan, 2007; Al-Baali et al., 2014). An important direction of research that is relevant to our work, studies distributed second order methods. Here is a non-exhaustive list of papers in this area: (Wang et al., 2018; Crane & Roosta, 2019; Zhang et al., 2020a; Islamov et al., 2021; Alimisis et al., 2021; Safaryan et al., 2022a).

The Distributed Compressed Gradient Descent (DCGD) algorithm, initially proposed by Khirirat et al. (2018), has seen improvements in various aspects, as documented in works such as (Li et al., 2020; Horváth et al., 2022). Its variance reduced version with gradients shifts was studied by Shulgin & Richtárik (2022) in the (strongly) convex setting. Additionally, there exists a substantial body of literature on other federated learning algorithms employing unbiased compressors (Alistarh et al., 2017; Mishchenko et al., 2019; Gorbunov et al., 2021; Mishchenko et al., 2022; Maranjyan et al., 2022; Horváth et al., 2023).

Variance reduction techniques have gained significant attention in the context of stochastic batch gradient descent that is prevalent in machine learning. Numerous algorithms have been developed in this regard, including well-known ones like SVRG (Johnson & Zhang, 2013), SAG (Schmidt et al., 2017), SDCA(Richtárik & Takáč, 2014), SAGA (Defazio et al., 2014), MISO (Mairal, 2015), and Katyusha (Allen-Zhu, 2017). An overview of more advanced methods can be found in (Gower et al., 2020). Notably, SVRG and Katyusha have been extended with loopless variants, namely L-SVRG and L-Katyusha (Kovalev et al., 2020; Qian et al., 2021). These loopless versions streamline the algorithms by eliminating the outer loop and introducing a biased coin-flip mechanism at each step. This simplification eases both the algorithms' structure and their analyses, while preserving their worst-case complexity bounds. L-SVRG, in particular, offers the advantage of setting the exit probability from the outer loop independently of the condition number, thus, enhancing both robustness and practical efficiency.

This technique of coin flipping allows to obtain variance reduction for the CGD algorithm. A relevant example is the DIANA algorithm proposed by Mishchenko et al. (2019). Its convergence was proved both in the convex and non-convex cases. Later, MARINA (Gorbunov et al., 2021) obtained the optimal convergence rate, improving in communication complexity compared to all previous first order methods. Finally, there is a line of work developing variance reduction in the federated setting using other methods and techniques (Chraibi et al., 2019; Hanzely & Richtárik, 2020; Dinh et al., 2020; Peng et al., 2022).

Another method to obtain variance reduction is based on momentum. It was initially studied by Cutkosky & Orabona (2019), where they propose the STORM algorithm, which is a stochastic gradient descent algorithm with a momentum term for non-convex objectives. They obtain stationarity guarantees using adaptive stepsizes with optimal convergence rates. However, they require the vari-

ance of the stochastic gradient to be bounded by a constant, which is impractical. Using momentum for variance reduction has since been widely studied (Liu et al., 2020; Khanduri et al., 2020; Tran-Dinh et al., 2022; Li et al., 2022).

## B  BASIC FACTS

In this section, we present some basic facts along with their proofs that will be used later in the analysis.

**Fact 1.** *For two matrices $\boldsymbol{A}, \boldsymbol{B} \in \mathbb{S}_+^d$, denote the $i$-th largest eigenvalues of $\boldsymbol{A}, \boldsymbol{B}$ as $\lambda_i(\boldsymbol{A}), \lambda_i(\boldsymbol{B})$, if $\boldsymbol{A} \succeq \boldsymbol{B}$, then the following holds*

$$\lambda_i(\boldsymbol{A}) \geq \lambda_i(\boldsymbol{B}). \tag{8}$$

*Proof.* According to the Courant-Fischer theorem, we write

$$\lambda_i(\boldsymbol{B}) \quad = \quad \max_{S:\dim S=i} \min_{x \in S \setminus \{0\}} \frac{x^\top \boldsymbol{B} x}{x^\top x}.$$

Let $S_{\max}^i$ be a subspace of dimension $i$ where the maximum is attained, we then have

$$\lambda_i(\boldsymbol{B}) \quad = \quad \min_{x \in S_{\max}^i \setminus \{0\}} \frac{x^\top \boldsymbol{B} x}{x^\top x}$$

$$\leq \quad \min_{x \in S_{\max}^i \setminus \{0\}} \frac{x^\top \boldsymbol{A} x}{x^\top x} \leq \max_{S:\dim S=i} \min_{x \in S \setminus \{0\}} \frac{x^\top \boldsymbol{A} x}{x^\top x} = \lambda_i(\boldsymbol{A}).$$

$\square$

The following is a generalization of the bias-variance decomposition for the matrix norm.

**Fact 2.** (Variance Decomposition)  *Given a matrix $\boldsymbol{M} \in \mathbb{S}_{++}^d$, any vector $c \in \mathbb{R}^d$, and a random vector $x \in \mathbb{R}^d$ such that $\mathbb{E}\left[\|x\|\right] \leq +\infty$, the following bound holds*

$$\mathbb{E}\left[\|x - \mathbb{E}[x]\|_{\boldsymbol{M}}^2\right] = \mathbb{E}\left[\|x - c\|_{\boldsymbol{M}}^2\right] - \|\mathbb{E}[x] - c\|_{\boldsymbol{M}}^2. \tag{9}$$

*Proof.* We have

$$\mathbb{E}\left[\|x - c\|_{\boldsymbol{M}}^2\right] - \|\mathbb{E}[x] - c\|_{\boldsymbol{M}}^2$$

$$= \mathbb{E}\left[x^\top \boldsymbol{M} x\right] - 2\mathbb{E}[x]^\top \boldsymbol{M} c + c^\top \boldsymbol{M} c - \mathbb{E}[x]^\top \boldsymbol{M} \mathbb{E}[x] + 2\mathbb{E}[x]^\top \boldsymbol{M} c - c^\top \boldsymbol{M} c$$

$$= \mathbb{E}\left[x^\top \boldsymbol{M} x\right] - \mathbb{E}[x]^\top \boldsymbol{M} \mathbb{E}[x]$$

$$= \mathbb{E}\left[x^\top \boldsymbol{M} x\right] - 2 \cdot \mathbb{E}[x]^\top \boldsymbol{M} \mathbb{E}[x] + \mathbb{E}[x]^\top \boldsymbol{M} \mathbb{E}[x]$$

$$= \mathbb{E}\left[\|x - \mathbb{E}[x]\|_{\boldsymbol{M}}^2\right].$$

This completes the proof. $\square$

**Fact 3.** *The map $(\boldsymbol{A}, \boldsymbol{B}, \boldsymbol{X}) \mapsto \boldsymbol{A} - \boldsymbol{X}\boldsymbol{B}^{-1}\boldsymbol{X}$ is jointly concave on $\mathbb{S}_+^d \times \mathbb{S}_{++}^d \times \mathbb{S}^d$. It is also monotone increasing in variables $\boldsymbol{A}$ and $\boldsymbol{B}$.*

We refer the reader to Corollary 1.5.3 of Bhatia (2009) for the details and the proof. The following is a result of Fact 1 and Fact 3.

**Fact 4.** *Suppose $\boldsymbol{L}_i \in \mathbb{S}_{++}^d$, for $i = 1, \ldots, n$. Then, for every matrix $\boldsymbol{X} \in \mathbb{S}_{++}^d$, we define the following mapping*

$$f(\boldsymbol{X}, \boldsymbol{L}_1, \ldots, \boldsymbol{L}_n) = \frac{1}{n} \sum_{i=1}^{n} \lambda_{\max}(\boldsymbol{L}_i) \cdot \lambda_{\max}\left(\boldsymbol{L}_i \boldsymbol{X}^{-1}\right) \cdot \lambda_{\max}\left(\boldsymbol{X}^{-1}\right).$$

*Then the above mapping is monotone decreasing in $\boldsymbol{X}$.*

*Proof.* First we notice that from Fact 3 the mapping $\boldsymbol{X} \mapsto \boldsymbol{X}^{-1}$ is monotone decreasing. The latter means that if we have any $\boldsymbol{X}_1, \boldsymbol{X}_2 \in \mathbb{S}_{++}^d$ such that $\boldsymbol{X}_1 \succeq \boldsymbol{X}_2$, we have

$$\boldsymbol{X}_1^{-1} \preceq \boldsymbol{X}_2^{-1}.$$

Then it immediately follows, due to Fact 1, that

$$0 < \lambda_{\max}(\boldsymbol{X}_1^{-1}) \le \lambda_{\max}(\boldsymbol{X}_2^{-1}).$$

We also notice that the relation $\lambda_{\max}\left(\boldsymbol{L}_i \boldsymbol{X}^{-1}\right) = \lambda_{\max}\left(\boldsymbol{L}_i^{\frac{1}{2}} \boldsymbol{X}^{-1} \boldsymbol{L}_i^{\frac{1}{2}}\right) = \lambda_{\max}\left(\boldsymbol{X}^{-1} \boldsymbol{L}_i\right)$, and that the mapping $\boldsymbol{X} \mapsto \boldsymbol{L}_i^{\frac{1}{2}} \boldsymbol{X}^{-1} \boldsymbol{L}_i^{\frac{1}{2}}$ is also monotone decreasing for every $i \in [n]$, so we have

$$0 < \lambda_{\max}\left(\boldsymbol{L}_i \boldsymbol{X}_1^{-1}\right) \le \lambda_{\max}\left(\boldsymbol{L}_i \boldsymbol{X}_2^{-1}\right).$$

Since we have the coefficient $\lambda_{\max}\left(\boldsymbol{L}_i\right) > 0$, it follows that,

$$f(\boldsymbol{X}_1, \boldsymbol{L}_1, \ldots, \boldsymbol{L}_n) \le f(\boldsymbol{X}_2, \boldsymbol{L}_1, \ldots, \boldsymbol{L}_n).$$

This means that $f(\boldsymbol{X})$ is monotone decreasing in $\boldsymbol{X}$. $\qquad\square$

**Fact 5.** *For any two matrices $\boldsymbol{A}, \boldsymbol{B} \in \mathbb{S}_{++}^d$, the following relation regarding their largest eigenvalue holds*

$$\lambda_{\max}\left(\boldsymbol{A}\boldsymbol{B}\right) \le \lambda_{\max}\left(\boldsymbol{A}\right) \cdot \lambda_{\max}\left(\boldsymbol{B}\right). \tag{10}$$

*Proof.* Using the Courant-Fischer theorem, we can write

$$
\begin{aligned}
\lambda_{\max}\left(\boldsymbol{A}\boldsymbol{B}\right) &= \min_{S:\dim S=d} \max_{x \in S\setminus\{0\}} \frac{x^\top \boldsymbol{A}\boldsymbol{B} x}{x^\top x} \\
&= \max_{x \in \mathbb{R}^d\setminus\{0\}} \frac{x^\top \boldsymbol{A}\boldsymbol{B} x}{x^\top x} \\
&\le \max_{x \in \mathbb{R}^d\setminus\{0\}} \frac{x^\top \boldsymbol{A} x}{x^\top x} \cdot \max_{x \in \mathbb{R}^d\setminus\{0\}} \frac{x^\top \boldsymbol{B} x}{x^\top x} \\
&= \lambda_{\max}\left(\boldsymbol{A}\right) \cdot \lambda_{\max}\left(\boldsymbol{B}\right).
\end{aligned}
$$

$\square$

**Fact 6.** *Given matrix $\boldsymbol{Q} \in \mathbb{S}_{++}^d$ and its matrix norm $\|\cdot\|_{\boldsymbol{Q}}$, its associated dual norm is $\|\cdot\|_{\boldsymbol{Q}^{-1}}$.*

*Proof.* Let us first recall the definition of the dual norm $\|\cdot\|_*$. For any vector $z \in \mathbb{R}^d$, it is defined as

$$\|z\|_* := \sup\{z^\top x : \|x\|_{\boldsymbol{Q}} \le 1\}.$$

Solving this optimization problem is equivalent to solving $\sup\{z^\top x : \|x\|_{\boldsymbol{Q}}^2 = 1\}$. The Lagrange function is given as

$$f(x, \lambda) = z^\top x - \lambda\left(\|x\|_{\boldsymbol{Q}}^2 - 1\right) = z^\top x - \lambda\left(x^\top \boldsymbol{Q} x - 1\right).$$

Computing the derivatives we deduce that

$$\frac{\partial f(x, \lambda)}{\partial x} = z - 2\lambda \cdot \boldsymbol{Q} x = 0, \qquad \frac{\partial f(x, \lambda)}{\partial \lambda} = \|x\|_{\boldsymbol{Q}}^2 - 1 = 0.$$

This leads to

$$\lambda = \frac{\|z\|_{\boldsymbol{Q}^{-1}}}{2}, \qquad x = \frac{\boldsymbol{Q}^{-1} z}{\|z\|_{\boldsymbol{Q}^{-1}}}.$$

As a result, we have

$$
\begin{aligned}
\sup\{z^\top x : \|x\|_{\boldsymbol{Q}} \le 1\} &= \sup\{z^\top x : \|x\|_{\boldsymbol{Q}}^2 = 1\} \\
&= z^\top z = \frac{z^\top \boldsymbol{Q}^{-1} z}{\|z\|_{\boldsymbol{Q}^{-1}}} = \|z\|_{\boldsymbol{Q}^{-1}}.
\end{aligned}
$$

$\square$

# C PROPERTIES OF MATRIX SMOOTHNESS

## C.1 THE MATRIX LIPSCHITZ-CONTINUOUS GRADIENT

In this section we describe some properties of matrix smoothness, matrix gradient Lipschitzness and their relations. The following proposition describes a sufficient condition for the matrix Lipschitz-continuity of the gradient.

**Proposition 2.** *Given twice continuously differentiable function $f : \mathbb{R}^d \mapsto \mathbb{R}$ with bounded Hessian,*

$$\nabla^2 f(x) \preceq \boldsymbol{L}, \tag{11}$$

*where $\boldsymbol{L} \in \mathbb{S}_{++}^d$ and the generalized inequality holds for any $x \in \mathbb{R}^d$. Then $f$ satisfies (3) with the matrix $\boldsymbol{L}$.*

The below proposition is a variant of Proposition 1 and it characterizes the smoothness matrix of the objective function $f$, given the smoothness matrices of the component functions $f_i$.

**Proposition 3.** *Assume that $f_i$ has $\boldsymbol{L}_i$-Lipschitz continuous gradient for every $i \in [n]$, then function $f$ has $\boldsymbol{L}$-Lipschitz gradient with $\boldsymbol{L} \in \mathbb{S}_{++}^d$ satisfying*

$$\boldsymbol{L} \cdot \lambda_{\min}(\boldsymbol{L}) = \frac{1}{n} \sum_{i=1}^{n} \lambda_{\max}(\boldsymbol{L}_i) \cdot \boldsymbol{L}_i. \tag{12}$$

### C.1.1 QUADRATICS

Given a matrix $\boldsymbol{A} \in \mathbb{S}_{++}^d$ and a vector $b \in \mathbb{R}^d$, consider the function $f(x) = \frac{1}{2} x^\top \boldsymbol{A} x + b^\top x + c$. Then its gradient is computed as $\nabla f(x) = \boldsymbol{A} x + b$ and $\nabla^2 f(x) = \boldsymbol{A}$. Inserting gradients formula into (3) we deduce

$$\sqrt{(x - y)^\top \boldsymbol{A} \boldsymbol{L}^{-1} \boldsymbol{A} (x - y)} \leq \sqrt{(x - y)^\top \boldsymbol{L} (x - y)},$$

for any $x, y \in \mathbb{R}^d$. This reduces to

$$\boldsymbol{A} \boldsymbol{L}^{-1} \boldsymbol{A} \preceq \boldsymbol{L}. \tag{13}$$

Since $\boldsymbol{A} \in \mathbb{S}_{++}^d$, we can also rewrite (13) as

$$\boldsymbol{A}^{\frac{1}{2}} \boldsymbol{L}^{-1} \boldsymbol{A}^{\frac{1}{2}} \preceq \boldsymbol{A}^{-\frac{1}{2}} \boldsymbol{L} \boldsymbol{A}^{-\frac{1}{2}},$$

which is equivalent to

$$\boldsymbol{A} \preceq \boldsymbol{L}. \tag{14}$$

Therefore, the "best" $\boldsymbol{L} \in \mathbb{S}_{++}^d$ that satisfies (3) is $\boldsymbol{L} = \boldsymbol{A} = \nabla^2 f(x)$, for every $x \in \mathbb{R}^d$. Now, let us look at a more general setting. Consider $f$ given as follows,

$$f(x) = \sum_{i=1}^{s} \phi_i(\boldsymbol{M}_i x), \tag{}$$

where $\boldsymbol{M}_i \in \mathbb{R}^{q_i \times d}$. Here $f : \mathbb{R}^d \mapsto \mathbb{R}$ is the sum of functions $\phi_i : \mathbb{R}^{q_i} \mapsto \mathbb{R}$. We assume that each function $\phi_i$ has matrix $\boldsymbol{L}_i$ Lipschitz gradient. We have the following lemma regarding the matrix gradient Lipschitzness of $f$.

**Proposition 4.** *Assume that functions $f$ and $\{\phi_i\}_{i=1}^{s}$ are described above. Then function $f$ has $\boldsymbol{L}$-Lipschitz gradient, if the following condition is satisfied:*

$$\sum_{i=1}^{s} \lambda_{\max}\left(\boldsymbol{L}_i^{\frac{1}{2}} \boldsymbol{M}_i \boldsymbol{L}^{-1} \boldsymbol{M}_i^\top \boldsymbol{L}_i^{\frac{1}{2}}\right) = 1. \tag{15}$$

Note that Proposition 4 is a generalization of the previous case of quadratics, if we pick $s = 1$, $\boldsymbol{M}_i = \boldsymbol{A}^{\frac{1}{2}}$ and $\phi_1(x) = x^\top \boldsymbol{I}_d x$, the condition becomes $\boldsymbol{L} = \boldsymbol{A}$, which is exactly the solution given by (14). Thus we recover the result for quadratics. The linear term $bx + c$ is ignored in this case. In Proposition 4, we only intend to give a way of finding a matrix $\boldsymbol{L} \in \mathbb{S}_{++}^d$, so that $f$ has $\boldsymbol{L}$-Lipschitz gradient. This does not mean, however, the $\boldsymbol{L}$ here is optimal. The proof is deferred to Appendix C.3.

## C.2 COMPARISON OF THE DIFFERENT SMOOTHNESS CONDITIONS

Let us recall the definition of matrix smoothness.

**Definition 3.** ($L$-smoothness) *Assume that $f : \mathbb{R}^d \to \mathbb{R}$ is a continuously differentiable function and matrix $L \in \mathbb{S}_{++}^d$. We say that $f$ is $L$-smooth if for all $x, y \in \mathbb{R}^d$*

$$f(y) \leq f(x) + \langle \nabla f(x), x - y \rangle + \frac{1}{2} \|x - y\|_L^2. \tag{16}$$

We provide a proposition here which describes an equivalent form of stating $L$-matrix smoothness of a function $f$. This proposition is used to illustrate the relation between matrix smoothness and matrix Lipschitz gradient.

**Proposition 5.** *Let function $f : \mathbb{R}^d \to \mathbb{R}$ be continuously differentiable. Then the following statements are equivalent.*

   (i) *$f$ is $L$-matrix smooth.*

   (ii) *$\langle \nabla f(x) - \nabla f(y), x - y \rangle \leq \|x - y\|_L^2$ for all $x, y \in \mathbb{R}^d$.*

The two propositions, Proposition 6 and Proposition 7, formulated below illustrate the relation between matrix smoothness of $f$ and matrix gradient Lipschitzness of $f$.

**Proposition 6.** *Assume $f : \mathbb{R}^d \mapsto \mathbb{R}$ is a continuously differentiable function, and its gradient is $L$-Lipschitz continuous with $L \in \mathbb{S}_{++}^d$. Then function $f$ is $L$-matrix smooth.*

**Proposition 7.** *Assume $f : \mathbb{R}^d \to \mathbb{R}$ is a continuously differentiable function. Assume also that $f$ is convex and $L$-matrix smooth. Then $\nabla f$ is $L$-Lipschitz continuous.*

The next proposition shows that standard Lipschitzness of the gradient of a function is an immediate consequence of matrix Lipschitzness.

**Proposition 8.** *Assume that the gradient of $f$ is $L$-Lipschitz continuous. Then $\nabla f$ is also $L$-Lipschitz with $L = \lambda_{\max}(L)$.*

## C.3 PROOFS OF THE PROPOSITIONS REGARDING SMOOTHNESS

### C.3.1 PROOF OF PROPOSITION 1

We start with the definition of $L$-Lipschitz gradient of function $f$, and pick two arbitrary points $x, y \in \mathbb{R}^d$,

$$\|\nabla f(x) - \nabla f(y)\|_{L^{-1}}^2 \quad = \quad \left\| \frac{1}{n} \sum_{i=1}^{n} (\nabla f_i(x) - \nabla f_i(y)) \right\|_{L^{-1}}^2.$$

Applying the convexity of $\|\cdot\|_{L^{-1}}^2$, we have

$$\|\nabla f(x) - \nabla f(y)\|_{L^{-1}}^2 \quad \leq \quad \frac{1}{n} \sum_{i=1}^{n} \|\nabla f_i(x) - \nabla f_i(y)\|_{L^{-1}}^2.$$

For each term within the summation, we use the definition of matrix norms and replace the matrix $L^{-1}$ with $L_i^{-1/2} L_i^{1/2} L^{-1} L_i^{1/2} L_i^{-1/2}$, for every $i = 1, \ldots, n$:

$$\|\nabla f(x) - \nabla f(y)\|_{L^{-1}}^2 \quad = \quad \frac{1}{n} \sum_{i=1}^{n} \left( L_i^{-\frac{1}{2}} (\nabla f_i(x) - \nabla f_i(y)) \right)^{\top} L_i^{\frac{1}{2}} L^{-1} L_i^{\frac{1}{2}} \left( L_i^{-\frac{1}{2}} (\nabla f_i(x) - \nabla f_i(y)) \right)$$

$$\leq \quad \frac{1}{n} \sum_{i=1}^{n} \lambda_{\max} \left( L_i^{\frac{1}{2}} L^{-1} L_i^{\frac{1}{2}} \right) \left\| L_i^{-\frac{1}{2}} (\nabla f_i(x) - \nabla f_i(y)) \right\|^2$$

$$= \quad \frac{1}{n} \sum_{i=1}^{n} \lambda_{\max} \left( L_i^{\frac{1}{2}} L^{-1} L_i^{\frac{1}{2}} \right) \|\nabla f_i(x) - \nabla f_i(y)\|_{L_i^{-1}}^2.$$

Using the assumption that the gradient of each function $f_i$ is $\boldsymbol{L}_i$-Lipschitz, we obtain,

$$\|\nabla f(x) - \nabla f(y)\|_{\boldsymbol{L}^{-1}}^2 \leq \frac{1}{n} \sum_{i=1}^{n} \lambda_{\max}\left(\boldsymbol{L}_i^{\frac{1}{2}} \boldsymbol{L}^{-1} \boldsymbol{L}_i^{\frac{1}{2}}\right) \|x - y\|_{\boldsymbol{L}_i}^2.$$

Replacing $\boldsymbol{L}_i^{-1}$ with $\boldsymbol{L}^{-1/2} \boldsymbol{L}^{1/2} \boldsymbol{L}_i^{-1} \boldsymbol{L}^{1/2} \boldsymbol{L}^{-1/2}$

$$\|\nabla f(x) - \nabla f(y)\|_{\boldsymbol{L}^{-1}}^2 = \frac{1}{n} \sum_{i=1}^{n} \lambda_{\max}\left(\boldsymbol{L}_i^{\frac{1}{2}} \boldsymbol{L}^{-1} \boldsymbol{L}_i^{\frac{1}{2}}\right) \cdot \left[(\boldsymbol{L}^{\frac{1}{2}}(x-y))^\top \boldsymbol{L}^{-\frac{1}{2}} \boldsymbol{L}_i \boldsymbol{L}^{-\frac{1}{2}} (\boldsymbol{L}^{\frac{1}{2}}(x-y))\right]$$

$$\leq \frac{1}{n} \sum_{i=1}^{n} \lambda_{\max}\left(\boldsymbol{L}_i^{\frac{1}{2}} \boldsymbol{L}^{-1} \boldsymbol{L}_i^{\frac{1}{2}}\right) \cdot \lambda_{\max}\left(\boldsymbol{L}^{-\frac{1}{2}} \boldsymbol{L}_i \boldsymbol{L}^{-\frac{1}{2}}\right) \left\|\boldsymbol{L}^{\frac{1}{2}}(x-y)\right\|^2.$$

Using Fact 5, we are deduce the following bound,

$$\|\nabla f(x) - \nabla f(y)\|_{\boldsymbol{L}^{-1}}^2 \leq \left(\frac{1}{n} \sum_{i=1}^{n} \lambda_{\max}\left(\boldsymbol{L}^{-1}\right) \cdot \lambda_{\max}\left(\boldsymbol{L}_i\right) \cdot \lambda_{\max}\left(\boldsymbol{L}_i \boldsymbol{L}^{-1}\right)\right) \cdot \|x - y\|_{\boldsymbol{L}}^2$$

$$= \|x - y\|_{\boldsymbol{L}}^2.$$

### C.3.2 PROOF OF PROPOSITION 2

We start with picking any two vector $x, y \in \mathbb{R}^d$. We have

$$\|\nabla f(x) - \nabla f(y)\|_{\boldsymbol{L}^{-1}}^2$$

$$= \left\|\int_0^1 \nabla^2 f(\theta x + (1-\theta)y)(x-y)\, d\theta\right\|_{\boldsymbol{L}^{-1}}^2$$

$$= (x-y)^\top \left(\int_0^1 \nabla^2 f(\theta x + (1-\theta)y)\, d\theta\right)^\top \boldsymbol{L}^{-1} \left(\int_0^1 \nabla^2 f(\theta x + (1-\theta)y)\, d\theta\right)(x-y).$$

Denote $\boldsymbol{F} := \int_0^1 \nabla^2 f(\theta x + (1-\theta)y)\, d\theta$, notice that $\boldsymbol{F}$ is a symmetric matrix. Then, the previous identity becomes

$$\|\nabla f(x) - \nabla f(y)\|_{\boldsymbol{L}^{-1}}^2 = (x-y)^\top \boldsymbol{F}^\top \boldsymbol{L}^{-1} \boldsymbol{F} (x-y).$$

From the definition of $\boldsymbol{F}$ and the bounded Hessian assumption, we have $\boldsymbol{F} \preceq \boldsymbol{L}$. Let us prove that $\boldsymbol{F} \boldsymbol{L}^{-1} \boldsymbol{F} \preceq \boldsymbol{L}$:

$$\boldsymbol{F} \boldsymbol{L}^{-1} \boldsymbol{F} \preceq \boldsymbol{L} \iff \boldsymbol{L}^{-\frac{1}{2}} \boldsymbol{F} \boldsymbol{L} \boldsymbol{F} \boldsymbol{L}^{-\frac{1}{2}} \preceq \boldsymbol{I}_d$$

$$\iff \boldsymbol{L}^{-\frac{1}{2}} \boldsymbol{F} \boldsymbol{L}^{-\frac{1}{2}} \cdot \boldsymbol{L}^{-\frac{1}{2}} \boldsymbol{F} \boldsymbol{L}^{-\frac{1}{2}} \preceq \boldsymbol{I}_d$$

$$\iff \boldsymbol{L}^{-\frac{1}{2}} \boldsymbol{F} \boldsymbol{L}^{-\frac{1}{2}} \preceq \boldsymbol{I}_d$$

$$\iff \boldsymbol{F} \preceq \boldsymbol{L}.$$

This means that

$$\|\nabla f(x) - \nabla f(y)\|_{\boldsymbol{L}^{-1}}^2 \leq (x-y)^\top \boldsymbol{L}(x-y) = \|x - y\|_{\boldsymbol{L}}^2,$$

which completes the proof.

### C.3.3 PROOF OF PROPOSITION 3

Suppose $\boldsymbol{L}$ is a symmetric positive definite matrix satisfying (12). Let us now show that the function $\nabla f$ is $\boldsymbol{L}$-Lipschitz continuous. We start with picking any two points $x, y \in \mathbb{R}^d$, and notice that

$$\|\nabla f(x) - \nabla f(y)\|_{\boldsymbol{L}^{-1}}^2 = \left\|\frac{1}{n} \sum_{i=1}^{n} (\nabla f_i(x) - \nabla f_i(y))\right\|_{\boldsymbol{L}^{-1}}^2.$$

Applying Jensen's inequality, we obtain

$$\|\nabla f(x) - \nabla f(y)\|_{\boldsymbol{L}^{-1}}^2 \leq \frac{1}{n} \sum_{i=1}^{n} \|\nabla f_i(x) - \nabla f_i(y)\|_{\boldsymbol{L}^{-1}}^2.$$

We then re-weight the norm appears in the summation individually,

$$\|\nabla f(x) - \nabla f(y)\|_{\boldsymbol{L}^{-1}}^2 \leq \frac{1}{n} \sum_{i=1}^{n} \left(\nabla f_i(x) - \nabla f_i(y)\right)^\top \boldsymbol{L}_i^{-\frac{1}{2}} \boldsymbol{L}_i^{\frac{1}{2}} \boldsymbol{L}^{-1} \boldsymbol{L}_i^{\frac{1}{2}} \boldsymbol{L}_i^{-\frac{1}{2}} \left(\nabla f_i(x) - \nabla f_i(y)\right)$$

$$\leq \frac{1}{n} \sum_{i=1}^{n} \lambda_{\max}\left(\boldsymbol{L}_i\right) \cdot \lambda_{\max}\left(\boldsymbol{L}^{-1}\right) \cdot \|\nabla f_i(x) - \nabla f_i(y)\|_{\boldsymbol{L}_i^{-1}}^2 .$$

Utilizing the assumption that each $f_i$ has $\boldsymbol{L}_i$ Lipschitz gradient, we obtain

$$\|\nabla f(x) - \nabla f(y)\|_{\boldsymbol{L}^{-1}}^2 \leq \frac{1}{n} \sum_{i=1}^{n} \lambda_{\max}\left(\boldsymbol{L}_i\right) \cdot \lambda_{\max}\left(\boldsymbol{L}^{-1}\right) \cdot \|x - y\|_{\boldsymbol{L}_i}^2$$

$$= \|x - y\|_{\lambda_{\max}(\boldsymbol{L}^{-1}) \cdot \frac{1}{n} \sum_{i=1}^{n} \lambda_{\max}(\boldsymbol{L}_i) \cdot \boldsymbol{L}_i}^2 \overset{(12)}{=} \|x - y\|_{\boldsymbol{L}}^2 .$$

### C.3.4 PROOF OF PROPOSITION 4

For any $x$ and $y$ from $\mathbb{R}^d$, we have

$$\|\nabla f(x) - \nabla f(y)\|_{\boldsymbol{L}^{-1}}$$

$$= \left\| \sum_{i=1}^{s} \boldsymbol{M}_i^\top \nabla \phi_i(\boldsymbol{M}_i x) - \sum_{i=1}^{s} \boldsymbol{M}_i^\top \nabla \phi_i(\boldsymbol{M}_i y) \right\|_{\boldsymbol{L}^{-1}}$$

$$= s \cdot \left\| \frac{1}{s} \sum_{i=1}^{s} \boldsymbol{M}_i^\top \left(\nabla \phi_i(\boldsymbol{M}_i x) - \nabla \phi_i\left(\boldsymbol{M}_i y\right)\right) \right\|_{\boldsymbol{L}^{-1}} .$$

Applying the convexity of the norm $\|\cdot\|_{\boldsymbol{L}^{-1}}$,

$$\|\nabla f(x) - \nabla f(y)\|_{\boldsymbol{L}^{-1}} \leq s \cdot \frac{1}{s} \sum_{i=1}^{s} \left\| \boldsymbol{M}_i^\top \left(\nabla \phi_i(\boldsymbol{M}_i x) - \nabla \phi_i(\boldsymbol{M}_i y)\right) \right\|_{\boldsymbol{L}^{-1}} .$$

Expanding the norm and applying the replacement trick for above $\boldsymbol{L}$ and $\boldsymbol{M}_i$, we obtain

$$\|\nabla f(x) - \nabla f(y)\|_{\boldsymbol{L}^{-1}}$$

$$= \sum_{i=1}^{s} \sqrt{\left(\nabla \phi_i(\boldsymbol{M}_i x) - \nabla \phi_i(\boldsymbol{M}_i y)\right)^\top \boldsymbol{M}_i \boldsymbol{L}^{-1} \boldsymbol{M}_i^\top \left(\nabla \phi_i(\boldsymbol{M}_i x) - \nabla \phi_i(\boldsymbol{M}_i y)\right)}$$

$$= \sum_{i=1}^{s} \sqrt{\boldsymbol{B}_i^\top \boldsymbol{L}_i^{\frac{1}{2}} \boldsymbol{M}_i \boldsymbol{L}^{-1} \boldsymbol{M}_i^\top \boldsymbol{L}_i^{\frac{1}{2}} \boldsymbol{B}_i}$$

$$\leq \sum_{i=1}^{s} \sqrt{\lambda_{\max}\left(\boldsymbol{L}_i^{\frac{1}{2}} \boldsymbol{M}_i \boldsymbol{L}^{-1} \boldsymbol{M}_i^\top \boldsymbol{L}_i^{\frac{1}{2}}\right)} \cdot \|\nabla \phi_i(\boldsymbol{M}_i x) - \nabla \phi_i(\boldsymbol{M}_i y)\|_{\boldsymbol{L}_i^{-1}} ,$$

where $\boldsymbol{B}_i := \boldsymbol{L}_i^{-\frac{1}{2}} \left(\nabla \phi_i(\boldsymbol{M}_i x) - \nabla \phi_i(\boldsymbol{M}_i y)\right)$. Due to the assumption that the gradient of $\phi_i$ is $\boldsymbol{L}_i$-Lipschitz, we have

$$\|\nabla f(x) - \nabla f(y)\|_{\boldsymbol{L}^{-1}}$$

$$\leq \sum_{i=1}^{s} \sqrt{\lambda_{\max}\left(\boldsymbol{L}_i^{\frac{1}{2}} \boldsymbol{M}_i \boldsymbol{L}^{-1} \boldsymbol{M}_i^\top \boldsymbol{L}_i^{\frac{1}{2}}\right)} \cdot \|\boldsymbol{M}_i(x - y)\|_{\boldsymbol{L}_i}$$

$$= \sum_{i=1}^{s} \sqrt{\lambda_{\max}\left(\boldsymbol{L}_i^{\frac{1}{2}} \boldsymbol{M}_i \boldsymbol{L}^{-1} \boldsymbol{M}_i^\top \boldsymbol{L}_i^{\frac{1}{2}}\right)} \cdot \sqrt{\left[\boldsymbol{L}^{\frac{1}{2}}\left(x - y\right)\right]^\top \boldsymbol{L}^{-\frac{1}{2}} \boldsymbol{M}_i^\top \boldsymbol{L}_i \boldsymbol{M}_i \boldsymbol{L}^{-\frac{1}{2}} \left[\boldsymbol{L}^{\frac{1}{2}}\left(x - y\right)\right]}$$

$$\leq \sum_{i=1}^{s} \sqrt{\lambda_{\max}\left(\boldsymbol{L}_i^{\frac{1}{2}} \boldsymbol{M}_i \boldsymbol{L}^{-1} \boldsymbol{M}_i^\top \boldsymbol{L}_i^{\frac{1}{2}}\right) \cdot \lambda_{\max}\left(\boldsymbol{L}^{-\frac{1}{2}} \boldsymbol{M}_i^\top \boldsymbol{L}_i \boldsymbol{M}_i \boldsymbol{L}^{-\frac{1}{2}}\right)} \cdot \|x - y\|_{\boldsymbol{L}}$$

$$\leq \sum_{i=1}^{s} \lambda_{\max}\left(\boldsymbol{L}_i^{\frac{1}{2}} \boldsymbol{M}_i \boldsymbol{L}^{-1} \boldsymbol{M}_i^\top \boldsymbol{L}_i^{\frac{1}{2}}\right) \cdot \|x - y\|_{\boldsymbol{L}} ,$$

where the last inequality is due to the fact that,

$$\lambda_{\max}\left(\boldsymbol{L}_i^{\frac{1}{2}}\boldsymbol{M}_i\boldsymbol{L}^{-1}\boldsymbol{M}_i^{\top}\boldsymbol{L}_i^{\frac{1}{2}}\right) = \lambda_{\max}\left(\boldsymbol{L}^{-\frac{1}{2}}\boldsymbol{M}_i^{\top}\boldsymbol{L}_i\boldsymbol{M}_i\boldsymbol{L}^{-\frac{1}{2}}\right).$$

Recalling the condition of the proposition:

$$\sum_{i=1}^{s}\lambda_{\max}\left(\boldsymbol{L}_i^{\frac{1}{2}}\boldsymbol{M}_i\boldsymbol{L}^{-1}\boldsymbol{M}_i^{\top}\boldsymbol{L}_i^{\frac{1}{2}}\right) = 1,$$

we deduce

$$\|\nabla f(x) - \nabla f(y)\|_{\boldsymbol{L}^{-1}} \le \|x - y\|_{\boldsymbol{L}}.$$

### C.3.5   PROOF OF PROPOSITION 5

(i) $\to$ (ii). If $f$ is $\boldsymbol{L}$-matrix smooth, then for all $x, y \in \mathbb{R}^d$, we have

$$f(x) \le f(y) + \langle \nabla f(y), x - y \rangle + \frac{1}{2}\|x - y\|_{\boldsymbol{L}}^2,$$

and

$$f(y) \le f(x) + \langle \nabla f(x), y - x \rangle + \frac{1}{2}\|x - y\|_{\boldsymbol{L}}^2.$$

Summing up these two inequalities we get

$$\langle \nabla f(x) - \nabla f(y), x - y \rangle \le \|x - y\|_{\boldsymbol{L}}^2.$$

(ii) $\to$ (i). Choose any $x, y \in \mathbb{R}^d$, and define $z = x + t(y - x)$, then we have,

$$\begin{aligned}
f(y) &= f(x) + \int_0^1 \langle \nabla f(x + t(y - x)), y - x \rangle \, \mathrm{d}t \\
&= f(x) + \int_0^1 \langle \nabla f(z), y - x \rangle \, \mathrm{d}t \\
&= f(x) + \langle \nabla f(x), y - x \rangle + \int_0^1 \langle \nabla f(z) - \nabla f(x), y - x \rangle \, \mathrm{d}t \\
&= f(x) + \langle \nabla f(x), y - x \rangle + \int_0^1 \langle \nabla f(z) - \nabla f(x), z - x \rangle \cdot \frac{1}{t} \, \mathrm{d}t.
\end{aligned}$$

Using the assumption that for any $x, z \in \mathbb{R}^d$, we have

$$\langle \nabla f(z) - \nabla f(x), z - x \rangle \le \|z - x\|_{\boldsymbol{L}}^2.$$

Plug this back into the previous identity, we obtain

$$\begin{aligned}
f(y) &\le f(x) + \langle \nabla f(x), y - x \rangle + \int_0^1 \|z - x\|_{\boldsymbol{L}}^2 \cdot \frac{1}{t} \, \mathrm{d}t \\
&= f(x) + \langle \nabla f(x), y - x \rangle + \int_0^1 \|y - x\|_{\boldsymbol{L}}^2 \cdot t \, \mathrm{d}t \\
&= f(x) + \langle \nabla f(x), y - x \rangle + \frac{1}{2}\|y - x\|_{\boldsymbol{L}}^2.
\end{aligned}$$

### C.3.6   PROOF OF PROPOSITION 6

We start with picking any two points $x, y \in \mathbb{R}^d$, using the generalized Cauchy-Schwarz inequality for dual norm, we have

$$\begin{aligned}
\langle \nabla f(x) - \nabla f(y), x - y \rangle &\le \|\nabla f(x) - \nabla f(y)\|_{\boldsymbol{L}^{-1}} \cdot \|x - y\|_{\boldsymbol{L}} \\
&\overset{(3)}{\le} \|x - y\|_{\boldsymbol{L}} \cdot \|x - y\|_{\boldsymbol{L}} \\
&= \|x - y\|_{\boldsymbol{L}}^2
\end{aligned}$$

According to Proposition 5, this indicates that function $f$ is $\boldsymbol{L}$-matrix smooth.

### C.3.7 Proof of Proposition 7

Using Proposition 5, we know that for any $x, y \in \mathbb{R}^d$, we have

$$\langle \nabla f(x) - \nabla f(y), x - y \rangle \leq \|x - y\|_{\boldsymbol{L}}^2. \tag{17}$$

Now we pick any three points $x, y, z \in \mathbb{R}^d$. Using the $\boldsymbol{L}$-smoothness of $f$, we have

$$f(x + z) \geq f(x) + \langle \nabla f(x), z \rangle + \frac{1}{2} \|z\|_{\boldsymbol{L}}^2. \tag{18}$$

Using the convexity of $f$ we have

$$\langle \nabla f(y), x + z - y \rangle \leq f(x + z) - f(y). \tag{19}$$

Combining (18) and (19), we obtain

$$\langle \nabla f(y), x + z - y \rangle \leq f(x) - f(y) + \langle \nabla f(x), z \rangle + \frac{1}{2} \|z\|_{\boldsymbol{L}}^2.$$

Rearranging terms we get

$$\langle \nabla f(y) - \nabla f(x), z \rangle - \frac{1}{2} \|z\|_{\boldsymbol{L}}^2 \leq f(x) - f(y) - \langle \nabla f(y), x - y \rangle.$$

The inequality holds for any $z$ for fixed $x$ and $y$, and the left hand side is maximized (w.r.t. $z$) when $z = \boldsymbol{L}^{-1} (\nabla f(y) - \nabla f(x))$. Plugging it in, we get

$$\frac{1}{2} \|\nabla f(x) - \nabla f(y)\|_{\boldsymbol{L}^{-1}}^2 \leq f(x) - f(y) - \langle \nabla f(y), x - y \rangle. \tag{20}$$

By symmetry we can also obtain

$$\frac{1}{2} \|\nabla f(y) - \nabla f(x)\|_{\boldsymbol{L}^{-1}}^2 \leq f(y) - f(x) - \langle \nabla f(x), y - x \rangle.$$

Adding (20) and its counterpart together, we get

$$\|\nabla f(x) - \nabla f(y)\|_{\boldsymbol{L}^{-1}}^2 \leq \langle \nabla f(x) - \nabla f(y), x - y \rangle. \tag{21}$$

Combing (21) and (17), it follows

$$\|\nabla f(x) - \nabla f(y)\|_{\boldsymbol{L}^{-1}}^2 \leq \|x - y\|_{\boldsymbol{L}}^2.$$

Note that $\boldsymbol{L}$ and $\boldsymbol{L}^{-1}$ are both positive definite matrices, so it is equivalent to

$$\|\nabla f(x) - \nabla f(y)\|_{\boldsymbol{L}^{-1}} \leq \|x - y\|_{\boldsymbol{L}}.$$

This completes the proof.

### C.3.8 Proof of Proposition 8

Let us start with picking any two points $x, y \in \mathbb{R}^d$. With the matrix $\boldsymbol{L}$-Lipschitzness of the gradient of function $f$, we have

$$\|\nabla f(x) - \nabla f(y)\|_{\boldsymbol{L}^{-1}}^2 \leq \|x - y\|_{\boldsymbol{L}}^2.$$

This implies

$$(x - y)^\top \boldsymbol{L}(x - y) - (\nabla f(x) - \nabla f(y))^\top \boldsymbol{L}^{-1} (\nabla f(x) - \nabla f(y)) \geq 0.$$

Define function $f(\boldsymbol{X}) := a^\top \boldsymbol{X} a - b^\top \boldsymbol{X}^{-1} b$ for $\boldsymbol{X} \in \mathbb{S}_{++}^d$, where $a, b \in \mathbb{R}^d$ are fixed vectors. Then $f$ is monotone increasing in $\boldsymbol{X}$. This can be shown in the following way, picking two matrices $\boldsymbol{X}_1, \boldsymbol{X}_2 \in \mathbb{S}_{++}^d$, where $\boldsymbol{X}_1 \succeq \boldsymbol{X}_2$. It is easy to see that $-\boldsymbol{X}_1^{-1} \succeq -\boldsymbol{X}_2^{-1}$, since from Fact 3 the map $\boldsymbol{X} \mapsto -\boldsymbol{X}^{-1}$ is monotone increasing for $\boldsymbol{X} \in \mathbb{S}_{++}^d$. Thus,

$$f(\boldsymbol{X}_1) - f(\boldsymbol{X}_2) = (x - y)^\top (\boldsymbol{X}_1 - \boldsymbol{X}_2)(x - y)$$
$$+ (\nabla f(x) - \nabla f(y))^\top (-\boldsymbol{X}_1^{-1} - (-\boldsymbol{X}_2^{-1})) (\nabla f(x) - \nabla f(y)) \geq 0.$$

As a result, $f(\lambda_{\max}(\boldsymbol{L}) \cdot \boldsymbol{I}_d) \geq f(\boldsymbol{L}) \geq 0$, due to the fact that $\lambda_{\max}(\boldsymbol{L}) \cdot \boldsymbol{I}_d \succeq \boldsymbol{L}$. It remains to notice that

$$f(\lambda_{\max}(\boldsymbol{L}) \cdot \boldsymbol{I}_d) = \lambda_{\max}(\boldsymbol{L}) \|x - y\|^2 - \frac{1}{\lambda_{\max}(\boldsymbol{L})} \|\nabla f(x) - \nabla f(y)\|^2 \geq 0,$$

which yields

$$\|\nabla f(x) - \nabla f(y)\|^2 \leq \lambda_{\max}^2(\boldsymbol{L}) \|x - y\|^2.$$

Since we are working with $\boldsymbol{L} \in \mathbb{S}_{++}^d$, the above inequality implies

$$\|\nabla f(x) - \nabla f(y)\| \leq \lambda_{\max}(\boldsymbol{L}) \|x - y\|.$$

## D  ANALYSIS OF DET-MARINA

### D.1  TECHNICAL LEMMAS

We first state some technical lemmas.

**Lemma 1** (Descent lemma). *Assume that function $f$ is $L$ smooth, and $x^{k+1} = x^k - D \cdot g^k$, where $D \in \mathbb{S}_{++}^d$. Then we will have*

$$f(x^{k+1}) \leq f(x^k) - \frac{1}{2} \left\| \nabla f(x^k) \right\|_D^2 + \frac{1}{2} \left\| g^k - \nabla f(x^k) \right\|_D^2 - \frac{1}{2} \left\| x^{k+1} - x^k \right\|_{D^{-1} - L}.$$

The following lemma is obtained for any sketch matrix $S \in \mathbb{S}_+^d$ and any two positive definite matrices $D$ and $L$.

**Lemma 2** (Property of sketch matrix). *For any sketch matrix $S \in \mathbb{S}_+^d$, a vector $t \in \mathbb{R}^d$, and matrices $D, L \in \mathbb{S}_{++}^d$, we have*

$$\mathbb{E}\left[ \left\| St - t \right\|_D^2 \right] \leq \lambda_{\max} \left( L^{\frac{1}{2}} \left( \mathbb{E}\left[ SDS \right] - D \right) L^{\frac{1}{2}} \right) \cdot \left\| t \right\|_{L^{-1}}^2. \tag{22}$$

**Lemma 3.** *Assume that Definition 1 holds and $h_i^0 = \nabla f_i(x^0)$, then for $h_i^{k+1}$ from Algorithm 2, we have for any $D \in \mathbb{S}_{++}^d$*

$$\left\| h^{k+1} - \nabla f(x^{k+1}) \right\|_D^2 = \left\| h_i^{k+1} - \nabla f_i(x^{k+1}) \right\|_D^2 = 0,$$

*and*

$$\left\| h_i^{k+1} - h_i^k \right\|_{L_i^{-1}}^2 \leq \left\| x^{k+1} - x^k \right\|_{L_i}^2.$$

The following lemmas describe the recurrence applied to terms in the Lyapunov function.

**Lemma 4.** *Suppose $h^{k+1}$ and $g^{k+1}$ are from Algorithm 2, then the following recurrence relation holds,*

$$\mathbb{E}\left[ \left\| g^{k+1} - h^{k+1} \right\|_D^2 \right]$$
$$\leq \frac{2\Lambda_{D,\mathcal{S}} \cdot \lambda_{\max}\left( D^{-1} \right) \cdot \lambda_{\max}\left( D \right)}{n^2} \sum_{i=1}^n \lambda_{\max}\left( L_i \right) \mathbb{E}\left[ \left\| h_i^{k+1} - h_i^k \right\|_{L_i^{-1}}^2 \right]$$
$$+ \frac{2a^2 \Lambda_{D,\mathcal{S}} \cdot \lambda_{\max}\left( D^{-1} \right)}{n^2} \sum_{i=1}^n \mathbb{E}\left[ \left\| g_i^k - h_i^k \right\|_D^2 \right] + (1-a)^2 \mathbb{E}\left[ \left\| g^k - h^k \right\|_D^2 \right], \tag{23}$$

*where $\Lambda_{D,\mathcal{S}} = \lambda_{\max}\left( \mathbb{E}\left[ S_i^k D S_i^k \right] - D \right)$ for $D \in \mathbb{S}_{++}^d$ and $S_i^k \sim \mathcal{S}$.*

**Lemma 5.** *Suppose $h_i^{k+1}$ and $g_i^{k+1}$ for $i \in [n]$ are from Algorithm 2, then the following recurrence holds,*

$$\mathbb{E}\left[ \left\| g_i^{k+1} - h_i^{k+1} \right\|_D^2 \right]$$
$$\leq \left( 2a^2 \lambda_{\max}\left( D^{-1} \right) \cdot \Lambda_{D,\mathcal{S}} + (1-a)^2 \right) \cdot \mathbb{E}\left[ \left\| g_i^k - h_i^k \right\|_D^2 \right]$$
$$+ 2\lambda_{\max}\left( D^{-1} \right) \cdot \lambda_{\max}\left( D \right) \cdot \Lambda_{D,\mathcal{S}} \cdot \lambda_{\max}\left( L_i \right) \cdot \mathbb{E}\left[ \left\| h_i^{k+1} - h_i^k \right\|_{L_i^{-1}}^2 \right].$$

### D.2  PROOF OF THEOREM 1

According to Lemma 1, we have

$$\mathbb{E}\left[ f(x^{k+1}) \right] \leq \mathbb{E}\left[ f(x^k) \right] - \mathbb{E}\left[ \frac{1}{2} \left\| \nabla f(x^k) \right\|_D^2 \right] + \mathbb{E}\left[ \frac{1}{2} \left\| g^k - \nabla f(x^k) \right\|_D^2 \right]$$
$$- \mathbb{E}\left[ \frac{1}{2} \left\| x^{k+1} - x^k \right\|_{D^{-1} - L}^2 \right]. \tag{24}$$

We then use the definition of $g^{k+1}$ to derive an upper bound for $\mathbb{E}\left[\left\|g^{k+1} - \nabla f(x^{k+1})\right\|_{\boldsymbol{D}}^2\right]$. Notice that,

$$g^{k+1} = \begin{cases} \nabla f(x^{k+1}) & \text{with probability } p, \\ g^k + \frac{1}{n}\sum_{i=1}^n \boldsymbol{S}_i^k\left(\nabla f_i(x^{k+1}) - \nabla f_i(x^k)\right) & \text{with probability } 1-p. \end{cases}$$

As a result, from the tower property,

$$\mathbb{E}\left[\left\|g^{k+1} - \nabla f(x^{k+1})\right\|_{\boldsymbol{D}}^2 \mid x^{k+1}, x^k\right]$$

$$= \mathbb{E}\left[\mathbb{E}\left[\left\|g^{k+1} - \nabla f(x^{k+1})\right\|_{\boldsymbol{D}}^2 \mid x^{k+1}, x^k, c_k\right]\right]$$

$$= p \cdot \left\|\nabla f(x^{k+1}) - \nabla f(x^{k+1})\right\|_{\boldsymbol{D}}^2$$

$$+ (1-p) \cdot \mathbb{E}\left[\left\|g^k + \frac{1}{n}\sum_{i=1}^n \boldsymbol{S}_i^k(\nabla f_i(x^{k+1}) - \nabla f_i(x^k)) - \nabla f(x^{k+1})\right\|_{\boldsymbol{D}}^2 \mid x^{k+1}, x^k\right]$$

$$= (1-p) \cdot \mathbb{E}\left[\left\|g^k + \frac{1}{n}\sum_{i=1}^n \boldsymbol{S}_i^k(\nabla f_i(x^{k+1}) - \nabla f_i(x^k)) - \nabla f(x^{k+1})\right\|_{\boldsymbol{D}}^2 \mid x^{k+1}, x^k\right].$$

Using Fact 2, we have

$$\mathbb{E}\left[\left\|g^{k+1} - \nabla f(x^{k+1})\right\|_{\boldsymbol{D}}^2 \mid x^{k+1}, x^k\right]$$

$$= (1-p) \cdot \mathbb{E}\left[\left\|\frac{1}{n}\sum_{i=1}^n \boldsymbol{S}_i^k(\nabla f_i(x^{k+1}) - \nabla f_i(x^k)) - \left(\nabla f(x^{k+1}) - \nabla f(x^k)\right)\right\|_{\boldsymbol{D}}^2 \mid x^{k+1}, x^k\right]$$

$$+ (1-p) \cdot \left\|g^k - \nabla f(x^k)\right\|_{\boldsymbol{D}}^2$$

$$= (1-p) \cdot \mathbb{E}\left[\left\|\frac{1}{n}\sum_{i=1}^n \left(\boldsymbol{S}_i^k(\nabla f_i(x^{k+1}) - \nabla f_i(x^k)) - \left(\nabla f_i(x^{k+1}) - \nabla f_i(x^k)\right)\right)\right\|_{\boldsymbol{D}}^2 \mid x^{k+1}, x^k\right]$$

$$+ (1-p) \cdot \left\|g^k - \nabla f(x^k)\right\|_{\boldsymbol{D}}^2.$$

Notice that the sketch matrix is unbiased, thus we have

$$\mathbb{E}\left[\boldsymbol{S}_i^k\left(\nabla f_i(x^{k+1}) - \nabla f_i(x^k)\right) \mid x^{k+1}, x^k\right] = \nabla f_i(x^{k+1}) - \nabla f_i(x^k),$$

and any two random vectors in the set $\{\boldsymbol{S}_i^k(\nabla f_i(x^{k+1}) - \nabla f_i(x^k))\}_{i=1}^n$ are independent from each other, if $x^{k+1}$ and $x^k$ are fixed. Therefore, we have

$$\mathbb{E}\left[\left\|g^{k+1} - \nabla f(x^{k+1})\right\|_{\boldsymbol{D}}^2 \mid x^{k+1}, x^k\right]$$

$$= \frac{1-p}{n^2}\sum_{i=1}^n \mathbb{E}\left[\left\|\boldsymbol{S}_i^k(\nabla f_i(x^{k+1}) - \nabla f_i(x^k)) - (\nabla f_i(x^{k+1}) - \nabla f_i(x^k))\right\|_{\boldsymbol{D}}^2 \mid x^{k+1}, x^k\right]$$

$$+ (1-p) \cdot \left\|g^k - \nabla f(x^k)\right\|_{\boldsymbol{D}}^2. \tag{25}$$

Lemma 2 yields

$$\mathbb{E}\left[\left\|\boldsymbol{S}_i^k(\nabla f_i(x^{k+1}) - \nabla f_i(x^k)) - (\nabla f_i(x^{k+1}) - \nabla f_i(x^k))\right\|_{\boldsymbol{D}}^2 \mid x^{k+1}, x^k\right]$$

$$\leq \lambda_{\max}\left(\boldsymbol{L}_i^{\frac{1}{2}}\left(\mathbb{E}\left[\boldsymbol{S}_i^k \boldsymbol{D} \boldsymbol{S}_i^k\right] - \boldsymbol{D}\right)\boldsymbol{L}_i^{\frac{1}{2}}\right)\left\|\nabla f_i(x^{k+1}) - \nabla f_i(x^k)\right\|_{\boldsymbol{L}_i^{-1}}^2. \tag{26}$$

Assumption 2 implies

$$\mathbb{E}\left[\left\|\boldsymbol{S}_i^k(\nabla f_i(x^{k+1}) - \nabla f_i(x^k)) - (\nabla f_i(x^{k+1}) - \nabla f_i(x^k))\right\|_{\boldsymbol{D}}^2 \mid x^{k+1}, x^k\right]$$

$$\leq \lambda_{\max}\left(\boldsymbol{L}_i^{\frac{1}{2}}\left(\mathbb{E}\left[\boldsymbol{S}_i^k \boldsymbol{D} \boldsymbol{S}_i^k\right] - \boldsymbol{D}\right)\boldsymbol{L}_i^{\frac{1}{2}}\right)\left\|x^{k+1} - x^k\right\|_{\boldsymbol{L}_i}^2. \tag{27}$$

Plugging (27) into (25), we deduce

$$\mathbb{E}\left[\left\|g^{k+1} - \nabla f(x^{k+1})\right\|_{\boldsymbol{D}}^2 \mid x^{k+1}, x^k\right]$$

$$\leq \frac{1-p}{n^2}\sum_{i=1}^n \lambda_{\max}\left(\boldsymbol{L}_i^{\frac{1}{2}}\left(\mathbb{E}\left[\boldsymbol{S}_i^k \boldsymbol{D}\boldsymbol{S}_i^k\right] - \boldsymbol{D}\right)\boldsymbol{L}_i^{\frac{1}{2}}\right)\left\|x^{k+1} - x^k\right\|_{\boldsymbol{L}_i}^2 + (1-p)\cdot\left\|g^k - \nabla f(x^k)\right\|_{\boldsymbol{D}}^2.$$

Replacing $\boldsymbol{L}_i^{-1}$ with $\boldsymbol{L}^{-1/2}\boldsymbol{L}^{1/2}\boldsymbol{L}_i^{-1}\boldsymbol{L}^{1/2}\boldsymbol{L}^{-1/2}$, we denote that

$$\lambda_i := \lambda_{\max}\left(\boldsymbol{L}_i^{\frac{1}{2}}\left(\mathbb{E}\left[\boldsymbol{S}_i^k \boldsymbol{D}\boldsymbol{S}_i^k\right] - \boldsymbol{D}\right)\boldsymbol{L}_i^{\frac{1}{2}}\right),$$

and rewrite the $\boldsymbol{L}_i$-norm in the first term of RHS by the $\boldsymbol{L}$-norm:

$$\mathbb{E}\left[\left\|g^{k+1} - \nabla f(x^{k+1})\right\|_{\boldsymbol{D}}^2 \mid x^{k+1}, x^k\right]$$

$$= \frac{1-p}{n^2}\sum_{i=1}^n \lambda_i \cdot \left(\boldsymbol{L}^{\frac{1}{2}}(x^{k+1} - x^k)\right)^\top \boldsymbol{L}^{-\frac{1}{2}}\boldsymbol{L}_i\boldsymbol{L}^{-\frac{1}{2}}\left(\boldsymbol{L}^{\frac{1}{2}}(x^{k+1} - x^k)\right)$$

$$\quad + (1-p)\left\|g^k - \nabla f(x^k)\right\|_{\boldsymbol{D}}^2$$

$$\leq \frac{1-p}{n^2}\sum_{i=1}^n \lambda_i \cdot \lambda_{\max}\left(\boldsymbol{L}^{-\frac{1}{2}}\boldsymbol{L}_i\boldsymbol{L}^{-\frac{1}{2}}\right)\left\|x^{k+1} - x^k\right\|_{\boldsymbol{L}}^2 + (1-p)\cdot\left\|g^k - \nabla f(x^k)\right\|_{\boldsymbol{D}}^2.$$

We further use Fact 5 to upper bound $\lambda_{\max}\left(\boldsymbol{L}_i^{\frac{1}{2}}\left(\mathbb{E}\left[\boldsymbol{S}_i^k \boldsymbol{D}\boldsymbol{S}_i^k\right] - \boldsymbol{D}\right)\boldsymbol{L}_i^{\frac{1}{2}}\right)$ by the product of $\lambda_{\max}\left(\boldsymbol{L}_i\right)$ and $\lambda_{\max}\left(\mathbb{E}\left[\boldsymbol{S}_i^k \boldsymbol{D}\boldsymbol{S}_i^k\right] - \boldsymbol{D}\right)$. This allows us to simplify the expression since $\lambda_{\max}\left(\mathbb{E}\left[\boldsymbol{S}_i^k \boldsymbol{D}\boldsymbol{S}_i^k\right] - \boldsymbol{D}\right)$ is independent of the index $i$. Notice that we have already defined

$$R(\boldsymbol{D}, \mathcal{S}) = \frac{1}{n}\sum_{i=1}^n \lambda_{\max}\left(\mathbb{E}\left[\boldsymbol{S}_i^k \boldsymbol{D}\boldsymbol{S}_i^k\right] - \boldsymbol{D}\right)\cdot\lambda_{\max}\left(\boldsymbol{L}_i\right)\cdot\lambda_{\max}\left(\boldsymbol{L}^{-\frac{1}{2}}\boldsymbol{L}_i\boldsymbol{L}^{-\frac{1}{2}}\right).$$

Taking expectation, using tower property and using the definition above, we deduce

$$\mathbb{E}\left[\left\|g^{k+1} - \nabla f(x^{k+1})\right\|_{\boldsymbol{D}}^2\right]$$

$$\leq \frac{(1-p)\cdot R(\boldsymbol{D}, \mathcal{S})}{n}\mathbb{E}\left[\left\|x^{k+1} - x^k\right\|_{\boldsymbol{L}}^2\right] + (1-p)\mathbb{E}\left[\left\|g^k - \nabla f(x^k)\right\|_{\boldsymbol{D}}^2\right]. \tag{28}$$

We construct the following Lyapunov function $\Phi_k$,

$$\Phi_k = f(x^k) - f^\star + \frac{1}{2p}\left\|g^k - \nabla f(x^k)\right\|_{\boldsymbol{D}}^2. \tag{29}$$

Using (24) and (28), we are able to get

$$\mathbb{E}\left[\Phi_{k+1}\right] \leq \frac{1}{2p}\left[\frac{(1-p)\cdot R(\boldsymbol{D}, \mathcal{S})}{n}\mathbb{E}\left[\left\|x^{k+1} - x^k\right\|_{\boldsymbol{L}}^2\right] + (1-p)\cdot\mathbb{E}\left[\left\|g^k - \nabla f(x^k)\right\|_{\boldsymbol{D}}^2\right]\right]$$

$$\quad + \mathbb{E}\left[f(x^k) - f^\star\right] - \frac{1}{2}\mathbb{E}\left[\left\|\nabla f(x^k)\right\|_{\boldsymbol{D}}^2\right] + \frac{1}{2}\mathbb{E}\left[\left\|g^k - \nabla f(x^k)\right\|_{\boldsymbol{D}}^2\right]$$

$$\quad - \frac{1}{2}\mathbb{E}\left[\left\|x^{k+1} - x^k\right\|_{\boldsymbol{D}^{-1}-\boldsymbol{L}}^2\right]$$

$$= \mathbb{E}\left[\Phi_k\right] - \frac{1}{2}\mathbb{E}\left[\left\|\nabla f(x^k)\right\|_{\boldsymbol{D}}^2\right]$$

$$\quad + \left(\frac{(1-p)\cdot R(\boldsymbol{D}, \mathcal{S})}{2np}\mathbb{E}\left[\left\|x^{k+1} - x^k\right\|_{\boldsymbol{L}}^2\right] - \frac{1}{2}\mathbb{E}\left[\left\|x^{k+1} - x^k\right\|_{\boldsymbol{D}^{-1}-\boldsymbol{L}}^2\right]\right)$$

$$= \mathbb{E}\left[\Phi_k\right] - \frac{1}{2}\mathbb{E}\left[\left\|\nabla f(x^k)\right\|_{\boldsymbol{D}}^2\right]$$

$$\quad + \frac{1}{2}\left(\frac{(1-p)\cdot R(\boldsymbol{D}, \mathcal{S})}{np}\mathbb{E}\left[\left\|x^{k+1} - x^k\right\|_{\boldsymbol{L}}^2\right] - \mathbb{E}\left[\left\|x^{k+1} - x^k\right\|_{\boldsymbol{D}^{-1}-\boldsymbol{L}}^2\right]\right).$$

We can rewrite the last term as

$$\mathbb{E}\left[(x^{k+1} - x^k)^\top \left[\frac{(1-p) \cdot R(\boldsymbol{D}, \mathcal{S})}{np} \boldsymbol{L} + \boldsymbol{L} - \boldsymbol{D}^{-1}\right](x^{k+1} - x^k)\right]. \tag{30}$$

We require the matrix in between to be negative semi-definite, which is

$$\boldsymbol{D}^{-1} \succeq \left(\frac{(1-p) \cdot R(\boldsymbol{D}, \mathcal{S})}{np} + 1\right)\boldsymbol{L}.$$

This leads to the result that the expression (30) is always non-positive. After dropping the last term, the relation between $\mathbb{E}\left[\Phi_{k+1}\right]$ and $\mathbb{E}\left[\Phi_k\right]$ becomes

$$\mathbb{E}\left[\Phi_{k+1}\right] \leq \mathbb{E}\left[\Phi_k\right] - \frac{1}{2}\mathbb{E}\left[\left\|\nabla f(x^k)\right\|_{\boldsymbol{D}}^2\right].$$

Unrolling this recurrence, we get

$$\frac{1}{K}\sum_{k=0}^{K-1} \mathbb{E}\left[\left\|\nabla f(x^k)\right\|_{\boldsymbol{D}}^2\right] \leq \frac{2\left(\mathbb{E}\left[\Phi_0\right] - \mathbb{E}\left[\Phi_K\right]\right)}{K}. \tag{31}$$

The left hand side can viewed as $\mathbb{E}\left[\left\|\nabla f(\tilde{x}^K)\right\|_{\boldsymbol{D}}^2\right]$, where $\tilde{x}^K$ is drawn uniformly at random from $\{x_k\}_{k=0}^{K-1}$. From $\Phi_K > 0$, we obtain

$$
\begin{aligned}
\frac{2\left(\mathbb{E}\left[\Phi_0\right] - \mathbb{E}\left[\Phi_K\right]\right)}{K} &\leq \frac{2\Phi_0}{K} \\
&= \frac{2\left(f(x^0) - f^\star + \frac{1}{2p}\left\|g^0 - \nabla f(x^0)\right\|_{\boldsymbol{D}}^2\right)}{K} \\
&= \frac{2\left(f(x^0) - f^\star\right)}{K}.
\end{aligned}
$$

Plugging in the simplified result into (31), and performing determinant normalization, we get

$$\mathbb{E}\left[\left\|\nabla f(\tilde{x}^K)\right\|_{\frac{\boldsymbol{D}}{\det(\boldsymbol{D})^{1/d}}}^2\right] \leq \frac{2\left(f(x^0) - f^\star\right)}{\det(\boldsymbol{D})^{1/d}K}. \tag{32}$$

**Remark 7.** *We can achieve a slightly more refined stepsize condition than* (4) *for det-MARINA, which is given as follows*

$$\boldsymbol{D} \succeq \left(\frac{(1-p) \cdot \tilde{R}(\boldsymbol{D}, \mathcal{S})}{np} + 1\right)\boldsymbol{L}, \tag{33}$$

*where*

$$\tilde{R}(\boldsymbol{D}, \mathcal{S}) := \frac{1}{n}\sum_{i=1}^n \lambda_{\max}\left(\boldsymbol{L}_i^{\frac{1}{2}}\left(\mathbb{E}\left[\boldsymbol{S}_i^k \boldsymbol{D} \boldsymbol{S}_i^k\right] - \boldsymbol{D}\right)\boldsymbol{L}_i^{\frac{1}{2}}\right) \cdot \lambda_{\max}\left(\boldsymbol{L}^{-\frac{1}{2}}\boldsymbol{L}_i \boldsymbol{L}^{-\frac{1}{2}}\right).$$

*This is obtained if we do not use Fact 5 to upper bound $\lambda_{\max}\left(\boldsymbol{L}_i^{\frac{1}{2}}\left(\mathbb{E}\left[\boldsymbol{S}_i^k \boldsymbol{D} \boldsymbol{S}_i^k\right] - \boldsymbol{D}\right)\boldsymbol{L}_i^{\frac{1}{2}}\right)$ by the product of $\lambda_{\max}\left(\boldsymbol{L}_i\right)$ and $\lambda_{\max}\left(\mathbb{E}\left[\boldsymbol{S}_i^k \boldsymbol{D} \boldsymbol{S}_i^k\right] - \boldsymbol{D}\right)$. However, (33) results in a condition that is much harder to solve even if we assume $\boldsymbol{D} = \gamma \cdot \boldsymbol{W}$. So instead of using the more refined condition (33), we turn to (4). Notice that both of the two conditions (33) and (4) reduce to the stepsize condition for MARINA in the scalar setting.*

### D.3 COMPARISON OF DIFFERENT STEPSIZES

In Corollary 3, we focus on the special stepsize where we fix $\boldsymbol{W} = \boldsymbol{L}^{-1}$, and show that in this case det-MARINA always beats MARINA in terms of both iteration and communication complexities. However, other choices for $\boldsymbol{W}$ are also possible. Specifically, we consider the cases where $\boldsymbol{W} = \text{diag}^{-1}(\boldsymbol{L})$ and $\boldsymbol{W} = \boldsymbol{I}_d$.

### D.3.1 THE DIAGONAL CASE

We consider $W = \mathrm{diag}^{-1}(L)$. The following corollary describes the optimal stepsize and the iteration complexity.

**Corollary 8.** *If we take $W = \mathrm{diag}^{-1}(L)$ in Corollary 1, then the optimal stepsize satisfies*

$$D^*_{\mathrm{diag}^{-1}(L)} = \frac{2}{1 + \sqrt{1 + 4\alpha\beta \cdot \Lambda_{\mathrm{diag}^{-1}(L),\mathcal{S}}}} \cdot \mathrm{diag}^{-1}(L). \tag{34}$$

*This stepsize results in a better iteration complexity of det-MARINA compared to scalar MARINA.*

From this corollary we know that det-MARINA has a better iteration complexity when $W = \mathrm{diag}^{-1}(L)$. And since the same sketch is used for MARINA and det-MARINA, the communication complexity is improved as well. However, in general there is no clear relation between the iteration complexity of $W = L^{-1}$ case and $W = \mathrm{diag}^{-1}(L)$ case. This is also confirmed by one of our experiments, see Figure 6 to see the comparison of det-MARINA using optimal stepsizes in different cases.

### D.3.2 THE IDENTITY CASE

In this setting, $W$ is the $d$-dimensional identity matrix $I_d$. Then the stepsize of our algorithm reduces to a scalar $\gamma$, where $\gamma$ is determined through Corollary 1. Notice that in this case we do not reduce to the standard MARINA case because we are still using the matrix Lipschitz gradient assumption with $L \in \mathbb{S}^d_{++}$.

**Corollary 9.** *If we take $W = I_d$, the optimal stepsize is given by*

$$D^*_{I_d} = \frac{2}{1 + \sqrt{1 + 4\alpha\beta \frac{1}{\lambda_{\max}(L)} \cdot \omega}} \cdot \frac{I_d}{\lambda_{\max}(L)}. \tag{35}$$

*This stepsize results in a better iteration complexity of det-MARINA compared to scalar MARINA.*

The result in this corollary tells us that using scalar stepsize with matrix Lipschitz gradient assumption alone can result in acceleration of MARINA. However, the use of matrix stepsize allows us to also take into consideration the "structure" of the stepsize, thus allows more flexibility. When the structure of the stepsize is chosen properly, combining matrix gradient Lipschitzness and matrix stepsize can result in a faster rate, as it can also be observed from the experiments in Figure 6. The choices of $W$ we consider here are in some sense inspired by the matrix stepsize GD, where the optimal stepsize is $L^{-1}$. In general, how to identify the best structure for the matrix stepsize remains a open problem.

### D.4 PROOFS OF THE COROLLARIES

### D.4.1 PROOF OF COROLLARY 1

We start with rewriting (4) as

$$\left(\frac{1-p}{np} \cdot R(D,\mathcal{S}) + 1\right) D^{\frac{1}{2}} L D^{\frac{1}{2}} \preceq I_d.$$

Plugging in the definition of $R(D,\mathcal{S})$ and $D = \gamma W$, we get

$$\gamma\left(\frac{1-p}{np} \cdot \frac{1}{n}\sum_{i=1}^{n} \lambda_{\max}(L_i)\lambda_{\max}(L^{-1}L_i) \cdot \lambda_{\max}\left(\mathbb{E}\left[S_i^k W S_i^k\right] - W\right) \cdot \gamma + 1\right) W^{\frac{1}{2}} L W^{\frac{1}{2}} \preceq I_d.$$

This generalized inequality is equivalent to the following inequality,

$$\gamma\left(\frac{1-p}{np} \cdot \frac{1}{n}\sum_{i=1}^{n} \lambda_{\max}(L_i)\lambda_{\max}(L^{-1}L_i) \cdot \lambda_{\max}\left(\mathbb{E}\left[S_i^k W S_i^k\right] - W\right) \cdot \gamma + 1\right) \cdot \lambda_{\max}\left(W^{\frac{1}{2}} L W^{\frac{1}{2}}\right) \leq 1,$$

which is a quadratic inequality on $\gamma$. Notice that we have already defined

$$\alpha = \frac{1-p}{np}; \qquad \beta = \frac{1}{n}\sum_{i=1}^{n} \lambda_{\max}\left(\boldsymbol{L}_i\right) \cdot \lambda_{\max}\left(\boldsymbol{L}^{-1}\boldsymbol{L}_i\right);$$

$$\Lambda_{\boldsymbol{W},\mathcal{S}} = \lambda_{\max}\left(\mathbb{E}\left[\boldsymbol{S}_i^k \boldsymbol{W} \boldsymbol{S}_i^k\right] - \boldsymbol{W}\right); \qquad \lambda_{\boldsymbol{W}} = \lambda_{\max}^{-1}\left(\boldsymbol{W}^{\frac{1}{2}}\boldsymbol{L}\boldsymbol{W}^{\frac{1}{2}}\right).$$

As a result, the above inequality can be written equivalently as

$$\alpha\beta\Lambda_{\boldsymbol{W},\mathcal{S}} \cdot \gamma^2 + \gamma - \lambda_{\boldsymbol{W}} \le 0,$$

which yields the upper bound on $\gamma$

$$\gamma \le \frac{\sqrt{1 + 4\alpha\beta \cdot \Lambda_{\boldsymbol{W},\mathcal{S}}\lambda_{\boldsymbol{W}}} - 1}{2\alpha\beta \cdot \Lambda_{\boldsymbol{W},\mathcal{S}}}.$$

Since $\sqrt{1 + 4\alpha\beta \cdot \Lambda_{\boldsymbol{W},\mathcal{S}}\lambda_{\boldsymbol{W}}} + 1 > 0$, we can simplify the result as

$$\gamma \le \frac{2\lambda_{\boldsymbol{W}}}{1 + \sqrt{1 + 4\alpha\beta \cdot \Lambda_{\boldsymbol{W},\mathcal{S}}\lambda_{\boldsymbol{W}}}}.$$

### D.4.2 PROOF OF COROLLARY 3

It is obvious that (7) directly follows from plugging $\boldsymbol{W} = \boldsymbol{L}^{-1}$ into (6). The optimal stepsize is obtained as the product of $\gamma$ and $\boldsymbol{L}^{-1}$. The iteration complexity of MARINA, according to Gorbunov et al. (2021), is

$$K \ge K_1 = \mathcal{O}\left(\frac{\Delta_0 L}{\varepsilon^2}\left(1 + \sqrt{\frac{(1-p)\omega}{pn}}\right)\right). \tag{36}$$

On the other hand,

$$\det(\boldsymbol{L})^{\frac{1}{d}} \le \lambda_{\max}\left(\boldsymbol{L}\right) = L. \tag{37}$$

In addition, using the inequality

$$\sqrt{1 + 4t} \le 1 + 2\sqrt{t}, \tag{38}$$

which holds for any $t \ge 0$, we have the following bound

$$\frac{\left(1 + \sqrt{1 + 4\alpha\beta \cdot \Lambda_{\boldsymbol{L}^{-1},\mathcal{S}}}\right)}{2} \le 1 + \sqrt{\alpha\beta \cdot \Lambda_{\boldsymbol{L}^{-1},\mathcal{S}}}.$$

Next we prove that

$$1 + \sqrt{\alpha\beta \cdot \Lambda_{\boldsymbol{L}^{-1},\mathcal{S}}} \le 1 + \sqrt{\frac{(1-p)}{pn} \cdot \omega}, \tag{39}$$

which is equivalent to proving

$$\frac{1}{n}\sum_{i=1}^{n} \lambda_{\max}\left(\boldsymbol{L}_i\right) \lambda_{\max}\left(\boldsymbol{L}_i \boldsymbol{L}^{-1}\right) \cdot \lambda_{\max}\left(\mathbb{E}\left[\boldsymbol{S}_i^k \boldsymbol{L}^{-1} \boldsymbol{S}_i^k\right] - \boldsymbol{L}^{-1}\right) \le \omega.$$

The left hand side can be upper bounded by,

$$\frac{1}{n}\sum_{i=1}^{n} \lambda_{\max}\left(\boldsymbol{L}_i\right) \lambda_{\max}\left(\boldsymbol{L}^{-1}\boldsymbol{L}_i\right) \cdot \lambda_{\max}\left(\boldsymbol{L}^{-1}\right) \cdot \frac{\lambda_{\max}\left(\mathbb{E}\left[\boldsymbol{S}_i^k \boldsymbol{L}^{-1} \boldsymbol{S}_i^k\right] - \boldsymbol{L}^{-1}\right)}{\lambda_{\max}\left(\boldsymbol{L}^{-1}\right)}$$

$$\le \frac{\lambda_{\max}\left(\mathbb{E}\left[\boldsymbol{S}_i^k \boldsymbol{L}^{-1} \boldsymbol{S}_i^k\right] - \boldsymbol{L}^{-1}\right)}{\lambda_{\max}\left(\boldsymbol{L}^{-1}\right)},$$

where the inequality is a consequence of Proposition 1. We further bound the last term with

$$\frac{\lambda_{\max}\left(\mathbb{E}\left[\boldsymbol{S}_i^k \boldsymbol{L}^{-1} \boldsymbol{S}_i^k\right] - \boldsymbol{L}^{-1}\right)}{\lambda_{\max}\left(\boldsymbol{L}^{-1}\right)} = \lambda_{\max}\left(\mathbb{E}\left[\boldsymbol{S}_i^k \cdot \frac{\boldsymbol{L}^{-1}}{\lambda_{\max}(\boldsymbol{L}^{-1})} \cdot \boldsymbol{S}_i^k\right] - \frac{\boldsymbol{L}^{-1}}{\lambda_{\max}\left(\boldsymbol{L}^{-1}\right)}\right)$$

$$\le \lambda_{\max}\left(\mathbb{E}\left[\boldsymbol{S}_i^k \boldsymbol{S}_i^k\right] - \boldsymbol{I}_d\right) =: \omega.$$

Here, the last inequality is due to the monotonicity of the mapping $\boldsymbol{X} \mapsto \lambda_{\max}\left(\mathbb{E}\left[\boldsymbol{S}_i^k \boldsymbol{X} \boldsymbol{S}_i^k\right] - \boldsymbol{X}\right)$ with $\boldsymbol{X} \in \mathbb{S}_{++}^d$, which can be shown as follows, let us pick any $\boldsymbol{X}_1, \boldsymbol{X}_2 \in \mathbb{S}_{++}^d$ and $\boldsymbol{X}_1 \preceq \boldsymbol{X}_2$,

$$\left(\mathbb{E}\left[\boldsymbol{S}_i^k \boldsymbol{X}_2 \boldsymbol{S}_i^k\right] - \boldsymbol{X}_2\right) - \left(\mathbb{E}\left[\boldsymbol{S}_i^k \boldsymbol{X}_1 \boldsymbol{S}_i^k\right] - \boldsymbol{X}_1\right) = \mathbb{E}\left[\boldsymbol{S}_i^k \left(\boldsymbol{X}_2 - \boldsymbol{X}_1\right) \boldsymbol{S}_i^k\right] - \left(\boldsymbol{X}_2 - \boldsymbol{X}_1\right) \succeq \boldsymbol{O}_d.$$

The above inequality is due to the convexity of the mapping $\boldsymbol{S}_i^k \mapsto \boldsymbol{S}_i^k \boldsymbol{X} \boldsymbol{S}_i^k$. As a result, we have

$$\lambda_{\max}\left(\mathbb{E}\left[\boldsymbol{S}_i^k \boldsymbol{X}_2 \boldsymbol{S}_i^k\right] - \boldsymbol{X}_2\right) \geq \lambda_{\max}\left(\mathbb{E}\left[\boldsymbol{S}_i^k \boldsymbol{X}_1 \boldsymbol{S}_i^k\right] - \boldsymbol{X}_1\right),$$

whenever $\boldsymbol{X}_2 \succeq \boldsymbol{X}_1$. Due to the fact that

$$\frac{\boldsymbol{L}^{-1}}{\lambda_{\max}\left(\boldsymbol{L}^{-1}\right)} \preceq \boldsymbol{I}_d,$$

we have

$$\lambda_{\max}\left(\mathbb{E}\left[\boldsymbol{S}_i^k \cdot \frac{\boldsymbol{L}^{-1}}{\lambda_{\max}(\boldsymbol{L}^{-1})} \cdot \boldsymbol{S}_i^k\right] - \frac{\boldsymbol{L}^{-1}}{\lambda_{\max}\left(\boldsymbol{L}^{-1}\right)}\right) \leq \lambda_{\max}\left(\mathbb{E}\left[\boldsymbol{S}_i^k \cdot \boldsymbol{I}_d \cdot \boldsymbol{S}_i^k\right] - \boldsymbol{I}_d\right) = \omega.$$

Combining (37) and (39), we know that the iteration complexity of det-MARINA is always better than that of MARINA.

### D.4.3 PROOF OF COROLLARY 4

The number of bits sent in expectation is

$$\mathcal{O}(d + K(pd + (1-p)\zeta_{\mathcal{S}})) = \mathcal{O}((Kp+1)d + (1-p)K\zeta_{\mathcal{S}}).$$

The special case where we choose $p = \zeta_{\mathcal{S}}/d$ indicates that

$$\alpha = \frac{1-p}{np} = \frac{1}{n}\left(\frac{d}{\zeta_{\mathcal{S}}} - 1\right).$$

In order to reach an error of $\varepsilon^2$, we need

$$K = \mathcal{O}\left(\frac{\Delta_0 \cdot \det(\boldsymbol{L})^{\frac{1}{d}}}{\varepsilon^2} \cdot \left(1 + \sqrt{1 + \frac{4\beta}{n}\left(\frac{d}{\zeta_{\mathcal{S}}} - 1\right) \cdot \Lambda_{\boldsymbol{L}^{-1},\mathcal{S}}}\right)\right),$$

which is the iteration complexity. Applying once again (38) and using the fact that $p = \zeta_{\mathcal{S}}/d$, the communication complexity in this case is given by

$$\mathcal{O}\left(d + \frac{\Delta_0 \cdot \det(\boldsymbol{L})^{\frac{1}{d}}}{\varepsilon^2} \cdot \left(1 + \sqrt{1 + \frac{4\beta}{n}\left(\frac{d}{\zeta_{\mathcal{S}}} - 1\right) \cdot \Lambda_{\boldsymbol{L}^{-1},\mathcal{S}}}\right) \cdot (pd + (1-p)\zeta_{\mathcal{S}})\right)$$

$$\leq \mathcal{O}\left(d + \frac{2\Delta_0 \cdot \det(\boldsymbol{L})^{\frac{1}{d}}}{\varepsilon^2} \cdot \left(1 + \sqrt{\frac{\beta}{n}\left(\frac{d}{\zeta_{\mathcal{S}}} - 1\right) \cdot \Lambda_{\boldsymbol{L}^{-1},\mathcal{S}}}\right) \cdot (pd + (1-p)\zeta_{\mathcal{S}})\right)$$

$$\leq \mathcal{O}\left(d + \frac{4\Delta_0 \cdot \det(\boldsymbol{L})^{\frac{1}{d}}}{\varepsilon^2} \cdot \left(\zeta_{\mathcal{S}} + \sqrt{\frac{\beta \cdot \Lambda_{\boldsymbol{L}^{-1},\mathcal{S}}}{n} \cdot \zeta_{\mathcal{S}}(d - \zeta_{\mathcal{S}})}\right)\right).$$

Ignoring the coefficient we get

$$\mathcal{O}\left(d + \frac{\Delta_0 \cdot \det(\boldsymbol{L})^{\frac{1}{d}}}{\varepsilon^2} \cdot \left(\zeta_{\mathcal{S}} + \sqrt{\frac{\beta \cdot \Lambda_{\boldsymbol{L}^{-1},\mathcal{S}}}{n} \cdot \zeta_{\mathcal{S}}(d - \zeta_{\mathcal{S}})}\right)\right).$$

### D.4.4 PROOF OF COROLLARY 8

Applying Corollary 1, notice that in this case

$$\lambda_{\mathrm{diag}^{-1}(\boldsymbol{L})} = \lambda_{\max}^{-1}\left(\mathrm{diag}^{-\frac{1}{2}}(\boldsymbol{L}) \boldsymbol{L} \, \mathrm{diag}^{-\frac{1}{2}}(\boldsymbol{L})\right) = 1,$$

we obtain $\boldsymbol{D}_{\mathrm{diag}^{-1}(\boldsymbol{L})}^*$. The iteration complexity is given by

$$\mathcal{O}\left(\frac{\det\left(\mathrm{diag}(\boldsymbol{L})\right)^{\frac{1}{d}} \cdot \Delta_0}{\varepsilon^2} \cdot \left(\frac{1 + \sqrt{1 + 4\alpha\beta\Lambda_{\mathrm{diag}^{-1}(\boldsymbol{L}),\mathcal{S}}}}{2}\right)\right).$$

We now compare it to the iteration complexity of MARINA, which is given in (36). We know that each diagonal element $\boldsymbol{L}_{jj}$ satisfies $\boldsymbol{L}_{jj} \leq \lambda_{\max}(\boldsymbol{L}) = L$ for $j = 1, \ldots, d$. As a result,

$$\det(\mathrm{diag}(\boldsymbol{L}))^{\frac{1}{d}} \leq L. \tag{40}$$

From (38), we deduce

$$\frac{1 + \sqrt{1 + 4\alpha\beta \cdot \Lambda_{\mathrm{diag}^{-1}(\boldsymbol{L}),\mathcal{S}}}}{2} \leq 1 + \sqrt{\alpha\beta \cdot \Lambda_{\mathrm{diag}^{-1}(\boldsymbol{L}),\mathcal{S}}}.$$

Now, let us prove the below inequality

$$1 + \sqrt{\alpha\beta \cdot \Lambda_{\mathrm{diag}^{-1}(\boldsymbol{L}),\mathcal{S}}} \leq 1 + \sqrt{\frac{(1-p)}{pn} \cdot \omega}. \tag{41}$$

The latter is equivalent to

$$\beta \cdot \Lambda_{\mathrm{diag}^{-1}(\boldsymbol{L}),\mathcal{S}} \leq \omega.$$

Plugging in the definition of $\beta$, $\omega$ and $\Lambda_{\mathrm{diag}^{-1}(\boldsymbol{L}),\mathcal{S}}$ and using the relation given in Proposition 1, we obtain,

$$\lambda_{\max}\left(\mathbb{E}\left[\boldsymbol{S}_i^k \frac{\mathrm{diag}^{-1}(\boldsymbol{L})}{\lambda_{\max}(\boldsymbol{L}^{-1})} \boldsymbol{S}_i^k - \frac{\mathrm{diag}^{-1}(\boldsymbol{L})}{\lambda_{\max}(\boldsymbol{L}^{-1})}\right]\right) \leq \lambda_{\max}\left(\mathbb{E}\left[\boldsymbol{S}_i^k \boldsymbol{I}_d \boldsymbol{S}_i^k\right] - \boldsymbol{I}_d\right).$$

Thus, it is enough to prove that

$$\frac{\mathrm{diag}^{-1}(\boldsymbol{L})}{\lambda_{\max}(\boldsymbol{L}^{-1})} \preceq \boldsymbol{I}_d.$$

We can further simplify the above inequality as

$$\lambda_{\min}(\boldsymbol{L}) \leq \lambda_{\min}(\mathrm{diag}(\boldsymbol{L})),$$

which is always true for any $\boldsymbol{L} \in \mathbb{S}_{++}^d$. Combining (40) and (41) we conclude the proof.

### D.4.5 PROOF OF COROLLARY 9

Using the explicit formula for the optimal stepsize $\boldsymbol{D}_{\boldsymbol{I}_d}^*$, we deduce the following iteration complexity for

$$\mathcal{O}\left(\frac{\lambda_{\max}(\boldsymbol{L})\Delta_0}{\varepsilon^2} \cdot \left(\frac{1 + \sqrt{1 + 4\alpha\beta \frac{\omega}{\lambda_{\max}(\boldsymbol{L})}}}{2}\right)\right). \tag{42}$$

Recall that $\lambda_{\max}(\boldsymbol{L}) = L$, we obtain using (38) that

$$\frac{1 + \sqrt{1 + 4\alpha\beta \frac{\omega}{\lambda_{\max}(\boldsymbol{L})}}}{2} \leq 1 + \sqrt{\alpha\beta \frac{\omega}{\lambda_{\max}(\boldsymbol{L})}}.$$

The comparison of two iteration complexities, given in (42) and (36) reduces to

$$1 + \sqrt{\alpha\beta \frac{\omega}{\lambda_{\max}(\boldsymbol{L})}} \leq 1 + \sqrt{\frac{1-p}{np}\omega}.$$

This is equivalent to

$$\beta \cdot \frac{1}{\lambda_{\max}(\boldsymbol{L})} \leq 1.$$

Utilizing Proposition 1, the above inequality can be rewritten as

$$\frac{1}{\lambda_{\max}(\boldsymbol{L}^{-1}) \cdot \lambda_{\max}(\boldsymbol{L})} \leq 1,$$

which is exactly

$$\lambda_{\min}(\boldsymbol{L}) \leq \lambda_{\max}(\boldsymbol{L}).$$

## E  ANALYSIS OF DET-DASHA

### E.1  PROOF OF THEOREM 2

Using Lemma 1 and taking expectations, we are able to obtain

$$
\mathbb{E}\left[f(x^{k+1})\right]
$$
$$
\leq \mathbb{E}\left[f(x^k)\right] - \frac{1}{2}\mathbb{E}\left[\left\|\nabla f(x^k)\right\|_{\boldsymbol{D}}^2\right] - \frac{1}{2}\mathbb{E}\left[\left\|x^{k+1} - x^k\right\|_{\boldsymbol{D}^{-1}-\boldsymbol{L}}^2\right] + \frac{1}{2}\mathbb{E}\left[\left\|g^k - \nabla f(x^k)\right\|_{\boldsymbol{D}}^2\right]
$$
$$
\leq \mathbb{E}\left[f(x^k)\right] - \frac{1}{2}\mathbb{E}\left[\left\|\nabla f(x^k)\right\|_{\boldsymbol{D}}^2\right] - \frac{1}{2}\mathbb{E}\left[\left\|x^{k+1} - x^k\right\|_{\boldsymbol{D}^{-1}-\boldsymbol{L}}^2\right]
$$
$$
+ \mathbb{E}\left[\frac{1}{2}\left\|g^k - h^k + h^k - \nabla f(x^k)\right\|_{\boldsymbol{D}}^2\right]
$$
$$
\leq \mathbb{E}\left[f(x^k)\right] - \frac{1}{2}\mathbb{E}\left[\left\|\nabla f(x^k)\right\|_{\boldsymbol{D}}^2\right] - \frac{1}{2}\mathbb{E}\left[\left\|x^{k+1} - x^k\right\|_{\boldsymbol{D}^{-1}-\boldsymbol{L}}^2\right]
$$
$$
+ \mathbb{E}\left[\left\|g^k - h^k\right\|_{\boldsymbol{D}}^2 + \left\|h^k - \nabla f(x^k)\right\|_{\boldsymbol{D}}^2\right], \tag{43}
$$

where the last step is due to the convexity of the norm. Using Lemma 4, we obtain

$$
\mathbb{E}\left[\left\|g^{k+1} - h^{k+1}\right\|_{\boldsymbol{D}}^2\right] \leq \frac{2\omega_{\boldsymbol{D}}\cdot\lambda_{\max}\left(\boldsymbol{D}\right)}{n^2}\sum_{i=1}^n \lambda_{\max}\left(\boldsymbol{L}_i\right)\mathbb{E}\left[\left\|h_i^{k+1} - h_i^k\right\|_{\boldsymbol{L}_i^{-1}}^2\right]
$$
$$
+ \frac{2a^2\omega_{\boldsymbol{D}}}{n^2}\sum_{i=1}^n \mathbb{E}\left[\left\|g_i^k - h_i^k\right\|_{\boldsymbol{D}}^2\right] + (1-a)^2\mathbb{E}\left[\left\|g^k - h^k\right\|_{\boldsymbol{D}}^2\right]. \tag{44}
$$

Using Lemma 5, we get

$$
\mathbb{E}\left[\left\|g_i^{k+1} - h_i^{k+1}\right\|_{\boldsymbol{D}}^2\right] \leq \left(2a^2\omega_{\boldsymbol{D}} + (1-a)^2\right)\cdot\mathbb{E}\left[\left\|g_i^k - h_i^k\right\|_{\boldsymbol{D}}^2\right]
$$
$$
+ 2\omega_{\boldsymbol{D}}\cdot\lambda_{\max}\left(\boldsymbol{D}\right)\cdot\lambda_{\max}\left(\boldsymbol{L}_i\right)\cdot\mathbb{E}\left[\left\|h_i^{k+1} - h_i^k\right\|_{\boldsymbol{L}_i^{-1}}^2\right]. \tag{45}
$$

Now let us fix $\kappa \in [0, +\infty), \eta \in [0, +\infty)$ which we will determine later, and construct the following Lyapunov function $\Phi_k$

$$
\Phi_k = \mathbb{E}\left[f(x^k) - f^\star\right] + \kappa\cdot\mathbb{E}\left[\left\|g^k - h^k\right\|_{\boldsymbol{D}}^2\right] + \eta\cdot\mathbb{E}\left[\frac{1}{n}\sum_{i=1}^n\left\|g_i^k - h_i^k\right\|_{\boldsymbol{D}}^2\right]. \tag{46}
$$

Combining (43), (44) and (45), we get

$$
\Phi_{k+1}
$$
$$
\leq \mathbb{E}\left[f(x^k) - f^\star - \frac{1}{2}\left\|\nabla f(x^k)\right\|_{\boldsymbol{D}}^2\right]
$$
$$
+ \mathbb{E}\left[-\frac{1}{2}\left\|x^{k+1} - x^k\right\|_{\boldsymbol{D}^{-1}-\boldsymbol{L}}^2 + \left\|g^k - h^k\right\|_{\boldsymbol{D}}^2 + \left\|h^k - \nabla f(x^k)\right\|_{\boldsymbol{D}}^2\right]
$$
$$
+ \kappa(1-a)^2\mathbb{E}\left[\left\|g^k - h^k\right\|_{\boldsymbol{D}}^2\right] + \frac{2\kappa\cdot\omega_{\boldsymbol{D}}\lambda_{\max}\left(\boldsymbol{D}\right)}{n}\cdot\frac{1}{n}\sum_{i=1}^n\lambda_{\max}\left(\boldsymbol{L}_i\right)\mathbb{E}\left[\left\|h_i^{k+1} - h_i^k\right\|_{\boldsymbol{L}_i^{-1}}^2\right]
$$
$$
+ \frac{2a^2\omega_{\boldsymbol{D}}\cdot\kappa}{n}\cdot\frac{1}{n}\sum_{i=1}^n\mathbb{E}\left[\left\|g_i^k - h_i^k\right\|_{\boldsymbol{D}}^2\right] + \eta\left(2a^2\omega_{\boldsymbol{D}} + (1-a)^2\right)\cdot\frac{1}{n}\sum_{i=1}^n\mathbb{E}\left[\left\|g_i^k - h_i^k\right\|_{\boldsymbol{D}}^2\right]
$$
$$
+ 2\eta\cdot\omega_{\boldsymbol{D}}\cdot\lambda_{\max}\left(\boldsymbol{D}\right)\cdot\frac{1}{n}\sum_{i=1}^n\lambda_{\max}\left(\boldsymbol{L}_i\right)\cdot\mathbb{E}\left[\left\|h_i^{k+1} - h_i^k\right\|_{\boldsymbol{L}_i^{-1}}^2\right].
$$

Rearranging terms, and notice that $\left\|h^k - \nabla f(x^k)\right\|_D^2 = 0$,

$$\Phi_{k+1}$$

$$\leq \mathbb{E}\left[f(x^k) - f^\star\right] - \frac{1}{2}\mathbb{E}\left[\left\|\nabla f(x^k)\right\|_D^2\right]$$

$$- \frac{1}{2}\mathbb{E}\left[\left\|x^{k+1} - x^k\right\|_{D^{-1}-L}^2\right] + \left(1 + \kappa(1-a)^2\right)\mathbb{E}\left[\left\|g^k - h^k\right\|_D^2\right]$$

$$+ \left(\frac{2a^2\omega_D \cdot \kappa}{n} + \eta\left(2a^2\omega_D + (1-a)^2\right)\right) \cdot \frac{1}{n}\sum_{i=1}^n \mathbb{E}\left[\left\|g_i^k - h_i^k\right\|_D^2\right]$$

$$+ \left(\frac{2\kappa \cdot \omega_D \lambda_{\max}(D)}{n} + 2\eta \cdot \omega_D \cdot \lambda_{\max}(D)\right) \cdot \frac{1}{n}\sum_{i=1}^n \lambda_{\max}(L_i) \cdot \mathbb{E}\left[\left\|h_i^{k+1} - h_i^k\right\|_{L_i^{-1}}^2\right].$$

In order to proceed, we consider the choice of $\kappa$ and $\eta$, for $\kappa$,

$$1 + \kappa(1-a)^2 \leq \kappa. \tag{47}$$

It is then clear that the choice of $\kappa = \frac{1}{a}$ satisfies the condition. On the other hand, we look at the terms involving $\mathbb{E}\left[\left\|g_i^k - h_i^k\right\|_D^2\right]$, we can rewrite as

$$T_1 := \left(\frac{2a^2\omega_D \cdot \kappa}{n} + \eta\left(2a^2\omega_D + (1-a)^2\right)\right) \cdot \frac{1}{n}\sum_{i=1}^n \mathbb{E}\left[\left\|g_i^k - h_i^k\right\|_D^2\right].$$

Picking $\kappa = \frac{1}{a}$ and $a = \frac{1}{2\omega_D+1}$, the $T_1$ can be simplified as

$$T_1 = \left(\frac{2\omega_D}{n \cdot (2\omega_D + 1)} + \eta \cdot \frac{4\omega_D^2 + 2\omega_D}{(2\omega_D + 1)^2}\right) \cdot \frac{1}{n}\sum_{i=1}^n \mathbb{E}\left[\left\|g_i^k - h_i^k\right\|_D^2\right].$$

We pick $\eta$ so that it satisfies

$$\left(\frac{2\omega_D}{n \cdot (2\omega_D + 1)} + \eta \cdot \frac{4\omega_D^2 + 2\omega_D}{(2\omega_D + 1)^2}\right) \leq \eta. \tag{48}$$

Taking $\eta = \frac{2\omega_D}{n}$, which is the minimum value satisfying (48), we conclude that

$$T_1 \leq \eta \cdot \frac{1}{n}\sum_{i=1}^n \mathbb{E}\left[\left\|g_i^k - h_i^k\right\|_D^2\right]. \tag{49}$$

Combining (47) and (49), we are able to conclude that

$$\Phi_{k+1}$$

$$\leq \mathbb{E}\left[f(x^k) - f^\star\right] + \kappa \cdot \mathbb{E}\left[\left\|g^k - h^k\right\|_D^2\right] + \eta \cdot \frac{1}{n}\sum_{i=1}^n \mathbb{E}\left[\left\|g_i^k - h_i^k\right\|_D^2\right]$$

$$- \frac{1}{2}\mathbb{E}\left[\left\|\nabla f(x^k)\right\|_D^2\right] - \frac{1}{2}\mathbb{E}\left[\left\|x^{k+1} - x^k\right\|_{D^{-1}-L}^2\right]$$

$$+ \left(\frac{2\kappa \cdot \omega_D \lambda_{\max}(D)}{n} + 2\eta \cdot \omega_D \cdot \lambda_{\max}(D)\right) \cdot \frac{1}{n}\sum_{i=1}^n \lambda_{\max}(L_i) \cdot \mathbb{E}\left[\left\|h_i^{k+1} - h_i^k\right\|_{L_i^{-1}}^2\right].$$

Using the definition of $\Phi_k$ and Lemma 3, we obtain

$$\Phi_{k+1} \leq \Phi_k - \frac{1}{2}\mathbb{E}\left[\left\|\nabla f(x^k)\right\|_D^2\right] - \frac{1}{2}\mathbb{E}\left[\left\|x^{k+1} - x^k\right\|_{D^{-1}-L}^2\right]$$

$$\left(\frac{2\kappa \cdot \omega_D \lambda_{\max}(D)}{n} + 2\eta \cdot \omega_D \cdot \lambda_{\max}(D)\right) \cdot \frac{1}{n}\sum_{i=1}^n \lambda_{\max}(L_i) \cdot \mathbb{E}\left[\left\|x^{k+1} - x^k\right\|_{L_i}^2\right]$$

$$= \Phi_k - \frac{1}{2}\mathbb{E}\left[\left\|\nabla f(x^k)\right\|_D^2\right] + \mathbb{E}\left[\left\|x^{k+1} - x^k\right\|_N^2\right],$$

where $\boldsymbol{N} \in \mathbb{S}^d$ is defined as

$$\boldsymbol{N} := \left( \frac{2\kappa \cdot \omega_{\boldsymbol{D}} \lambda_{\max} (\boldsymbol{D})}{n} + 2\eta \cdot \omega_{\boldsymbol{D}} \cdot \lambda_{\max} (\boldsymbol{D}) \right) \cdot \frac{1}{n} \sum_{i=1}^n \lambda_{\max} (\boldsymbol{L}_i) \cdot \boldsymbol{L}_i - \frac{1}{2} \boldsymbol{D}^{-1} + \frac{1}{2} \boldsymbol{L}.$$

We require $\boldsymbol{N} \preceq \boldsymbol{O}_d$, which leads to the condition on $\boldsymbol{D}$:

$$\boldsymbol{D}^{-1} - \boldsymbol{L} - \frac{4\lambda_{\max} (\boldsymbol{D}) \cdot \omega_{\boldsymbol{D}} \cdot (4\omega_{\boldsymbol{D}} + 1)}{n} \cdot \frac{1}{n} \sum_{i=1}^n \lambda_{\max} (\boldsymbol{L}_i) \cdot \boldsymbol{L}_i \succeq \boldsymbol{O}_d.$$

Given the above condition is satisfied, we have the recurrence

$$\frac{1}{2} \mathbb{E} \left[ \left\| \nabla f(x^k) \right\|_{\boldsymbol{D}}^2 \right] \leq \Phi_k - \Phi_{k+1}$$

Summing up for $k = 0 \ldots K - 1$, we obtain

$$\sum_{k=0}^{K-1} \mathbb{E} \left[ \left\| \nabla f(x^k) \right\|_{\boldsymbol{D}}^2 \right] \leq 2(\Phi_0 - \Phi_k). \tag{50}$$

Notice that we also have

$$\Phi_0 = f(x^0) - f^\star + (2\omega_{\boldsymbol{D}} + 1) \left\| g^0 - h^0 \right\|_{\boldsymbol{D}}^2 + frac{2}\omega_{\boldsymbol{D}} n \cdot \frac{1}{n} \sum_{i=1}^n \left\| g_i^0 - h_i^0 \right\|^2$$

$$= f(x^0) - f^\star,$$

We divide both sides of (50) by $K$, and perform determinant normalization,

$$\frac{1}{K} \sum_{k=0}^{K-1} \mathbb{E} \left[ \left\| \nabla f(x^k) \right\|_{\frac{\boldsymbol{D}}{\det(\boldsymbol{D})^{1/d}}}^2 \right] \leq \frac{2(f(x^0) - f^\star)}{\det(\boldsymbol{D})^{1/d} \cdot K}.$$

This is to say

$$\mathbb{E} \left[ \left\| \nabla f(\tilde{x}^K) \right\|_{\frac{\boldsymbol{D}}{\det(\boldsymbol{D})^{1/d}}}^2 \right] \leq \frac{2(f(x^0) - f^\star)}{\det(\boldsymbol{D})^{1/d} \cdot K},$$

where $\tilde{x}^K$ is chosen uniformly randomly from the first K iterates of the algorithm.

### E.2 Proofs of the Corollaries

#### E.2.1 Proof of Corollary 2

Plug $\boldsymbol{D} = \gamma_{\boldsymbol{W}} \cdot \boldsymbol{W}$ into the stepsize condition in Theorem 2, we obtain

$$\frac{1}{\gamma_{\boldsymbol{W}}} \cdot \boldsymbol{W}^{-1} - \boldsymbol{L} - \frac{4\gamma_{\boldsymbol{W}} \cdot \lambda_{\max} (\boldsymbol{W}) \cdot \omega_{\boldsymbol{W}} (4\omega_{\boldsymbol{W}} + 1)}{n} \cdot \frac{1}{n} \sum_{i=1}^n \lambda_{\max} (\boldsymbol{L}_i) \cdot \boldsymbol{L}_i \succeq \boldsymbol{O}_d.$$

We then simplify the above condition as

$$\frac{1}{\gamma_{\boldsymbol{W}}} \cdot \boldsymbol{L}^{-\frac{1}{2}} \boldsymbol{W}^{-1} \boldsymbol{L}^{-\frac{1}{2}}$$

$$\succeq \boldsymbol{I}_d + \frac{4\gamma_{\boldsymbol{W}} \cdot \lambda_{\max} (\boldsymbol{W}) \cdot \omega_{\boldsymbol{W}} (4\omega_{\boldsymbol{W}} + 1)}{n} \cdot \boldsymbol{L}^{-\frac{1}{2}} \left( \frac{1}{n} \sum_{i=1}^n \lambda_{\max} (\boldsymbol{L}_i) \cdot \boldsymbol{L}_i \right) \boldsymbol{L}^{-\frac{1}{2}}.$$

Using Proposition 3, we have

$$\frac{1}{\gamma_{\boldsymbol{W}}} \cdot \boldsymbol{L}^{-\frac{1}{2}} \boldsymbol{W}^{-1} \boldsymbol{L}^{-\frac{1}{2}} - \frac{4\gamma_{\boldsymbol{W}} \cdot \lambda_{\max} (\boldsymbol{W}) \cdot \omega_{\boldsymbol{W}} (4\omega_{\boldsymbol{W}} + 1)}{n} \cdot \lambda_{\min} (\boldsymbol{L}) \cdot \boldsymbol{I}_d \succeq \boldsymbol{I}_d.$$

Taking the minimum eigenvalue of both sides, we obtain that,

$$\frac{1}{\gamma_{\boldsymbol{W}}} \cdot \lambda_{\min} \left( \boldsymbol{L}^{-\frac{1}{2}} \boldsymbol{W}^{-1} \boldsymbol{L}^{-\frac{1}{2}} \right) - \frac{4\gamma_{\boldsymbol{W}} \cdot \lambda_{\max} (\boldsymbol{W}) \cdot \omega_{\boldsymbol{W}} (4\omega_{\boldsymbol{W}} + 1)}{n} \cdot \lambda_{\min} (\boldsymbol{L}) \geq 1,$$

If we denote $C_{\boldsymbol{W}} := \frac{\lambda_{\max}(\boldsymbol{W}) \cdot \omega_{\boldsymbol{W}} (4\omega_{\boldsymbol{W}} + 1)}{n} > 0$, and $\lambda_{\boldsymbol{W}} := \lambda_{\max}^{-1} \left( \boldsymbol{L}^{\frac{1}{2}} \boldsymbol{W} \boldsymbol{L}^{\frac{1}{2}} \right)$, we can write

$$4 \cdot C_{\boldsymbol{W}} \cdot \lambda_{\min} (\boldsymbol{L}) \cdot \gamma_{\boldsymbol{W}}^2 + \gamma_{\boldsymbol{W}} - \lambda_{\boldsymbol{W}} \leq 0.$$

The solution is given by

$$\gamma_{\boldsymbol{W}} \leq \frac{2\lambda_{\boldsymbol{W}}}{1 + \sqrt{1 + 16 C_{\boldsymbol{W}} \lambda_{\min} (\boldsymbol{L}) \cdot \lambda_{\boldsymbol{W}}}}.$$

### E.2.2 PROOF OF COROLLARY 5

The best scaling factor in this case is given as, according to Corollary 2,

$$\gamma_{\boldsymbol{L}^{-1}} = \frac{2}{1 + \sqrt{1 + 16C_{\boldsymbol{L}^{-1}} \cdot \lambda_{\min}(\boldsymbol{L})}}.$$

In order to reach a $\varepsilon^2$ stationary point, we need

$$K \geq \frac{\det(\boldsymbol{L})^{\frac{1}{d}} \left( f(x^0) - f^\star \right)}{\varepsilon^2} \cdot \left( 1 + \sqrt{1 + 16C_{\boldsymbol{L}^{-1}} \cdot \lambda_{\min}(\boldsymbol{L})} \right).$$

### E.2.3 PROOF OF COROLLARY 6

The iteration complexity of det-DASHA is given by, according to, Corollary 5,

$$\mathcal{O}\left( \frac{f(x^0) - f^\star}{\epsilon^2} \cdot \left( 1 + \sqrt{1 + 16C_{\boldsymbol{L}^{-1}} \cdot \lambda_{\min}(\boldsymbol{L})} \right) \cdot \det(\boldsymbol{L})^{\frac{1}{d}} \right).$$

Using the inequality $\sqrt{1+t} \leq 1 + \sqrt{t}$ for $t > 0$, and leaving out the coefficients, we obtain

$$\mathcal{O}\left( \frac{f(x^0) - f^\star}{\epsilon^2} \cdot \left( 1 + \sqrt{C_{\boldsymbol{L}^{-1}} \cdot \lambda_{\min}(\boldsymbol{L})} \right) \cdot \det(\boldsymbol{L})^{\frac{1}{d}} \right).$$

Notice that

$$C_{\boldsymbol{L}^{-1}} \cdot \lambda_{\min}(\boldsymbol{L}) = \lambda_{\max}\left( \boldsymbol{L}^{-1} \right) \cdot \frac{\omega_{\boldsymbol{L}^{-1}}\left( 4\omega_{\boldsymbol{L}^{-1}} + 1 \right)}{n} \cdot \lambda_{\min}(\boldsymbol{L}) = \frac{\omega_{\boldsymbol{L}^{-1}}\left( 4\omega_{\boldsymbol{L}^{-1}+1} \right)}{n}.$$

As a result, the iteration complexity can be further simplified as

$$\mathcal{O}\left( \frac{f(x^0) - f^*}{\epsilon^2} \cdot \left( 1 + \frac{\omega_{\boldsymbol{L}^{-1}}}{\sqrt{n}} \right) \cdot \det(\boldsymbol{L})^{\frac{1}{d}} \right).$$

The iteration complexity of DASHA is, according to Tyurin & Richtárik (2024, Corollary 6.2)

$$\mathcal{O}\left( \frac{1}{\epsilon^2} \cdot \left( f(x^0) - f^\star \right) \left( L + \frac{\omega}{\sqrt{n}} \widehat{L} \right) \right),$$

where $\widehat{L} = \sqrt{\frac{1}{n} \sum_{i=1}^{n} L_i^2}$. Since $\det(\boldsymbol{L})^{\frac{1}{d}} \leq \lambda_{\max}(\boldsymbol{L}) = L$, and $L \leq \widehat{L}$, it is easy to see that compared to DASHA, det-DASHA has a better iteration complexity when the momentum is the same. Notice that those two algorithms use the same sketch, thus, it also indicates that the communication complexity of the two algorithms are the same.

### E.2.4 PROOF OF COROLLARY 7

The iteration complexity of det-MARINA is given by

$$\mathcal{O}\left( \frac{f(x^0) - f^\star}{\epsilon^2} \cdot \det(\boldsymbol{L})^{\frac{1}{d}} \cdot \left( 1 + \sqrt{\alpha\beta\Lambda_{\boldsymbol{L}^{-1},\mathcal{S}}} \right) \right),$$

after removing logarithmic factors. Plugging in the definitions we obtain in the case of $\omega_{\boldsymbol{L}^{-1}} + 1 = \frac{1}{p}$, we have

$$\mathcal{O}\left( \frac{f(x^0) - f^\star}{\epsilon^2} \cdot \det(\boldsymbol{L})^{\frac{1}{d}} \cdot \left( 1 + \frac{\omega_{\boldsymbol{L}^{-1}}}{n} \right) \right).$$

From the proof of Corollary 6, we know that the iteration complexity of det-DASHA is

$$\mathcal{O}\left( \frac{1}{\epsilon^2} \cdot \left( f(x^0) - f^\star \right) \left( L + \frac{\omega}{\sqrt{n}} \widehat{L} \right) \right).$$

It is easy to see that in this case the two algorithms have the same iteration complexity asymptotically. Notice that the communication complexity is the product of bytes sent per iteration and the number of iterations. det-DASHA clearly sends less bytes per iteration because it always sent the compressed gradient differences, which means that it has a better communication complexity than det-MARINA.

## F Distributed DET-CGD

This section is a brief summary of the distributed det-CGD algorithm and its theoretical analysis. The details can be found in (Li et al., 2024b). The algorithm follows the standard FL paradigm. See the pseudocode in Algorithm 3.

---

**Algorithm 3** Distributed det-CGD

---

1: **Input:** Starting point $x^0$, stepsize matrix $\boldsymbol{D}$, number of iterations $K$
2: **for** $k = 0, 1, 2, \ldots, K-1$ **do**
3:    The devices in parallel:
4:    sample $\boldsymbol{S}_i^k \sim \mathcal{S}$;
5:    compute $\boldsymbol{S}_i^k \nabla f_i(x^k)$;
6:    broadcast $\boldsymbol{S}_i^k \nabla f_i(x^k)$.
7:    The server:
8:    combines $g^k = \frac{1}{n} \sum_{i=1}^n \boldsymbol{S}_i^k \nabla f_i(x^k)$;
9:    computes $x^{k+1} = x^k - \boldsymbol{D} g^k$;
10:   broadcasts $x^{k+1}$.
11: **end for**
12: **Return:** $x^K$

---

Below is the main convergence result for the algorithm.

**Theorem 3.** *Suppose that $f$ is $\boldsymbol{L}$-smooth. Under the Assumptions 1,2, if the stepsize satisfies*

$$\boldsymbol{DLD} \preceq \boldsymbol{D}, \tag{51}$$

*then the following convergence bound is true for the iteration of Algorithm 3:*

$$\min_{0 \leq k \leq K-1} \mathbb{E}\left[\left\|\nabla f(x^k)\right\|^2_{\frac{\boldsymbol{D}}{\det(\boldsymbol{D})^{1/d}}}\right] \leq \frac{2(1 + \frac{\lambda_{\boldsymbol{D}}}{n})^K \left(f(x^0) - f^\star\right)}{\det(\boldsymbol{D})^{1/d} K} + \frac{2\lambda_{\boldsymbol{D}} \Delta^\star}{\det(\boldsymbol{D})^{1/d} n}, \tag{52}$$

*where $\Delta^\star := f^\star - \frac{1}{n} \sum_{i=1}^n f_i^\star$ and*

$$\lambda_{\boldsymbol{D}} := \max_i \left\{ \lambda_{\max}\left(\mathbb{E}\left[\boldsymbol{L}_i^{\frac{1}{2}}\left(\boldsymbol{S}_i^k - \boldsymbol{I}_d\right)\boldsymbol{DLD}\left(\boldsymbol{S}_i^k - \boldsymbol{I}_d\right)\boldsymbol{L}_i^{\frac{1}{2}}\right]\right) \right\}.$$

**Remark 8.** *On the right hand side of* (52) *we observe that increasing $K$ will only reduce the first term, that corresponds to the convergence error. Whereas, the second term, which does not depend on $K$, will remain constant, if the other parameters of the algorithm are fixed. This testifies to the neighborhood phenomenon which we discussed in Section 2.*

**Remark 9.** *If the stepsize satisfies the below conditions,*

$$\boldsymbol{DLD} \preceq \boldsymbol{D}, \quad \lambda_{\boldsymbol{D}} \leq \min\left\{\frac{n}{K}, \frac{n\varepsilon^2}{4\Delta^\star}\det(\boldsymbol{D})^{1/d}\right\}, \quad K \geq \frac{12(f(x^0) - f^\star)}{\det(\boldsymbol{D})^{1/d}\varepsilon^2}, \tag{53}$$

*then we obtain $\varepsilon$-stationary point.*

One can see that in the convergence guarantee of det-CGD in the distributed case, the result (52) is not variance-reduced. Because of this limitation, in order to reach a $\varepsilon$ stationary point, the stepsize condition in (53) is restrictive.

## G Extension of DET-CGD2 in MARINA Form

In this section we want to extend det-CGD2 into its variance reduced counterpart in MARINA form.

### G.1 Extension of DET-CGD2 to its variance reduced counterpart

We call det-MARINA as the extension of det-CGD1, and Algorithm 4 as the extension of det-CGD2 due to the difference in the order of applying sketches and stepsize matrices. The key difference

---

**Algorithm 4** det-CGD2-VR

---

1: **Input:** starting point $x^0$, stepsize matrix $\boldsymbol{D}$, probability $p \in (0, 1]$, number of iterations $K$
2: Initialize $g^0 = \boldsymbol{D} \cdot \nabla f(x^0)$
3: **for** $k = 0, 1, \ldots, K - 1$ **do**
4:     Sample $c_k \sim \mathrm{Be}(p)$
5:     Broadcast $g^k$ to all workers
6:     **for** $i = 1, 2, \ldots$ in parallel **do**
7:         $x^{k+1} = x^k - g^k$
8:         Set $g_i^{k+1} = \begin{cases} \boldsymbol{D} \cdot \nabla f_i(x^{k+1}) & \text{if } c_k = 1 \\ g^k + \boldsymbol{T}_i^k \boldsymbol{D} \left( \nabla f_i(x^{k+1}) - \nabla f_i(x^k) \right) & \text{if } c_k = 0 \end{cases}$
9:     **end for**
10:     $g^{k+1} = \frac{1}{n} \sum_{i=1}^n g_i^{k+1}$
11: **end for**
12: **Return:** $\tilde{x}^K$ chosen uniformly at random from $\{x^k\}_{k=0}^{K-1}$

---

between det-CGD1 and det-CGD2 is that in det-CGD1 the gradient is sketched first and then multiplied by the stepsize, while for det-CGD2, the gradient is multiplied by the stepsize first after which the product is sketched. The convergence for Algorithm 4 can be proved in a similar manner as Theorem 1.

**Theorem 4.** *Let Assumptions 1 and 2 hold, with the gradient of $f$ being $\boldsymbol{L}$-Lipschitz. If the stepsize matrix $\boldsymbol{D} \in \mathbb{S}_{++}^d$ satisfies*

$$\boldsymbol{D}^{-1} \succeq \left( \frac{(1-p) \cdot R'(\boldsymbol{D}, \mathcal{S})}{np} + 1 \right) \boldsymbol{L},$$

*where*

$$R'(\boldsymbol{D}, \mathcal{S}) = \frac{1}{n} \sum_{i=1}^n \lambda_{\max} \left( \boldsymbol{D} \mathbb{E} \left[ \boldsymbol{T}_i^k \boldsymbol{D}^{-1} \boldsymbol{T}_i^k \right] \boldsymbol{D} \boldsymbol{L}_i^{\frac{1}{2}} - \boldsymbol{L}_i^{\frac{1}{2}} \boldsymbol{D} \right) \cdot \lambda_{\max} \left( \boldsymbol{L}_i \right) \cdot \lambda_{\max} \left( \boldsymbol{L}^{-\frac{1}{2}} \boldsymbol{L}_i \boldsymbol{L}^{-\frac{1}{2}} \right).$$

*Then after $K$ iterations of Algorithm 4, we have*

$$\mathbb{E} \left[ \left\| \nabla f(\tilde{x}^K) \right\|_{\frac{\boldsymbol{D}}{\det(\boldsymbol{D})^{1/d}}}^2 \right] \leq \frac{2 \left( f(x^0) - f^\star \right)}{\det(\boldsymbol{D})^{1/d} \cdot K}.$$

*This is to say that in order to reach a $\varepsilon$-stationary point, we require*

$$K \geq \frac{2(f(x^0) - f^\star)}{\det(\boldsymbol{D})^{1/d} \cdot \varepsilon^2}.$$

If we look at the scalar case where $\boldsymbol{D} = \gamma \cdot \boldsymbol{I}_d$, $\boldsymbol{L}_i = L_i \cdot \boldsymbol{I}_d$ and $\boldsymbol{L} = L \cdot \boldsymbol{I}_d$, then the condition in Theorem 4 reduces to

$$\frac{(1-p)\omega L^2}{np} + L - \frac{1}{\gamma} \leq 0. \tag{54}$$

Notice that here $\omega = \lambda_{\max} \left( \mathbb{E} \left[ \left( \boldsymbol{T}_i^k \right)^2 \right] \right) - 1$, and we have $L^2 = \frac{1}{n} \sum_{i=1}^n L_i^2$, which is due to the relation given in Proposition 5. This condition coincides with the condition for convergence of MARINA. One may also check that, the update rule in Algorithm 4, is the same as MARINA in the scalar case. However, the condition given in Theorem 4 is not simpler than Theorem 1, contrary to the single-node case. We emphasize that Algorithm 4 is not suitable for the federated learning setting where the clients have limited resources. In order to perform the update, each client is required to store the stepsize matrix $\boldsymbol{D}$ which is of size $d \times d$. In the over-parameterized regime, the dataset size is $m \times d$ where $m$ is the number of data samples, and we have $d > m$. This means that the stepsize matrix each client needs to store is even larger than the dataset itself, which is unacceptable given the limited resources each client has.

### G.2 ANALYSIS OF ALGORITHM 4

We first present two lemmas which are necessary for the proofs of Theorem 4.

**Lemma 6.** *Assume that function $f$ is $\boldsymbol{L}$-smooth, and $x^{k+1} = x^k - g^k$, and matrix $\boldsymbol{D} \in \mathbb{S}_{++}^d$. Then we will have*

$$f(x^{k+1}) \leq f(x^k) - \frac{1}{2}\left\|\nabla f(x^k)\right\|_{\boldsymbol{D}}^2 + \frac{1}{2}\left\|\boldsymbol{D}\cdot\nabla f(x^k) - g^k\right\|_{\boldsymbol{D}^{-1}}^2 - \frac{1}{2}\left\|x^{k+1} - x^k\right\|_{\boldsymbol{D}^{-1}-\boldsymbol{L}}^2. \quad (55)$$

This lemma is formulated in a different way from Lemma 1 on purpose.

**Lemma 7.** *For any sketch matrix $\boldsymbol{T} \in \mathbb{S}_+^d$, vector $t \in \mathbb{R}^d$, matrix $\boldsymbol{D} \in \mathbb{S}_{++}^d$ and matrix $\boldsymbol{L} \in \mathbb{S}_{++}^d$, we have*

$$\mathbb{E}\left[\left\|\boldsymbol{T}\boldsymbol{D}t - \boldsymbol{D}t\right\|_{\boldsymbol{D}^{-1}}^2\right] \leq \lambda_{\max}\left(\boldsymbol{L}^{\frac{1}{2}}\boldsymbol{D}\mathbb{E}\left[\boldsymbol{T}\boldsymbol{D}^{-1}\boldsymbol{T}\right]\boldsymbol{D}\boldsymbol{L}^{\frac{1}{2}} - \boldsymbol{L}^{\frac{1}{2}}\boldsymbol{D}\boldsymbol{L}^{\frac{1}{2}}\right)\|t\|_{\boldsymbol{L}^{-1}}^2. \quad (56)$$

### G.3 PROOF OF THEOREM 4

We start with Lemma 6,

$$\mathbb{E}\left[f(x^{k+1})\right] \leq \mathbb{E}\left[f(x^k)\right] - \mathbb{E}\left[\frac{1}{2}\left\|\nabla f(x^k)\right\|_{\boldsymbol{D}}^2\right]$$
$$+ \mathbb{E}\left[\frac{1}{2}\left\|\boldsymbol{D}\cdot\nabla f(x^k) - g^k\right\|_{\boldsymbol{D}^{-1}}^2\right] - \mathbb{E}\left[\frac{1}{2}\left\|x^{k+1} - x^k\right\|_{\boldsymbol{D}^{-1}-\boldsymbol{L}}^2\right]. \quad (57)$$

Now we do the same as Theorem 1 and look at the term $\mathbb{E}\left[\left\|\boldsymbol{D}\cdot\nabla f(x^{k+1}) - g^{k+1}\right\|_{\boldsymbol{D}^{-1}}^2\right]$. Recall that $g^k$ here is given by

$$g^{k+1} = \begin{cases} \boldsymbol{D}\cdot\nabla f(x^{k+1}) & \text{with probability } p \\ g^k + \frac{1}{n}\sum_{i=1}^n \boldsymbol{T}_i^k\boldsymbol{D}\left(\nabla f_i(x^{k+1}) - \nabla f_i(x^k)\right) & \text{with probability } 1-p. \end{cases}$$

As a result, we have

$$\mathbb{E}\left[\left\|g^{k+1} - \boldsymbol{D}\nabla f(x^{k+1})\right\|_{\boldsymbol{D}^{-1}}^2 \mid x^{k+1}, x^k\right]$$

$$= \mathbb{E}\left[\mathbb{E}\left[\left\|g^{k+1} - \boldsymbol{D}\nabla f(x^{k+1})\right\|_{\boldsymbol{D}^{-1}}^2 \mid x^{k+1}, x^k, c_k\right]\right]$$

$$= p\cdot\left\|\boldsymbol{D}\nabla f(x^{k+1}) - \boldsymbol{D}\nabla f(x^{k+1})\right\|_{\boldsymbol{D}^{-1}}^2$$

$$+ (1-p)\cdot\mathbb{E}\left[\left\|g^k + \frac{1}{n}\sum_{i=1}^n \boldsymbol{T}_i^k\boldsymbol{D}\left(\nabla f_i(x^{k+1}) - \nabla f_i(x^k)\right) - \boldsymbol{D}\nabla f(x^{k+1})\right\|_{\boldsymbol{D}^{-1}}^2 \mid x^{k+1}, x^k\right]$$

$$= (1-p)\cdot\mathbb{E}\left[\left\|g^k + \frac{1}{n}\sum_{i=1}^n \boldsymbol{T}_i^k\boldsymbol{D}\left(\nabla f_i(x^{k+1}) - \nabla f_i(x^k)\right) - \boldsymbol{D}\nabla f(x^{k+1})\right\|_{\boldsymbol{D}^{-1}}^2 \mid x^{k+1}, x^k\right].$$

For the sake of presentation, we use $\mathrm{E}_k[\cdot]$ to denote the conditional expectation $\mathrm{E}\left[\cdot \mid x_k, x_{k+1}\right]$ on $x_k, x_{k+1}$. Using Fact 2 with $x = \frac{1}{n}\sum_{i=1}^n \boldsymbol{T}_i^k\boldsymbol{D}\left(\nabla f_i(x^{k+1}) - \nabla f_i(x^k)\right)$, $c = \boldsymbol{D}\nabla f(x^{k+1}) - g^k$, we are able to obtain that,

$$(1-p)\mathrm{E}_k\left[\left\|g^k + \frac{1}{n}\sum_{i=1}^n \boldsymbol{T}_i^k\boldsymbol{D}\left(\nabla f_i(x^{k+1}) - \nabla f_i(x^k)\right) - \boldsymbol{D}\nabla f(x^{k+1})\right\|_{\boldsymbol{D}^{-1}}^2\right]$$

$$= (1-p)\mathrm{E}_k\left[\left\|\frac{1}{n}\sum_{i=1}^n \boldsymbol{T}_i^k\boldsymbol{D}\left(\nabla f_i(x^{k+1}) - \nabla f_i(x^k)\right) - \boldsymbol{D}\left(\nabla f(x^{k+1}) - \nabla f(x^k)\right)\right\|_{\boldsymbol{D}^{-1}}^2\right]$$

$$+ (1-p)\left\|g^k - \nabla f(x^k)\right\|_{\boldsymbol{D}^{-1}}^2$$

$$= (1-p)\mathrm{E}_k\left[\left\|\frac{1}{n}\sum_{i=1}^n \left[\boldsymbol{T}_i^k\boldsymbol{D}\left(\nabla f_i(x^{k+1}) - \nabla f_i(x^k)\right) - \boldsymbol{D}\left(\nabla f_i(x^{k+1}) - \nabla f_i(x^k)\right)\right]\right\|_{\boldsymbol{D}^{-1}}^2\right]$$

$$+ (1-p)\left\|g^k - \nabla f(x^k)\right\|_{\boldsymbol{D}^{-1}}^2.$$

It is not hard to notice that for the sketch matrices we pick, the following identity holds due to the unbiasedness,

$$\mathrm{E}_k \left[ \boldsymbol{T}_i^k \boldsymbol{D}(\nabla f_i(x^{k+1}) - \nabla f_i(x^k)) \right] = \boldsymbol{D}(\nabla f_i(x^{k+1}) - \nabla f_i(x^k)),$$

and any two random vectors in the set $\left\{ \boldsymbol{T}_i^k \boldsymbol{D}(\nabla f_i(x^{k+1}) - \nabla f_i(x^k)) \right\}_{i=1}^n$ are independent if $x^{k+1}, x^k$ are fixed. As a result

$$\mathrm{E}_k \left[ \left\| g^{k+1} - \boldsymbol{D}\nabla f(x^{k+1}) \right\|_{\boldsymbol{D}^{-1}}^2 \right]$$

$$= \frac{1-p}{n^2} \sum_{i=1}^n \mathrm{E}_k \left[ \left\| \boldsymbol{T}_i^k \left( \boldsymbol{D}\nabla f_i(x^{k+1}) - \boldsymbol{D}\nabla f_i(x^k) \right) - \left( \boldsymbol{D}\nabla f_i(x^{k+1}) - \boldsymbol{D}\nabla f_i(x^k) \right) \right\|_{\boldsymbol{D}^{-1}}^2 \right]$$

$$+ (1-p) \cdot \left\| g^k - \boldsymbol{D}\nabla f(x^k) \right\|_{\boldsymbol{D}^{-1}}^2. \tag{58}$$

For each term within the summation, we can further upper bound it using Lemma 7

$$\mathrm{E}_k \left[ \left\| \boldsymbol{T}_i^k \left( \boldsymbol{D}\nabla f_i(x^{k+1}) - \boldsymbol{D}\nabla f_i(x^k) \right) - \left( \boldsymbol{D}\nabla f_i(x^{k+1}) - \boldsymbol{D}\nabla f_i(x^k) \right) \right\|_{\boldsymbol{D}^{-1}}^2 \right]$$

$$\leq \lambda_{\max} \left( \boldsymbol{L}_i^{\frac{1}{2}} \boldsymbol{D} \mathbb{E} \left[ \boldsymbol{T}_i^k \boldsymbol{D}^{-1} \boldsymbol{T}_i^k \right] \boldsymbol{D} \boldsymbol{L}_i^{\frac{1}{2}} - \boldsymbol{L}_i^{\frac{1}{2}} \boldsymbol{D} \boldsymbol{L}_i^{\frac{1}{2}} \right) \left\| \nabla f_i(x^{k+1}) - \nabla f_i(x^k) \right\|_{\boldsymbol{L}_i^{-1}}^2$$

$$\leq \lambda_{\max} \left( \boldsymbol{L}_i^{\frac{1}{2}} \boldsymbol{D} \mathbb{E} \left[ \boldsymbol{T}_i^k \boldsymbol{D}^{-1} \boldsymbol{T}_i^k \right] \boldsymbol{D} \boldsymbol{L}_i^{\frac{1}{2}} - \boldsymbol{L}_i^{\frac{1}{2}} \boldsymbol{D} \boldsymbol{L}_i^{\frac{1}{2}} \right) \left\| x^{k+1} - x^k \right\|_{\boldsymbol{L}_i}^2.$$

Where the last inequality is due to Assumption 2. Plugging back into (58), we get

$$\mathrm{E}_k \left[ \left\| g^{k+1} - \boldsymbol{D}\nabla f(x^{k+1}) \right\|_{\boldsymbol{D}^{-1}}^2 \right]$$

$$\leq \frac{1-p}{n^2} \sum_{i=1}^n \lambda_{\max} \left( \boldsymbol{L}_i^{\frac{1}{2}} \boldsymbol{D} \mathbb{E} \left[ \boldsymbol{T}_i^k \boldsymbol{D}^{-1} \boldsymbol{T}_i^k \right] \boldsymbol{D} \boldsymbol{L}_i^{\frac{1}{2}} - \boldsymbol{L}_i^{\frac{1}{2}} \boldsymbol{D} \boldsymbol{L}_i^{\frac{1}{2}} \right) \left\| x^{k+1} - x^k \right\|_{\boldsymbol{L}_i}^2$$

$$+ (1-p) \cdot \left\| g^k - \boldsymbol{D}\nabla f(x^k) \right\|_{\boldsymbol{D}^{-1}}^2.$$

Applying the replacement trick form the proof of Theorem 1, we obtain

$$\mathrm{E}_k \left[ \left\| g^{k+1} - \boldsymbol{D}\nabla f(x^{k+1}) \right\|_{\boldsymbol{D}^{-1}}^2 \right]$$

$$\leq \frac{1-p}{n^2} \sum_{i=1}^n \lambda_{\max} \left( \boldsymbol{L}_i^{\frac{1}{2}} \boldsymbol{D} \mathbb{E} \left[ \boldsymbol{T}_i^k \boldsymbol{D}^{-1} \boldsymbol{T}_i^k \right] \boldsymbol{D} \boldsymbol{L}_i^{\frac{1}{2}} - \boldsymbol{L}_i^{\frac{1}{2}} \boldsymbol{D} \boldsymbol{L}_i^{\frac{1}{2}} \right)$$

$$\times \left\langle \boldsymbol{L}^{\frac{1}{2}} \left( x^{k+1} - x^k \right), \left( \boldsymbol{L}^{-\frac{1}{2}} \boldsymbol{L}_i \boldsymbol{L}^{-\frac{1}{2}} \right) \cdot \boldsymbol{L}^{\frac{1}{2}} \left( x^{k+1} - x^k \right) \right\rangle + (1-p) \cdot \left\| g^k - \boldsymbol{D}\nabla f(x^k) \right\|_{\boldsymbol{D}^{-1}}^2$$

$$\leq \frac{1-p}{n^2} \sum_{i=1}^n \lambda_{\max} \left( \boldsymbol{L}_i^{\frac{1}{2}} \left( \boldsymbol{D} \mathbb{E} \left[ \boldsymbol{T}_i^k \boldsymbol{D}^{-1} \boldsymbol{T}_i^k \right] \boldsymbol{D} - \boldsymbol{D} \right) \boldsymbol{L}_i^{\frac{1}{2}} \right) \cdot \lambda_{\max} \left( \boldsymbol{L}^{-\frac{1}{2}} \boldsymbol{L}_i \boldsymbol{L}^{-\frac{1}{2}} \right) \left\| x^{k+1} - x^k \right\|_{\boldsymbol{L}}^2$$

$$+ (1-p) \cdot \left\| g^k - \boldsymbol{D}\nabla f(x^k) \right\|_{\boldsymbol{D}^{-1}}^2.$$

Applying Fact 5, we obtain

$$\mathrm{E}_k \left[ \left\| g^{k+1} - \boldsymbol{D}\nabla f(x^{k+1}) \right\|_{\boldsymbol{D}^{-1}}^2 \right]$$

$$\leq \frac{1-p}{n^2} \sum_{i=1}^n \lambda_{\max} \left( \boldsymbol{D} \mathbb{E} \left[ \boldsymbol{T}_i^k \boldsymbol{D}^{-1} \boldsymbol{T}_i^k \right] \boldsymbol{D} - \boldsymbol{D} \right) \lambda_{\max} (\boldsymbol{L}_i) \lambda_{\max} \left( \boldsymbol{L}^{-\frac{1}{2}} \boldsymbol{L}_i \boldsymbol{L}^{-\frac{1}{2}} \right) \left\| x^{k+1} - x^k \right\|_{\boldsymbol{L}}^2$$

$$+ (1-p) \cdot \left\| g^k - \boldsymbol{D}\nabla f(x^k) \right\|_{\boldsymbol{D}^{-1}}^2.$$

Recalling the definition of $R'(\boldsymbol{D}, \mathcal{S})$, we further simplify it to

$$\mathrm{E}_k \left[ \left\| g^{k+1} - \boldsymbol{D}\nabla f(x^{k+1}) \right\|_{\boldsymbol{D}^{-1}}^2 \right]$$

$$\leq \frac{(1-p) \cdot R'(\boldsymbol{D}, \mathcal{S})}{n} \left\| x^{k+1} - x^k \right\|_{\boldsymbol{L}}^2 + (1-p) \cdot \left\| g^k - \boldsymbol{D}\nabla f(x^k) \right\|_{\boldsymbol{D}^{-1}}^2.$$

Taking expectation again and using the tower property, we get

$$\mathbb{E}\left[\left\|g^{k+1} - \boldsymbol{D}\nabla f(x^{k+1})\right\|_{\boldsymbol{D}^{-1}}^2\right] \tag{59}$$

$$\leq \frac{(1-p)\cdot R'(\boldsymbol{D}, \mathcal{S})}{n}\mathbb{E}\left[\left\|x^{k+1} - x^k\right\|_{\boldsymbol{L}}^2\right] + (1-p)\cdot\mathbb{E}\left[\left\|g^k - \boldsymbol{D}\nabla f(x^k)\right\|_{\boldsymbol{D}^{-1}}^2\right]. \tag{60}$$

Construct the Lyapunov function $\Phi_k$ as follows,

$$\Phi_k = f(x^k) - f^\star + \frac{1}{2p}\left\|g^k - \boldsymbol{D}\nabla f(x^k)\right\|_{\boldsymbol{D}^{-1}}^2.$$

Utilizing (57) and (59), we are able to get

$$\mathbb{E}\left[\Phi_{k+1}\right] \leq \mathbb{E}\left[f(x^k) - f^\star\right] - \frac{1}{2}\mathbb{E}\left[\left\|\nabla f(x^k)\right\|_{\boldsymbol{D}}^2\right]$$

$$+ \frac{1}{2}\mathbb{E}\left[\left\|g^k - \boldsymbol{D}\nabla f(x^k)\right\|_{\boldsymbol{D}^{-1}}^2\right] - \frac{1}{2}\mathbb{E}\left[\left\|x^{k+1} - x^k\right\|_{\boldsymbol{D}^{-1}-\boldsymbol{L}}\right]$$

$$+ \frac{1}{2p}\cdot\frac{(1-p)R'(\boldsymbol{D}, \mathcal{S})}{n}\mathbb{E}\left[\left\|x^{k+1} - x^k\right\|_{\boldsymbol{L}}^2\right] + \frac{1-p}{2p}\mathbb{E}\left[\left\|g^k - \boldsymbol{D}\nabla f(x^k)\right\|_{\boldsymbol{D}^{-1}}^2\right]$$

$$= \mathbb{E}\left[\Phi_k\right] - \frac{1}{2}\mathbb{E}\left[\left\|\nabla f(x^k)\right\|_{\boldsymbol{D}}^2\right]$$

$$+ \frac{1}{2}\left(\frac{(1-p)R'(\boldsymbol{D}, \mathcal{S})}{np}\mathbb{E}\left[\left\|x^{k+1} - x^k\right\|_{\boldsymbol{L}}^2\right] - \mathbb{E}\left[\left\|x^{k+1} - x^k\right\|_{\boldsymbol{D}^{-1}-\boldsymbol{L}}^2\right]\right).$$

Now, notice that the last term in the above inequality is non-positive as guaranteed by the condition

$$\boldsymbol{D}^{-1} \succeq \left(\frac{(1-p)R'(\boldsymbol{D}, \mathcal{S})}{np} + 1\right)\boldsymbol{L}.$$

This leads to the recurrence after ignoring the last term,

$$\mathbb{E}\left[\Phi_{k+1}\right] \leq \mathbb{E}\left[\Phi_k\right] - \frac{1}{2}\mathbb{E}\left[\left\|\nabla f(x^k)\right\|_{\boldsymbol{D}}^2\right].$$

Unrolling this recurrence, we get

$$\frac{1}{K}\sum_{k=0}^{K-1}\mathbb{E}\left[\left\|\nabla f(x^k)\right\|_{\boldsymbol{D}}^2\right] \leq \frac{2\left(\mathbb{E}\left[\Phi_0\right] - \mathbb{E}\left[\Phi_K\right]\right)}{K}.$$

The left hand side can viewed as average over $\tilde{x}^K$, which is drawn uniformly at random from $\{x_k\}_{k=0}^{K-1}$, while the right hand side can be simplified as

$$\frac{2\left(\mathbb{E}\left[\Phi_0\right] - \mathbb{E}\left[\Phi_K\right]\right)}{K} \leq \frac{2\Phi_0}{K} = \frac{2\left(f(x^0) - f^\star + \frac{1}{2p}\left\|g^0 - \nabla f(x^0)\right\|_{\boldsymbol{D}}^2\right)}{K}.$$

Recalling that $g^0 = \nabla f(x^0)$ and performing determinant normalization as Li et al. (2024b), we get

$$\mathbb{E}\left[\left\|\nabla f(\tilde{x}^K)\right\|_{\frac{\boldsymbol{D}}{\det(\boldsymbol{D})^{1/d}}}^2\right] \leq \frac{2\left(f(x^0) - f^\star\right)}{\det(\boldsymbol{D})^{1/d}K}.$$

# H  PROOFS OF THE TECHNICAL LEMMAS

## H.1  PROOF OF LEMMA 1

Let $\bar{x}^{k+1} := x^k - \boldsymbol{D}\cdot\nabla f(x^k)$. Since $f$ has a matrix $\boldsymbol{L}$-Lipschitz gradient, $f$ is also $\boldsymbol{L}$-smooth. From the $\boldsymbol{L}$-smoothness of $f$, we have

$$\begin{aligned}
f(x^{k+1}) &\leq f(x^k) + \left\langle\nabla f(x^k), x^{k+1} - x^k\right\rangle + \frac{1}{2}\left\langle x^{k+1} - x^k, \boldsymbol{L}(x^{k+1} - x^k)\right\rangle \\
&= f(x^k) + \left\langle\nabla f(x^k) - g^k, x^{k+1} - x^k\right\rangle + \left\langle g^k, x^{k+1} - x^k\right\rangle + \frac{1}{2}\left\langle x^{k+1} - x^k, \boldsymbol{L}(x^{k+1} - x^k)\right\rangle.
\end{aligned}$$

We can merge the last two terms and obtain,

$$
\begin{aligned}
f(x^{k+1}) &\leq f(x^k) + \left\langle \nabla f(x^k) - g^k, -\boldsymbol{D} \cdot g^k \right\rangle - \left\langle x^{k+1} - x^k, \boldsymbol{D}^{-1}(x^{k+1} - x^k) \right\rangle \\
&\quad + \frac{1}{2} \left\langle x^{k+1} - x^k, \boldsymbol{L}(x^{k+1} - x^k) \right\rangle \\
&= f(x^k) + \left\langle \nabla f(x^k) - g^k, -\boldsymbol{D} \cdot g^k \right\rangle - \left\langle x^{k+1} - x^k, \left( \boldsymbol{D}^{-1} - \frac{1}{2}\boldsymbol{L} \right)(x^{k+1} - x^k) \right\rangle.
\end{aligned}
$$

We add and subtract $\left\langle \nabla f(x^k) - g^k, \boldsymbol{D} \cdot g^k \right\rangle$,

$$
\begin{aligned}
f(x^{k+1}) &\leq f(x^k) + \left\langle \nabla f(x^k) - g^k, \boldsymbol{D} \left( \nabla f(x^k) - g^k \right) \right\rangle - \left\langle \nabla f(x^k) - g^k, \boldsymbol{D} \cdot \nabla f(x^k) \right\rangle \\
&\quad - \left\langle x^{k+1} - x^k, \left( \boldsymbol{D}^{-1} - \frac{1}{2}\boldsymbol{L} \right)(x^{k+1} - x^k) \right\rangle \\
&= f(x^k) + \left\| \nabla f(x^k) - g^k \right\|_{\boldsymbol{D}}^2 - \left\langle x^{k+1} - \bar{x}^{k+1}, \boldsymbol{D}^{-1} \left( x^k - \bar{x}^{k+1} \right) \right\rangle \\
&\quad - \left\langle x^{k+1} - x^k, \left( \boldsymbol{D}^{-1} - \frac{1}{2}\boldsymbol{L} \right)(x^{k+1} - x^k) \right\rangle.
\end{aligned}
$$

Decomposing the term $\left\langle x^{k+1} - \bar{x}^{k+1}, \boldsymbol{D}^{-1} \left( x^k - \bar{x}^{k+1} \right) \right\rangle$, we get

$$
\begin{aligned}
f(x^{k+1}) &\leq f(x^k) + \left\| \nabla f(x^k) - g^k \right\|_{\boldsymbol{D}}^2 - \left\langle x^{k+1} - x^k, \left( \boldsymbol{D}^{-1} - \frac{1}{2}\boldsymbol{L} \right)(x^{k+1} - x^k) \right\rangle \\
&\quad - \frac{1}{2} \left( \left\| x^{k+1} - \bar{x}^{k+1} \right\|_{\boldsymbol{D}^{-1}}^2 + \left\| x^k - \bar{x}^{k+1} \right\|_{\boldsymbol{D}^{-1}}^2 - \left\| x^{k+1} - x^k \right\|_{\boldsymbol{D}^{-1}}^2 \right).
\end{aligned}
$$

Plugging in the definition of $x^{k+1}, \bar{x}^{k+1}$, we get

$$
\begin{aligned}
f(x^{k+1}) &\leq f(x^k) + \left\| \nabla f(x^k) - g^k \right\|_{\boldsymbol{D}}^2 - \left\| x^{k+1} - x^k \right\|_{\boldsymbol{D}^{-1} - \frac{1}{2}\boldsymbol{L}}^2 \\
&\quad - \frac{1}{2} \left( \left\| \boldsymbol{D}(\nabla f(x^k) - g^k) \right\|_{\boldsymbol{D}^{-1}}^2 + \left\| \boldsymbol{D} \cdot \nabla f(x^k) \right\|_{\boldsymbol{D}^{-1}}^2 - \left\| x^{k+1} - x^k \right\|_{\boldsymbol{D}^{-1}}^2 \right) \\
&= f(x^k) + \left\| \nabla f(x^k) - g^k \right\|_{\boldsymbol{D}}^2 - \left\| x^{k+1} - x^k \right\|_{\boldsymbol{D}^{-1} - \frac{1}{2}\boldsymbol{L}}^2 \\
&\quad - \frac{1}{2} \left( \left\| \nabla f(x^k) - g^k \right\|_{\boldsymbol{D}}^2 + \left\| \nabla f(x^k) \right\|_{\boldsymbol{D}}^2 - \left\| x^{k+1} - x^k \right\|_{\boldsymbol{D}^{-1}}^2 \right).
\end{aligned}
$$

Rearranging terms we get,

$$
\begin{aligned}
f(x^{k+1}) &\leq f(x^k) - \frac{1}{2} \left\| \nabla f(x^k) \right\|_{\boldsymbol{D}}^2 + \frac{1}{2} \left\| g^k - \nabla f(x^k) \right\|_{\boldsymbol{D}}^2 - \left\| x^{k+1} - x^k \right\|_{\boldsymbol{D}^{-1} - \frac{1}{2}\boldsymbol{L}}^2 + \frac{1}{2} \left\| x^{k+1} - x^k \right\|_{\boldsymbol{D}^{-1}}^2 \\
&= f(x^k) - \frac{1}{2} \left\| \nabla f(x^k) \right\|_{\boldsymbol{D}}^2 + \frac{1}{2} \left\| g^k - \nabla f(x^k) \right\|_{\boldsymbol{D}}^2 - \frac{1}{2} \left\| x^{k+1} - x^k \right\|_{\boldsymbol{D}^{-1} - \boldsymbol{L}}.
\end{aligned}
$$

## H.2 PROOF OF LEMMA 2

The definition of the weighted norm yields

$$
\begin{aligned}
\mathbb{E} \left[ \left\| \boldsymbol{S}t - t \right\|_{\boldsymbol{D}}^2 \right] &= \mathbb{E} \left[ \langle t, (\boldsymbol{S} - \boldsymbol{I}_d) \boldsymbol{D} (\boldsymbol{S} - \boldsymbol{I}_d) t \rangle \right] \\
&= \left\langle t, \mathbb{E} \left[ (\boldsymbol{S} - \boldsymbol{I}_d) \boldsymbol{D} (\boldsymbol{S} - \boldsymbol{I}_d) \right] t \right\rangle \\
&= \left\langle t, \boldsymbol{L}^{-\frac{1}{2}} \cdot \mathbb{E} \left[ \boldsymbol{L}^{\frac{1}{2}} (\boldsymbol{S} - \boldsymbol{I}_d) \boldsymbol{D} (\boldsymbol{S} - \boldsymbol{I}_d) \boldsymbol{L}^{\frac{1}{2}} \right] \cdot \boldsymbol{L}^{-\frac{1}{2}} t \right\rangle \\
&= \left\langle \boldsymbol{L}^{-\frac{1}{2}} t, \mathbb{E} \left[ \boldsymbol{L}^{\frac{1}{2}} (\boldsymbol{S} - \boldsymbol{I}_d) \boldsymbol{D} (\boldsymbol{S} - \boldsymbol{I}_d) \boldsymbol{L}^{\frac{1}{2}} \right] \cdot \boldsymbol{L}^{-\frac{1}{2}} t \right\rangle \\
&\leq \lambda_{\max} \left( \mathbb{E} \left[ \boldsymbol{L}^{\frac{1}{2}} (\boldsymbol{S} - \boldsymbol{I}_d) \boldsymbol{D} (\boldsymbol{S} - \boldsymbol{I}_d) \boldsymbol{L}^{\frac{1}{2}} \right] \right) \left\| \boldsymbol{L}^{-\frac{1}{2}} t \right\|^2 \\
&= \lambda_{\max} \left( \boldsymbol{L}^{\frac{1}{2}} (\mathbb{E} [\boldsymbol{S}\boldsymbol{D}\boldsymbol{S}] - \boldsymbol{D}) \boldsymbol{L}^{\frac{1}{2}} \right) \cdot \left\| t \right\|_{\boldsymbol{L}^{-1}}^2.
\end{aligned}
$$

## H.3 PROOF OF LEMMA 4

Throughout the following proof, we denote $\mathbb{E}_{\mathcal{S}}[\cdot]$ as taking expectation with respect to the randomness contained within the sketch sampled from distribution $\mathcal{S}$. We estimate the term $\mathbb{E}_{\mathcal{S}}\left[\left\|g^{k+1} - h^{k+1}\right\|_{\boldsymbol{D}}^2\right]$ in order to construct the Lyapunov function. For $\mathbb{E}_{\mathcal{S}}\left[\left\|g^{k+1} - h^{k+1}\right\|_{\boldsymbol{D}}^2\right]$, we have

$$
\mathbb{E}_{\mathcal{S}}\left[\left\|g^{k+1} - h^{k+1}\right\|_{\boldsymbol{D}}^2\right] = \mathbb{E}_{\mathcal{S}}\left[\left\|g^k + \frac{1}{n}\sum_{i=1}^n m_i^{k+1} - h^{k+1}\right\|_{\boldsymbol{D}}^2\right]
$$
$$
= \mathbb{E}_{\mathcal{S}}\left[\left\|g^k + \frac{1}{n}\sum_{i=1}^n \boldsymbol{S}_i^k\left(h_i^{k+1} - h_i^k - a(g_i^k - h_i^k)\right) - h^{k+1}\right\|_{\boldsymbol{D}}^2\right]
$$

Using Fact 3, we obtain

$$
\mathbb{E}_{\mathcal{S}}\left[\left\|g^{k+1} - h^{k+1}\right\|_{\boldsymbol{D}}^2\right]
$$
$$
= \mathbb{E}_{\mathcal{S}}\left[\left\|\frac{1}{n}\sum_{i=1}^n \boldsymbol{S}_i^k\left(h_i^{k+1} - h_i^k - a(g_i^k - h_i^k)\right) - \left(h^{k+1} - h^k - a(g^k - h^k)\right)\right\|_{\boldsymbol{D}}^2\right]
$$
$$
+ (1-a)^2\left\|h^k - g^k\right\|_{\boldsymbol{D}}^2
$$
$$
= \mathbb{E}_{\mathcal{S}}\left[\left\|\frac{1}{n}\sum_{i=1}^n \boldsymbol{S}_i^k\left(h_i^{k+1} - h_i^k - a(g_i^k - h_i^k)\right) - \frac{1}{n}\sum_{i=1}^n\left(h_i^{k+1} - h_i^k - a(g_i^k - h_i^k)\right)\right\|_{\boldsymbol{D}}^2\right]
$$
$$
+ (1-a)^2\left\|h^k - g^k\right\|_{\boldsymbol{D}}^2
$$
$$
= \frac{1}{n^2}\sum_{i=1}^n \mathbb{E}_{\mathcal{S}}\left[\left\|\boldsymbol{S}_i^k\left(h_i^{k+1} - h_i^k - a(g_i^k - h_i^k)\right) - \left(h_i^{k+1} - h_i^k - a(g_i^k - h_i^k)\right)\right\|_{\boldsymbol{D}}^2\right]
$$
$$
+ (1-a)^2\left\|h^k - g^k\right\|_{\boldsymbol{D}}^2.
$$

Here, the last identity is obtained from the unbiasedness of the sketches:

$$
\mathbb{E}_{\mathcal{S}}\left[\boldsymbol{S}_i^k\left(h_i^{k+1} - h_i^k - a(g_i^k - h_i^k)\right)\right] = h_i^{k+1} - h_i^k - a(g_i^k - h_i^k).
$$

We can further use Lemma 2, and obtain

$$
\mathbb{E}_{\mathcal{S}}\left[\left\|g^{k+1} - h^{k+1}\right\|_{\boldsymbol{D}}^2\right]
$$
$$
\leq \frac{1}{n^2}\sum_{i=1}^n \lambda_{\max}\left(\boldsymbol{D}^{-\frac{1}{2}}\left(\mathbb{E}\left[\boldsymbol{S}_i^k\boldsymbol{D}\boldsymbol{S}_i^k\right] - \boldsymbol{D}\right)\boldsymbol{D}^{-\frac{1}{2}}\right)\left\|h_i^{k+1} - h_i - a(g_i^k - h_i^k)\right\|_{\boldsymbol{D}}^2
$$
$$
+ (1-a)^2\left\|g^k - h^k\right\|_{\boldsymbol{D}}^2
$$
$$
\leq \frac{1}{n^2}\sum_{i=1}^n \lambda_{\max}\left(\boldsymbol{D}^{-1}\right)\cdot\lambda_{\max}\left(\mathbb{E}\left[\boldsymbol{S}_i^k\boldsymbol{D}\boldsymbol{S}_i^k\right] - \boldsymbol{D}\right)\left\|h_i^{k+1} - h_i^k - a(g_i^k - h_i^k)\right\|_{\boldsymbol{D}}^2
$$
$$
+ (1-a)^2\left\|g^k - h^k\right\|_{\boldsymbol{D}}^2.
$$

We can rewrite the above bound, after applying Jensen's inequality as

$$
\mathbb{E}_{\mathcal{S}}\left[\left\|g^{k+1} - h^{k+1}\right\|_{\boldsymbol{D}}^2\right] \leq \frac{2\Lambda_{\boldsymbol{D},\mathcal{S}}\cdot\lambda_{\max}\left(\boldsymbol{D}^{-1}\right)}{n^2}\sum_{i=1}^n \left\|h_i^{k+1} - h_i^k\right\|_{\boldsymbol{D}}^2
$$
$$
+ \frac{2a^2\Lambda_{\boldsymbol{D},\mathcal{S}}\cdot\lambda_{\max}\left(\boldsymbol{D}^{-1}\right)}{n^2}\sum_{i=1}^n \left\|g_i^k - h_i^k\right\|_{\boldsymbol{D}}^2
$$
$$
+ (1-a)^2\left\|g^k - h^k\right\|_{\boldsymbol{D}}^2.
$$

Notice that we have

$$\left\|h_i^{k+1} - h_i^k\right\|_{\boldsymbol{D}}^2 \le \lambda_{\max}\left(\boldsymbol{D}\right) \cdot \lambda_{\max}\left(\boldsymbol{L}_i\right) \cdot \left\|h_i^{k+1} - h_i^k\right\|_{\boldsymbol{L}_i^{-1}}^2 .$$

Thus, it is not hard to see that

$$\mathbb{E}_{\mathcal{S}}\left[\left\|g^{k+1} - h^{k+1}\right\|_{\boldsymbol{D}}^2\right] \le \frac{2\Lambda_{\boldsymbol{D},\mathcal{S}} \cdot \lambda_{\max}\left(\boldsymbol{D}^{-1}\right) \cdot \lambda_{\max}\left(\boldsymbol{D}\right)}{n^2} \sum_{i=1}^n \lambda_{\max}\left(\boldsymbol{L}_i\right) \left\|h_i^{k+1} - h_i^k\right\|_{\boldsymbol{L}_i^{-1}}^2$$

$$+ \frac{2a^2\Lambda_{\boldsymbol{D},\mathcal{S}} \cdot \lambda_{\max}\left(\boldsymbol{D}^{-1}\right)}{n^2} \sum_{i=1}^n \left\|g_i^k - h_i^k\right\|_{\boldsymbol{D}}^2$$

$$+ (1-a)^2 \left\|g^k - h^k\right\|_{\boldsymbol{D}}^2 .$$

We obtain the inequality in the lemma after taking expectation again and applying tower property.

## H.4 PROOF OF LEMMA 5

Similarly, we then try to bound the terms $\mathbb{E}_{\mathcal{S}}\left[\left\|g_i^{k+1} - h_i^{k+1}\right\|_{\boldsymbol{D}}^2\right]$. We start with

$$\mathbb{E}_{\mathcal{S}}\left[\left\|g_i^{k+1} - h_i^{k+1}\right\|_{\boldsymbol{D}}^2\right]$$

$$= \mathbb{E}_{\mathcal{S}}\left[\left\|g_i^k + \boldsymbol{S}_i^k\left(h_i^{k+1} - h_i^k - a(g_i^k - h_i^k)\right) - h_i^{k+1}\right\|_{\boldsymbol{D}}^2\right]$$

$$= \mathbb{E}_{\mathcal{S}}\left[\left\|\boldsymbol{S}_i^k\left(h_i^{k+1} - h_i^k - a(g_i^k - h_i^k)\right) - \left(h_i^{k+1} - h_i^k - a(g_i^k - h_i^k)\right) + (1-a)(h_i^k - g_i^k)\right\|_{\boldsymbol{D}}^2\right] .$$

Using Fact 3,

$$\mathbb{E}_{\mathcal{S}}\left[\left\|g_i^{k+1} - h_i^{k+1}\right\|_{\boldsymbol{D}}^2\right]$$

$$= \mathbb{E}_{\mathcal{S}}\left[\left\|\boldsymbol{S}_i^k\left(h_i^{k+1} - h_i^k - a(g_i^k - h_i^k)\right) - \left(h_i^{k+1} - h_i^k - a(g_i^k - h_i^k)\right)\right\|_{\boldsymbol{D}}^2\right]$$

$$+ (1-a)^2 \left\|h_i^k - g_i^k\right\|_{\boldsymbol{D}}^2 .$$

Using Lemma 2

$$\mathbb{E}_{\mathcal{S}}\left[\left\|g_i^{k+1} - h_i^{k+1}\right\|_{\boldsymbol{D}}^2\right]$$

$$\overset{(22)}{\le} \lambda_{\max}\left(\boldsymbol{D}^{-\frac{1}{2}}\left(\mathbb{E}\left[\boldsymbol{S}_i^k \boldsymbol{D} \boldsymbol{S}_i^k\right] - \boldsymbol{D}\right)\boldsymbol{D}^{-\frac{1}{2}}\right) \left\|h_i^{k+1} - h_i^k - a(g_i^k - h_i^k)\right\|_{\boldsymbol{D}}^2$$

$$+ (1-a)^2 \left\|g_i^k - h_i^k\right\|_{\boldsymbol{D}}^2$$

$$\le \lambda_{\max}\left(\boldsymbol{D}^{-1}\right) \cdot \Lambda_{\boldsymbol{D},\mathcal{S}} \left\|h_i^{k+1} - h_i^k - a(g_i^k - h_i^k)\right\|_{\boldsymbol{D}}^2 + (1-a)^2 \left\|g_i^k - h_i^k\right\|_{\boldsymbol{D}}^2$$

$$\le 2\lambda_{\max}\left(\boldsymbol{D}^{-1}\right) \cdot \Lambda_{\boldsymbol{D},\mathcal{S}} \left\|h_i^{k+1} - h_i^k\right\|_{\boldsymbol{D}}^2 + 2a^2\lambda_{\max}\left(\boldsymbol{D}^{-1}\right) \cdot \Lambda_{\boldsymbol{D},\mathcal{S}} \left\|g_i^k - h_i^k\right\|_{\boldsymbol{D}}^2$$

$$+ (1-a)^2 \left\|g_i^k - h_i^k\right\|_{\boldsymbol{D}}^2$$

$$\le 2\lambda_{\max}\left(\boldsymbol{D}^{-1}\right) \cdot \lambda_{\max}\left(\boldsymbol{D}\right) \cdot \Lambda_{\boldsymbol{D},\mathcal{S}} \cdot \lambda_{\max}\left(\boldsymbol{L}_i\right) \cdot \left\|h_i^{k+1} - h_i^k\right\|_{\boldsymbol{L}_i^{-1}}^2$$

$$+ 2a^2\lambda_{\max}\left(\boldsymbol{D}^{-1}\right) \cdot \Lambda_{\boldsymbol{D},\mathcal{S}} \left\|g_i^k - h_i^k\right\|_{\boldsymbol{D}}^2 + (1-a)^2 \left\|g_i^k - h_i^k\right\|_{\boldsymbol{D}}^2$$

$$= \left(2a^2\lambda_{\max}\left(\boldsymbol{D}^{-1}\right) \cdot \Lambda_{\boldsymbol{D},\mathcal{S}} + (1-a)^2\right) \left\|g_i^k - h_i^k\right\|_{\boldsymbol{D}}^2$$

$$+ 2\lambda_{\max}\left(\boldsymbol{D}^{-1}\right) \cdot \lambda_{\max}\left(\boldsymbol{D}\right) \cdot \Lambda_{\boldsymbol{D},\mathcal{S}} \cdot \lambda_{\max}\left(\boldsymbol{L}_i\right) \cdot \left\|h_i^{k+1} - h_i^k\right\|_{\boldsymbol{L}_i^{-1}}^2 .$$

Taking expectation again, and using tower property, we are able to obtain,

$$\mathbb{E}\left[\left\|g_i^{k+1} - h_i^{k+1}\right\|_{\boldsymbol{D}}^2\right]$$

$$\le \left(2a^2\lambda_{\max}\left(\boldsymbol{D}^{-1}\right) \cdot \Lambda_{\boldsymbol{D},\mathcal{S}} + (1-a)^2\right) \mathbb{E}\left[\left\|g_i^k - h_i^k\right\|_{\boldsymbol{D}}^2\right]$$

$$+ 2\lambda_{\max}\left(\boldsymbol{D}^{-1}\right) \cdot \lambda_{\max}\left(\boldsymbol{D}\right) \cdot \Lambda_{\boldsymbol{D},\mathcal{S}} \cdot \lambda_{\max}\left(\boldsymbol{L}_i\right) \cdot \mathbb{E}\left[\left\|h_i^{k+1} - h_i^k\right\|_{\boldsymbol{L}_i^{-1}}^2\right] .$$

### H.5 PROOF OF LEMMA 6

From Proposition 5, we know that the objective is $\boldsymbol{L}$-smooth. Let $\bar{x}^{k+1} = x^k - \boldsymbol{D} \cdot \nabla f(x^k)$, then $\boldsymbol{L}$-smoothness yields

$$
\begin{aligned}
f(x^{k+1}) & \leq f(x^k) + \langle \nabla f(x^k), x^{k+1} - x^k \rangle + \frac{1}{2} \langle x^{k+1} - x^k, \boldsymbol{L}(x^{k+1} - x^k) \rangle \\
& = f(x^k) + \langle \nabla f(x^k) - \boldsymbol{D}^{-1} \cdot g^k, x^{k+1} - x^k \rangle + \langle \boldsymbol{D}^{-1} \cdot g^k, x^{k+1} - x^k \rangle \\
& \quad + \frac{1}{2} \langle x^{k+1} - x^k, \boldsymbol{L}(x^{k+1} - x^k) \rangle \\
& = f(x^k) + \langle \nabla f(x^k) - \boldsymbol{D}^{-1} \cdot g^k, -g^k \rangle - \langle x^{k+1} - x^k, \boldsymbol{D}^{-1}(x^{k+1} - x^k) \rangle \\
& \quad + \frac{1}{2} \langle x^{k+1} - x^k, \boldsymbol{L}(x^{k+1} - x^k) \rangle .
\end{aligned}
$$

Simplifying the above inner-products we have,

$$
f(x^{k+1}) \leq f(x^k) + \langle \nabla f(x^k) - \boldsymbol{D}^{-1} \cdot g^k, -g^k \rangle - \left\langle x^{k+1} - x^k, \left( \boldsymbol{D}^{-1} - \frac{1}{2} \boldsymbol{L} \right) (x^{k+1} - x^k) \right\rangle .
$$

We then add and subtract $\langle \nabla f(x^k) - \boldsymbol{D}^{-1} \cdot g^k, \boldsymbol{D} \cdot \nabla f(x^k) \rangle$, which

$$
\begin{aligned}
f(x^{k+1}) & \leq f(x^k) + \langle \nabla f(x^k) - \boldsymbol{D}^{-1} \cdot g^k, \boldsymbol{D} \cdot \nabla f(x^k) - g^k \rangle - \langle \nabla f(x^k) - \boldsymbol{D}^{-1} \cdot g^k, \boldsymbol{D} \cdot \nabla f(x^k) \rangle \\
& \quad - \left\langle x^{k+1} - x^k, \left( \boldsymbol{D}^{-1} - \frac{1}{2} \boldsymbol{L} \right) (x^{k+1} - x^k) \right\rangle \\
& = f(x^k) + \left\| \nabla f(x^k) - \boldsymbol{D}^{-1} \cdot g^k \right\|_{\boldsymbol{D}}^2 - \langle \boldsymbol{D}^{-1}(x^{k+1} - \bar{x}^{k+1}), x^k - \bar{x}^{k+1} \rangle \\
& \quad - \left\langle x^{k+1} - x^k, \left( \boldsymbol{D}^{-1} - \frac{1}{2} \boldsymbol{L} \right) (x^{k+1} - x^k) \right\rangle .
\end{aligned}
$$

Decomposing the inner product term we deduce,

$$
\begin{aligned}
f(x^{k+1}) & \leq f(x^k) + \left\| \boldsymbol{D}^{-1} \left( \boldsymbol{D} \cdot \nabla f(x^k) - g^k \right) \right\|_{\boldsymbol{D}}^2 - \left\langle x^{k+1} - x^k, \left( \boldsymbol{D}^{-1} - \frac{1}{2} \boldsymbol{L} \right) (x^{k+1} - x^k) \right\rangle \\
& \quad - \frac{1}{2} \left( \left\| x^{k+1} - \bar{x}^{k+1} \right\|_{\boldsymbol{D}^{-1}}^2 + \left\| x^k - \bar{x}^{k+1} \right\|_{\boldsymbol{D}^{-1}}^2 - \left\| x^{k+1} - x^k \right\|_{\boldsymbol{D}^{-1}}^2 \right) \\
& = f(x^k) + \left\| \boldsymbol{D} \cdot \nabla f(x^k) - g^k \right\|_{\boldsymbol{D}^{-1}}^2 - \left\| x^{k+1} - x^k \right\|_{\boldsymbol{D}^{-1} - \frac{1}{2} \boldsymbol{L}}^2 \\
& \quad - \frac{1}{2} \left( \left\| \boldsymbol{D} \cdot \nabla f(x^k) - g^k \right\|_{\boldsymbol{D}^{-1}}^2 + \left\| \boldsymbol{D} \cdot \nabla f(x^k) \right\|_{\boldsymbol{D}^{-1}}^2 - \left\| x^{k+1} - x^k \right\|_{\boldsymbol{D}^{-1}}^2 \right) .
\end{aligned}
$$

Therefore,

$$
f(x^{k+1}) \leq f(x^k) + \frac{1}{2} \left\| \boldsymbol{D} \nabla f(x^k) - g^k \right\|_{\boldsymbol{D}^{-1}}^2 - \frac{1}{2} \left\| \nabla f(x^k) \right\|_{\boldsymbol{D}}^2 - \frac{1}{2} \left\| x^{k+1} - x^k \right\|_{\boldsymbol{D}^{-1} - \boldsymbol{L}}^2 .
$$

### H.6 PROOF OF LEMMA 7

We start with

$$
\begin{aligned}
\mathbb{E} \left[ \left\| \boldsymbol{T} \boldsymbol{D} t - \boldsymbol{D} t \right\|_{\boldsymbol{D}^{-1}}^2 \right] & = \mathbb{E} \left[ \left\| (\boldsymbol{T} - \boldsymbol{I}_d) \boldsymbol{D} t \right\|_{\boldsymbol{D}^{-1}}^2 \right] \\
& = \left\langle t, \mathbb{E} \left[ \boldsymbol{D}(\boldsymbol{T} - \boldsymbol{I}_d) \boldsymbol{D}^{-1}(\boldsymbol{T} - \boldsymbol{I}_d) \boldsymbol{D} \right] \cdot t \right\rangle \\
& = \left\langle t, \boldsymbol{D} \left( \mathbb{E} \left[ \boldsymbol{T} \boldsymbol{D}^{-1} \boldsymbol{T} \right] - \boldsymbol{D}^{-1} \right) \boldsymbol{D} \cdot t \right\rangle \\
& = \left\langle \boldsymbol{L}^{-\frac{1}{2}} t, \boldsymbol{L}^{\frac{1}{2}} \boldsymbol{D} \left( \mathbb{E} \left[ \boldsymbol{T} \boldsymbol{D}^{-1} \boldsymbol{T} \right] - \boldsymbol{D}^{-1} \right) \boldsymbol{D} \boldsymbol{L}^{\frac{1}{2}} \cdot \boldsymbol{L}^{-\frac{1}{2}} t \right\rangle \\
& \leq \lambda_{\max} \left( \boldsymbol{L}^{\frac{1}{2}} \boldsymbol{D} \mathbb{E} \left[ \boldsymbol{T} \boldsymbol{D}^{-1} \boldsymbol{T} \right] \boldsymbol{D} \boldsymbol{L}^{\frac{1}{2}} - \boldsymbol{L}^{\frac{1}{2}} \boldsymbol{D} \boldsymbol{L}^{\frac{1}{2}} \right) \cdot \left\| \boldsymbol{L}^{-\frac{1}{2}} t \right\|^2 \\
& = \lambda_{\max} \left( \boldsymbol{L}^{\frac{1}{2}} \boldsymbol{D} \mathbb{E} \left[ \boldsymbol{T} \boldsymbol{D}^{-1} \boldsymbol{T} \right] \boldsymbol{D} \boldsymbol{L}^{\frac{1}{2}} - \boldsymbol{L}^{\frac{1}{2}} \boldsymbol{D} \boldsymbol{L}^{\frac{1}{2}} \right) \cdot \left\| t \right\|_{\boldsymbol{L}^{-1}}^2
\end{aligned}
$$

This completes the proof.

# I EXPERIMENTS

In this section, we conduct numerical experiments to back up the theoretical results for det-MARINA and det-DASHA. The code for the experiments can be found in `https://anonymous.4open.science/r/detCGD-VR-Code-865B`. All the codes for the experiments are written in Python 3.11 with NumPy and SciPy package. The code was run on a machine with AMD Ryzen 9 5900HX Radeon Graphics @ 3.3 GHz and 8 cores 16 threads. The datasets in LibSVM are typically non-IID real world datasets, and it is randomly distributed across all the clients.

## I.1 THE SETTING

We first state the experiment setting. We are interested in the following logistic regression problem with a non-convex regularizer. The objective is given as

$$f(x) = \frac{1}{n} \sum_{i=1}^{n} f_i(x); \qquad f_i(x) = \frac{1}{m_i} \sum_{j=1}^{m_i} \log\left(1 + e^{-b_{i,j} \cdot \langle a_{i,j}, x \rangle}\right) + \lambda \cdot \sum_{t=1}^{d} \frac{x_t^2}{1 + x_t^2},$$

where $x \in \mathbb{R}^d$ is the model, $(a_{i,j}, b_{i,j}) \in \mathbb{R}^d \times \{-1, 1\}$ is one data point in the dataset of client $i$ whose size is $m_i$. The constant $\lambda > 0$ is the coefficient of the regularizer. Larger $\lambda$ means the model is more regular. For each function $f_i$, its Hessian can be upper bounded by

$$\boldsymbol{L}_i = \frac{1}{m_i} \sum_{i=1}^{m_i} \frac{a_i a_i^\top}{4} + 2\lambda \cdot \boldsymbol{I}_d;$$

and, therefore, the Hessian of $f$ is bounded by

$$\boldsymbol{L} = \frac{1}{\sum_{i=1}^{n} m_i} \sum_{i=1}^{n} \sum_{j=1}^{m_i} \frac{a_i a_i^\top}{4} + 2\lambda \cdot \boldsymbol{I}_d.$$

Due to Proposition 2, it immediately follows that $f_i$ and $f$ satisfy Definition 1 with $\boldsymbol{L}_i \in \mathbb{S}_{++}^d$ and $\boldsymbol{L} \in \mathbb{S}_{++}^d$, respectively.

In the following subsections, we perform several numerical experiments comparing the performance of DCGD, det-CGD, MARINA, DASHA, det-MARINA and det-DASHA. The datasets we used are from the LibSVM repository (Chang & Lin, 2011).

## I.2 COMPARISON OF ALL THE METHODS

In this section, we present several plots which compare all relevant methods to the det-MARINA and det-DASHA. The methods are the following: *(i)* DCGD with scalar stepsize $\gamma_2$, *(ii)* det-CGD with matrix stepsize $\boldsymbol{D}_3^*$, *(iii)* MARINA with scalar stepsize $\gamma_1$, *(iv)* DASHA with scalar stepsize $\gamma_4$, *(v)* det-MARINA with $\boldsymbol{D}_{\boldsymbol{L}^{-1}}^*$, *(vi)* det-DASHA with $\boldsymbol{D}_{\boldsymbol{L}^{-1}}^{**}$. Throughout the experiment, $\varepsilon = 0.01$, and $\lambda = 0.9$, we are using the same Rand-$\tau$ sketch for all the algorithms, and we run all the algorithms for a fixed number of iteration $K = 10000$.

It can be seen in Figure 2, the performance in terms of communication complexity of det-DASHA and det-MARINA is better than their scalar counterpart DASHA and MARINA respectively. This validates the efficiency of using a matrix stepsize over a scalar stepsize. Furthermore, we notice that det-DASHA and det-MARINA have better communication complexity in this case, compared to det-CGD. In addition, we observe variance reduction.

Notice that the optimal stepsizes of det-CGD and DCGD require information of function value differences at $x^\star$. Furthermore, the stepsizes are also constrained by the number of iterations $K$ and the error $\varepsilon^2$. Meanwhile, for the variance reduced methods, we do not require such considerations, which is much more practical in general.

## I.3 IMPROVEMENTS OVER MARINA

The purpose of this experiment is to compare the iteration complexity of MARINA, with det-MARINA using Rand-$\tau$ sketches, thus showing improvements of det-MARINA upon MARINA.

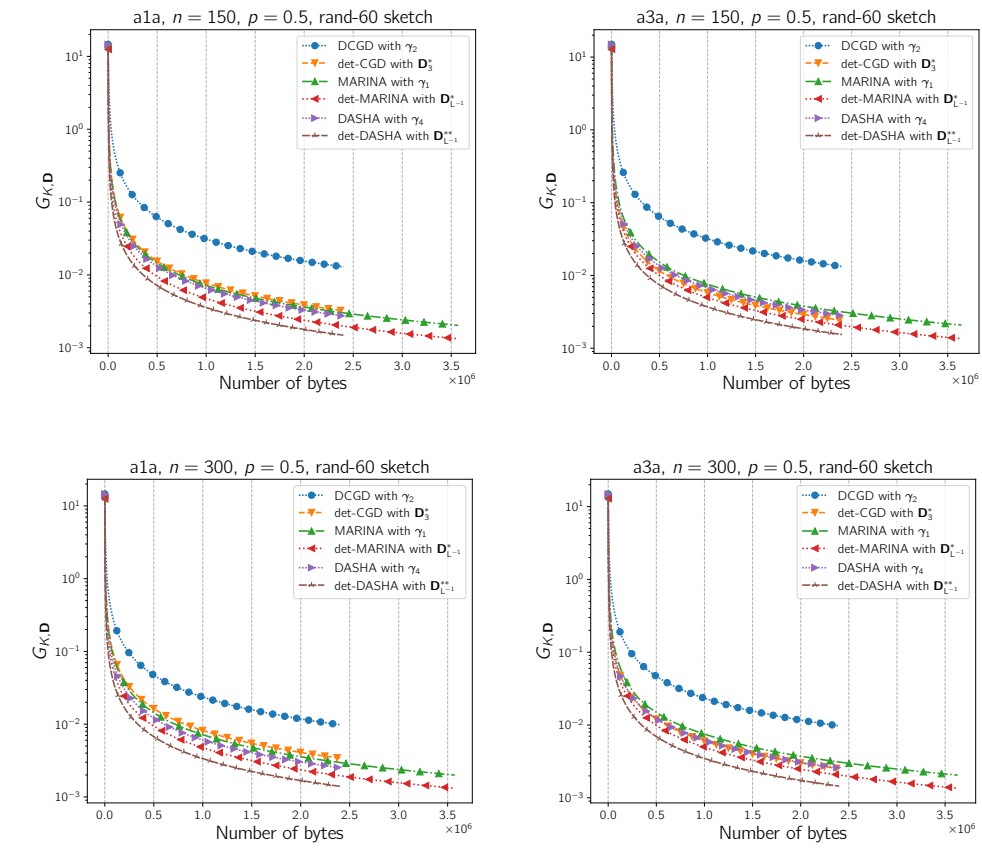

Figure 2: Comparison of DCGD with optimal scalar stepsize, det-CGD with matrix stepsize $D_3^*$, MARINA with optimal scalar stepsize, DASHA with optimal scalar stepsize, det-MARINA with optimal stepsize $D_{L^{-1}}^*$ and det-DASHA with optimal stepsize $D_{L^{-1}}^{**}$. Throughout the experiment, we are using Rand-$\tau$ sketch with $\tau = 60$, and each algorithm is run for a fixed number of iterations $K = 10000$. The momentum of DASHA is set as $1/2\omega+1$ and det-DASHA is $1/2\omega_D+1$. The notation $n$ in the title stands for the number of clients in each case, and $p$ stands for the probability used by MARINA and det-MARINA.

Using Theorem C.1 from (Gorbunov et al., 2021), we deduce the optimal stepsize for MARINA, is

$$\gamma_1 = \frac{1}{L\left(1 + \sqrt{\frac{(1-p)\omega}{pn}}\right)}, \tag{61}$$

where $\omega$ is the quantization coefficient. In particular, for the Rand-$\tau$ compressor $\omega = \frac{d}{\tau} - 1$. For the full definition see Section 1.3 of (Gorbunov et al., 2021). The stepsize for det-MARINA is determined through Corollary 1. We use the notation $D_W^*$ to denote the optimal stepsize for each choice of $W$, here we list some of the optimal stepsizes for different $W$, which are used in the experiment section. We have

$$D_{I_d}^* = \frac{2}{1 + \sqrt{1 + 4\alpha\beta\frac{1}{\lambda_{\max}(L)} \cdot \omega}} \cdot \frac{I_d}{\lambda_{\max}(L)},$$

$$D_{L^{-1}}^* = \frac{2}{1 + \sqrt{1 + 4\alpha\beta \cdot \lambda_{\max}\left(\mathbb{E}\left[S_i^k L^{-1} S_i^k\right] - L^{-1}\right)}} \cdot L^{-1},$$

$$D_{\mathrm{diag}^{-1}(L)}^* = \frac{2}{1 + \sqrt{1 + 4\alpha\beta \cdot \lambda_{\max}\left(\mathbb{E}\left[S_i^k \mathrm{diag}^{-1}(L) S_i^k\right] - \mathrm{diag}^{-1}(L)\right)}} \cdot \mathrm{diag}^{-1}(L) \tag{62}$$

In this experiment, we aim to compare det-MARINA with stepsize $\boldsymbol{D}^*_{\boldsymbol{L}^{-1}}$ to the standard MARINA with the optimal scalar stepsize. Rand-$\tau$ compressor is used in the comparison. Throughout the experiments, $\lambda$ is fixed at $0.3$. We set the $x$-axis to be the number of iterations, while $y$-axis to be the expectation of the corresponding matrix norm of the gradient of the function, which is defined as

$$G_{K,\boldsymbol{D}} = \mathbb{E}\left[\left\|\nabla f(\tilde{x}^K)\right\|^2_{\boldsymbol{D}/\det(\boldsymbol{D})^{1/d}}\right]. \tag{63}$$

Notice that this criterion is comparable to the standard Euclidean norm Li et al. (2024b), and for a fixed $\boldsymbol{D}$, we have

$$\lambda_{\min}\left(\frac{\boldsymbol{D}}{\det(\boldsymbol{D})^{1/d}}\right) \cdot \|\nabla f(x)\|^2 \leq \|\nabla f(x)\|^2_{\frac{\boldsymbol{D}}{\det(\boldsymbol{D})^{1/d}}} \leq \lambda_{\max}\left(\frac{\boldsymbol{D}}{\det(\boldsymbol{D})^{1/d}}\right) \cdot \|\nabla f(x)\|^2.$$

As it is illustrated in Figure 3, det-MARINA always has a faster convergence rate compared to MARINA if they use the same sketch, this justifies the result we have in Corollary 3. Notice that in some cases, det-MARINA with Rand-1 sketch even outperforms standard MARINA with Rand-80 sketch. This further demonstrates the superiority of matrix stepsizes and smoothness over the standard scalar setting.

## I.4 Improvements on non variance reduced methods

In this section, we compare two non-variance reduced methods, distributed compressed gradient descent (DCGD) and distributed det-CGD, with two variance reduced methods, MARINA, and det-MARINA. Rand-1 sketch is used throughout this experiment for all the algorithms, for non variance reduced method $\varepsilon^2$ is fixed at $0.01$ in order to determine the optimal stepsize. The purpose of this experiment is to show the advantages of variance reduced methods over non variance reduced methods. DCGD was initially proposed in (Khirirat et al., 2018). Later on DIANA was proposed in (Mishchenko et al., 2019) and then combined with variance reduction technique. Recently Shulgin & Richtárik (2022) proposed shifted DCGD, which is a shifted version of DCGD and proved its convergence in the (strongly) convex setting. A general analysis on SGD type methods in the non-convex world is provided by Khaled & Richtárik (2023), including DCGD and shifted DCGD. In our case, in order to determine the optimal scalar stepsize for DCGD, one can simply use Proposition 4 in (Khaled & Richtárik, 2023). One can check that in order to satisfy $\min_{0 \leq k \leq K-1} \mathbb{E}\left[\left\|\nabla f(x^k)\right\|^2\right] \leq \varepsilon^2$ the stepsize condition for DCGD in the non-convex case reduces to

$$\gamma_2 \leq \min\left\{\frac{1}{L}, \sqrt{\frac{n}{\omega L L_{\max} K}}, \frac{n\varepsilon^2}{4 L L_{\max}\omega \cdot \Delta^\star}\right\},$$

where $L$ is the smoothness constant for $f$, $L_i$ is the smoothness constant for $f_i$, $L_{\max} = \max_i L_i$, $K$ is the total number of iterations, $\Delta^\star = f(x^\star) - \frac{1}{n}\sum_{i=1}^n f_i(x^\star)$. The constant $\omega$ is associated with the compressor used in the algorithm, for Rand-$\tau$ sketch, it is $\frac{d}{\tau} - 1$. For distributed det-CGD according to Li et al. (2024b), the stepsize condition in order to satisfy $\min_{0 \leq k \leq K-1} \mathbb{E}\left[\|\nabla f(x)\|^2_{\boldsymbol{D}/\det(\boldsymbol{D})^{1/d}}\right] \leq \varepsilon^2$ is

$$\boldsymbol{D}\boldsymbol{L}\boldsymbol{D} \preceq \boldsymbol{D}, \qquad \lambda_{\boldsymbol{D}} \leq \min\left\{\frac{n}{K}, \frac{n\varepsilon^2}{4\Delta^\star}\det(\boldsymbol{D})^{1/d}\right\}, \tag{64}$$

where $\lambda_{\boldsymbol{D}}$ is defined as

$$\lambda_{\boldsymbol{D}} = \max_i\left\{\lambda_{\max}\left(\mathbb{E}\left[\boldsymbol{L}_i^{\frac{1}{2}}\left(\boldsymbol{S}_i^k - \boldsymbol{I}_d\right)\boldsymbol{D}\boldsymbol{L}\boldsymbol{D}\left(\boldsymbol{S}_i^k - \boldsymbol{I}_d\right)\boldsymbol{L}_i^{\frac{1}{2}}\right]\right)\right\}. \tag{65}$$

In general cases, there is no easy way to find a optimal stepsize matrix $\boldsymbol{D}$ satisfying (64), alternatively, we choose the optimal diagonal stepsize $\boldsymbol{D}_3^*$ similarly to (Li et al., 2024b). The stepsize condition for MARINA has already been described by (61). Note that we only consider MARINA, but not DIANA or shifted DCGD, because DIANA and shifted DCGD offer suboptimal rates compared to MARINA in the non-convex setting. For det-MARINA, we fix $\boldsymbol{W} = \boldsymbol{L}^{-1}$, and use $\boldsymbol{D}^*_{\boldsymbol{L}^{-1}}$ as the stepsize matrix. In theory, det-MARINA in this case should always out perform MARINA in terms of iteration complexity.

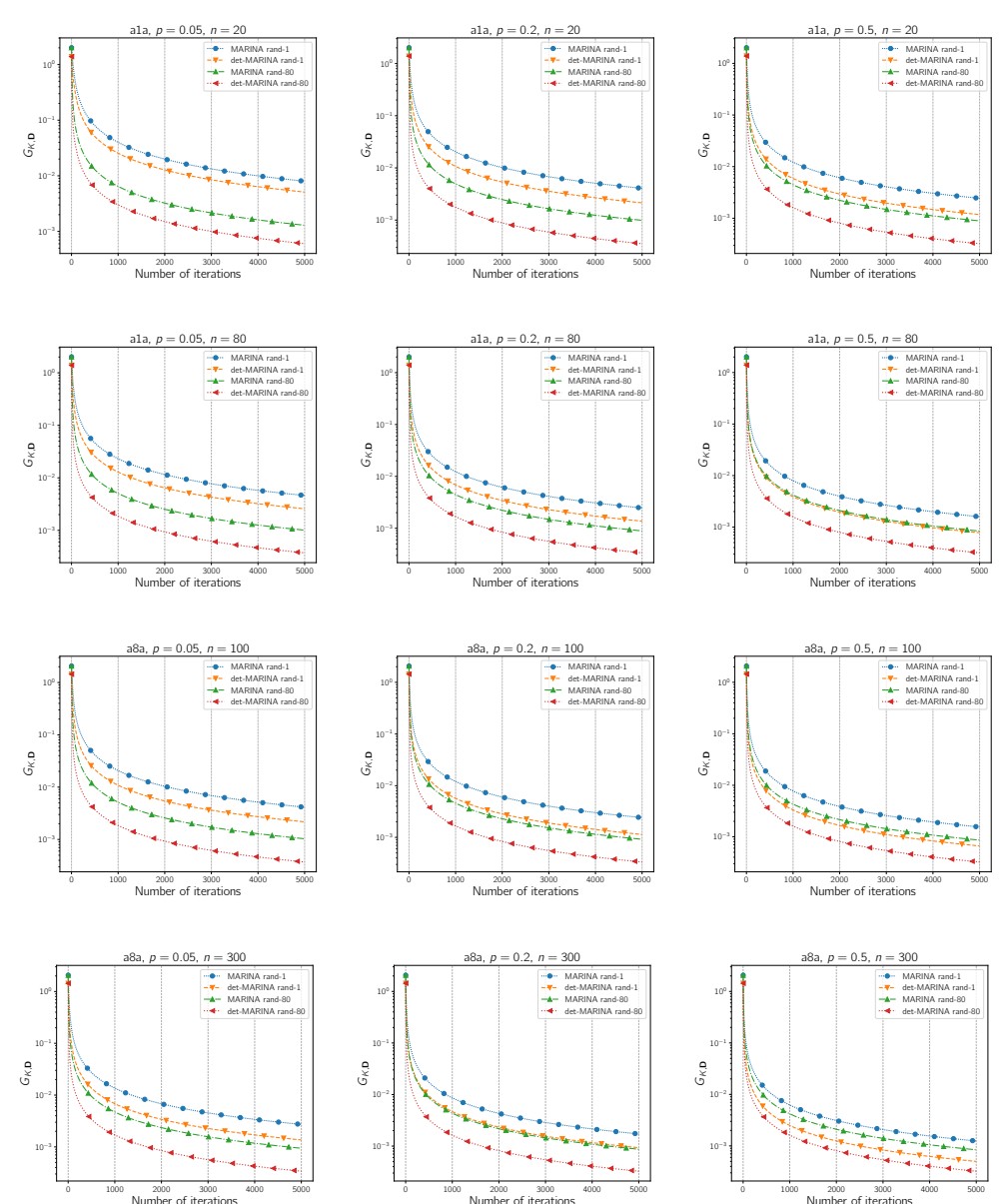

Figure 3: In this experiment, we aim to compare det-MARINA with stepsize $D^*_{L^{-1}}$ to the standard MARINA with the optimal scalar stepsize. Rand-$\tau$ compressor is used in the comparison. Throughout the experiments, $\lambda$ is fixed at $0.3$. Optimal stepsize is calculated in each case with respect to the sketch used. The $x$-axis denotes the number of iterations while the notation $G_{K,D}$ for the $y$-axis is defined in (63), which is the averaged matrix norm of the gradient. The notation $p$ in the title denotes the probability used in the two algorithms, $n$ denotes the number of clients in each setting.

In Figure 4, in each plot, we observe that det-MARINA outperforms MARINA and the rest of the non-variance reduced methods. This is expected, since our theory confirms that det-MARINA indeed has a better rate compared to MARINA, and the stepsizes of the non-variance reduced methods are negatively affected by the neighborhood. When $p$ is reasonably large, the variance reduced methods considered here outperform the non-variance reduced methods. In this experiment we consider only the comparison involving det-CGD.

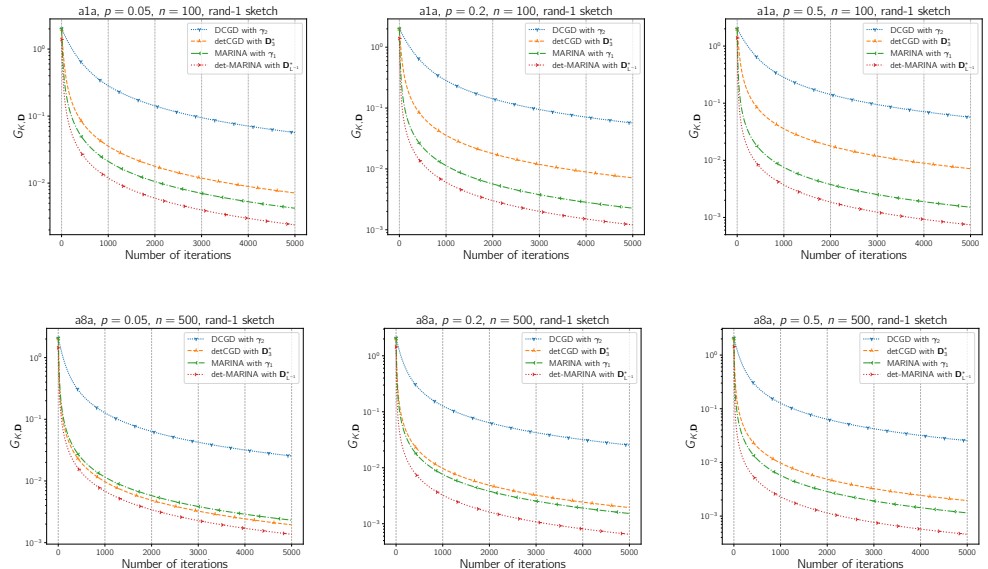

Figure 4: Comparison of DCGD with optimal scalar stepsize $\gamma_2$, det-CGD with optimal diagonal stepsize $\boldsymbol{D}_3^*$, MARINA with optimal scalar stepsize $\gamma_1$, and det-MARINA with optimal stepsize $\boldsymbol{D}_{\boldsymbol{L}^{-1}}^*$ with respect to $\boldsymbol{W} = \boldsymbol{L}^{-1}$. In each case, probability $p$ is chosen the set $\{0.05, 0.2, 0.5\}$ for MARINA and det-MARINA. $\lambda = 0.3$ is fixed throughout the experiment. The notation $n$ in the title indicates the number of clients in each case.

## I.5 IMPROVEMENTS OVER DET-CGD

In this section, we compare det-CGD in the distributed case with det-MARINA, which are both algorithms using matrix stepsizes and matrix smoothness. The purpose of this experiment is to show that det-MARINA improves on the current state of the art matrix stepsize compressed gradient method when the objective function is non-convex. Throughout the experiment, $\lambda = 0.3$ is fixed, and for det-CGD, $\varepsilon^2 = 0.01$ is fixed in order to determine its stepsize. For a thorough comparison, we select the stepsize for det-CGD in the following way. Let us denote the stepsize as $\boldsymbol{D} = \gamma_{\boldsymbol{W}} \cdot \boldsymbol{W}$, where $\gamma_{\boldsymbol{W}} \in \mathbb{R}_{++}, \boldsymbol{W} \in \mathbb{S}_{++}^d$. We first fix a matrix $\boldsymbol{W}$, in this case, we pick $\boldsymbol{W}$ from the set $\{\boldsymbol{L}^{-1}, \operatorname{diag}^{-1}(\boldsymbol{L}), \boldsymbol{I}_d\}$, and then we determine the optimal scaling $\gamma_{\boldsymbol{W}}$ for each case using the condition given in (Li et al., 2024b) (see (64) and (65)). Then, we denote the matrix stepsizes for det-CGD

$$\boldsymbol{D}_1 = \gamma_{\boldsymbol{I}_d} \cdot \boldsymbol{I}_d, \qquad \boldsymbol{D}_2 = \gamma_{\operatorname{diag}^{-1}(\boldsymbol{L})} \cdot \operatorname{diag}^{-1}(\boldsymbol{L}), \qquad \boldsymbol{D}_3 = \gamma_{\boldsymbol{L}^{-1}} \cdot \boldsymbol{L}^{-1}. \tag{66}$$

For det-MARINA, we use the stepsize $\boldsymbol{D}_{\boldsymbol{L}^{-1}}^*$, which is described in (62). In this experiment, we compare det-CGD using three stepsizes $\boldsymbol{D}_1, \boldsymbol{D}_2, \boldsymbol{D}_3$ with det-MARINA using stepsize $\boldsymbol{D}_{\boldsymbol{L}^{-1}}^*$.

From Figure 5, it is clear that det-MARINA outperforms det-CGD with all matrix optimal stepsizes with respect to a fixed $\boldsymbol{W}$ considered here. This is expected, since the convergence rate of non-variance reduced methods are affected by its neighborhood. This experiment demonstrates the advantages of det-MARINA over det-CGD, and is also supported by our theory. Notice that though different $\boldsymbol{W}$ are considered for det-CGD, their convergence rates are similar, which is also mentioned by Li et al. (2024b).

## I.6 COMPARING DIFFERENT STEPSIZE CHOICES

This experiment is designed to see the how det-MARINA works under different stepsize choices. As it is mentioned in Appendix I.3, for each choice of $\boldsymbol{W} \in \mathbb{S}_{++}^d$, an optimal stepsize $\boldsymbol{D}_{\boldsymbol{W}}^*$ can be determined. Here we compare det-MARINA using three different stepsize choices $\boldsymbol{D}_{\boldsymbol{L}^{-1}}^*, \boldsymbol{D}_{\operatorname{diag}^{-1}(\boldsymbol{L})}^*$ and $\boldsymbol{D}_{\boldsymbol{I}_d}^*$. There stepsizes are explicitly defined in (62). Throughout the experiment, we fix $\lambda = 0.3$, Rand-1 sketch is used in all cases.

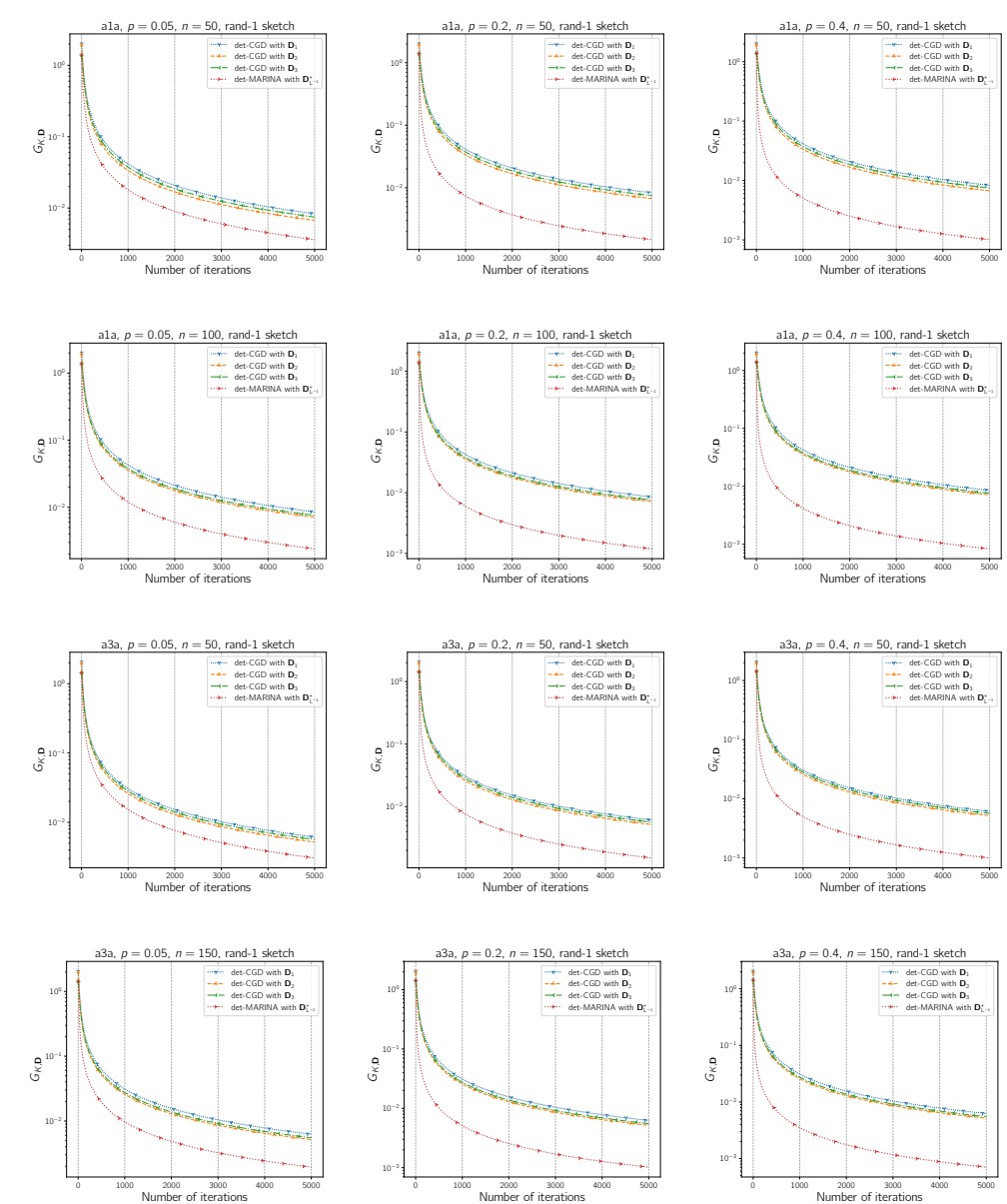

Figure 5: Comparison of det-CGD with matrix stepsize $D_1$, $D_2$ and $D_3$ and det-MARINA with optimal matrix stepsize with respect to $W = L^{-1}$. The stepsizes $\{D_i\}_{i=1}^3$ are described in (66). Throughout the experiment $\varepsilon^2$ is fixed at $0.01$, the notation $p$ in the title refers to the probability for det-MARINA, $n$ denotes the number of clients considered, Rand-1 sketch is used in all cases for all the algorithms.

We can observe from Figure 6 that, in almost all cases det-MARINA with stepsize $D^*_{\mathrm{diag}^{-1}(L)}$ and $D^*_{L^{-1}}$ outperforms det-MARINA with $D^*_{I_d}$. As det-MARINA with $D^*_{I_d}$ can be viewed as MARINA using scalar stepsize but under matrix Lipschitz gradient assumption, this demonstrates the effectiveness of using a matrix stepsize over the scalar stepsize. However, in Figure 6, there are cases where det-MARINA with $D^*_{\mathrm{diag}^{-1}(L)}$ outperforms $D^*_{L^{-1}}$. This tells us the two stepsizes are perhaps incomparable in general cases. This is similar to det-CGD, where optimal stepsizes with respect to a subspace associated with a fixed $W^{-1}$ are incomparable.

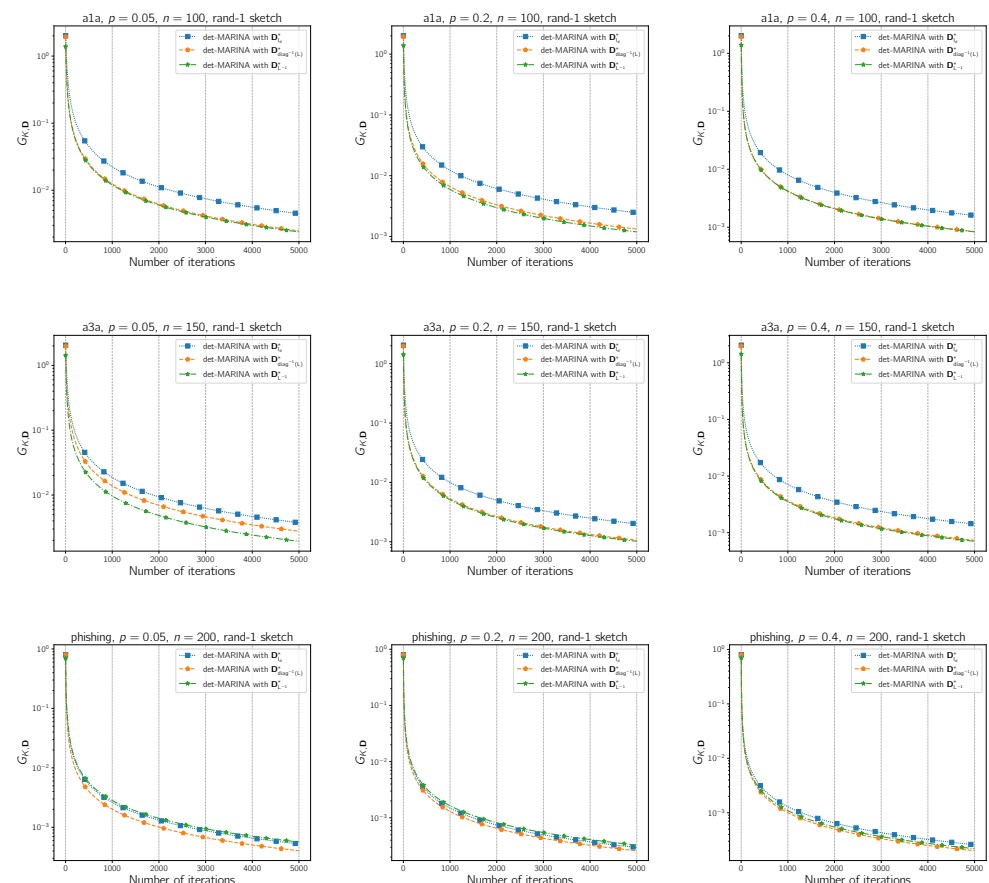

Figure 6: Comparison of det-MARINA with matrix stepsize $\boldsymbol{D}_{\boldsymbol{I}_d}^*$, $\boldsymbol{D}_{\text{diag}^{-1}(\boldsymbol{L})}^*$ and $\boldsymbol{D}_{\boldsymbol{L}^{-1}}^*$. The stepsizes are defined in (62). Throughout the experiment, $\lambda = 0.3$ is fixed, Rand-1 sketch is used in all cases. The notation $p$ in the title indicates the probability of sending the true gradient for det-MARINA, $n$ denotes the number of clients considered.

## I.7 COMPARING COMMUNICATION COMPLEXITY

In this section, we perform an experiment on how different probabilities $p$ will affect the overall communication complexity of det-MARINA. We use $\boldsymbol{D}_{\boldsymbol{L}^{-1}}^*$ as the stepsize, which is determined with respect to the sketch used. Rand-$\tau$ sketches are used in these experiments, and we vary the minibatch size $\tau$ to provide a more comprehensive comparison. For Rand-$\tau$ sketch $\boldsymbol{S}$ and any $\boldsymbol{A} \in \mathbb{S}_{++}^d$, one can show that

$$\mathbb{E}\left[\boldsymbol{S}\boldsymbol{A}\boldsymbol{S}\right] = \frac{d}{\tau}\left(\frac{d-\tau}{d-1}\operatorname{diag}(\boldsymbol{A}) + \frac{\tau-1}{d-1}\boldsymbol{A}\right). \tag{67}$$

Combining (67) and (62), we can find out the corresponding matrix stepsize easily. In the experiment, a fixed number of iterations ($K = 5000$) is performed for each det-MARINA with the corresponding stepsize.

As it can be observed from Figure 7, in each dataset, the communication complexity tends to increase with the increase of probability $p$. However, when the number of iteration is fixed, a larger $p$ often means a faster rate of convergence. This difference in communication complexity is more obvious when we are using the Rand-1 sketch. In real federated learning settings, there is often constraints on network bandwidth from clients to the server. Thus, trading off between communication complexity and iteration complexity, i.e. selecting the compression mechanism carefully to guarantee a acceptable speed that satisfies the bandwidth constraints, becomes important.

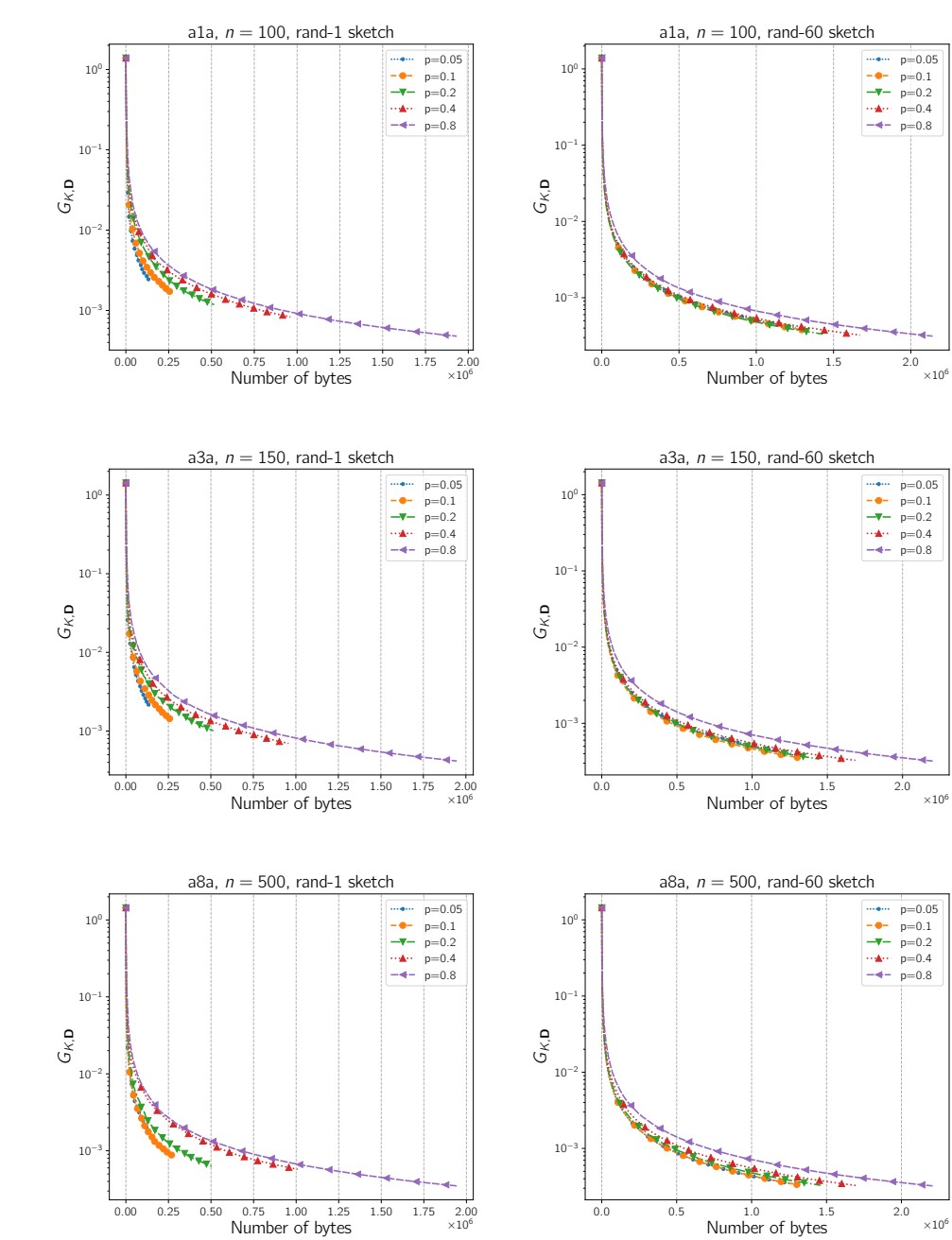

Figure 7: Comparison of det-MARINA with stepsize $\boldsymbol{D}_{\boldsymbol{L}^{-1}}^{*}$ using different probability $p$. The probability $p$ here is chosen from the set $\{0.05, 0.1, 0.2, 0.4, 0.8\}$. The notation $n$ in the title denote the number of clients considered. The $x$-axis is now the number of bytes sent from a single node to the server. In each case, det-MARINA is run for a fixed number of iterations $K = 5000$.

## I.8 COMPARISON OF DASHA AND DET-DASHA

In this experiment we plan to compare the performance of original DASHA with det-DASHA. Throughout the experiments, $\lambda$ is fixed at $0.3$. The same Rand-$\tau$ sketch is used in the two algo-

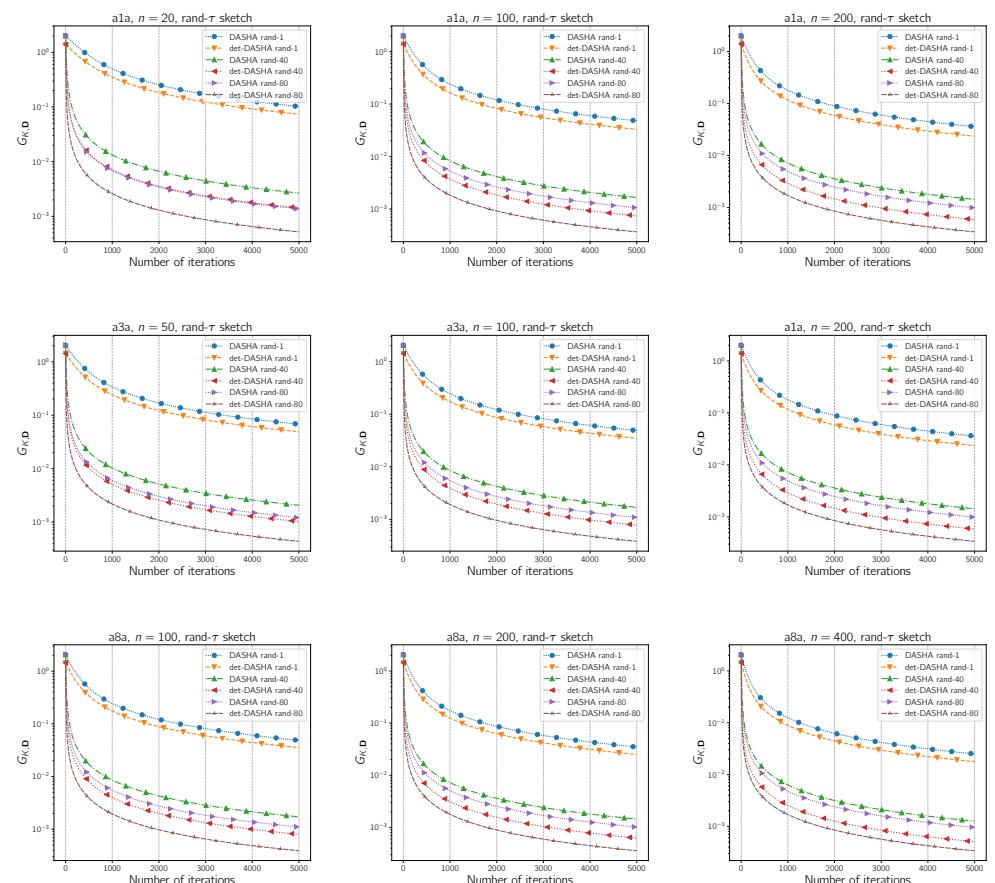

Figure 8: Comparison of det-DASHA with matrix stepsize $\boldsymbol{D}^{**}_{\boldsymbol{L}^{-1}}$ and DASHA with optimal scalar stepsize $\gamma$ using different Rand-$\tau$ sketches. $\lambda = 0.3$ is fixed throughout the experiments. Optimal stepsize is calculated in each case with respect to the sketch used. The $x$-axis denotes the number of iterations while the notation $G_{K,\boldsymbol{D}}$ for the $y$-axis denotes the averaged matrix norm of the gradient. The notation $n$ denotes the number of clients in each setting.

rithms. The stepsize condition on DASHA when the momentum is set as $a = \frac{1}{2\omega+1}$ is given as

$$\gamma_4 \leq \left( L + \sqrt{\frac{16\omega(2\omega+1)}{n}}\widehat{L} \right)^{-1},$$

according to Theorem 6.1 of Tyurin & Richtárik (2024). Here the $L$ is the smoothness constant of the function $f$, while $\widehat{L}$ satisfies $\widehat{L}^2 = \frac{1}{n}\sum_{i=1}^{n} L_i^2$ where $L_i$ is the smoothness constant of local objective $f_i$. In theory we can pick $\widehat{L} = L$. Similarly, according to Corollary 2, the optimal stepsize matrix $\boldsymbol{D}^{**}_{\boldsymbol{L}^{-1}}$ is given as

$$\boldsymbol{D}^{**}_{\boldsymbol{L}^{-1}} = \frac{2}{1 + \sqrt{1 + 16C_{\boldsymbol{L}^{-1}} \cdot \lambda_{\min}(\boldsymbol{L})}} \cdot \boldsymbol{L}^{-1}, \tag{68}$$

when the momentum is given as $a = \frac{1}{2\omega_D+1}$. We compare the performance of DASHA with $\omega$ and det-DASHA with $\boldsymbol{D}^{**}_{\boldsymbol{L}^{-1}}$ using the same sketch where the total number of clients are different.

As it can be observed in Figure 8, det-DASHA with matrix stepsize $\boldsymbol{D}^{**}_{\boldsymbol{L}^{-1}}$ outperforms DASHA with optimal scalar stepsize using the same sketch in every setting we considered. Note that since the same sketch is used in the two algorithm, the number of bits transferred in each iteration is also the same for the two algorithms. This essentially indicates that det-DASHA has better iteration

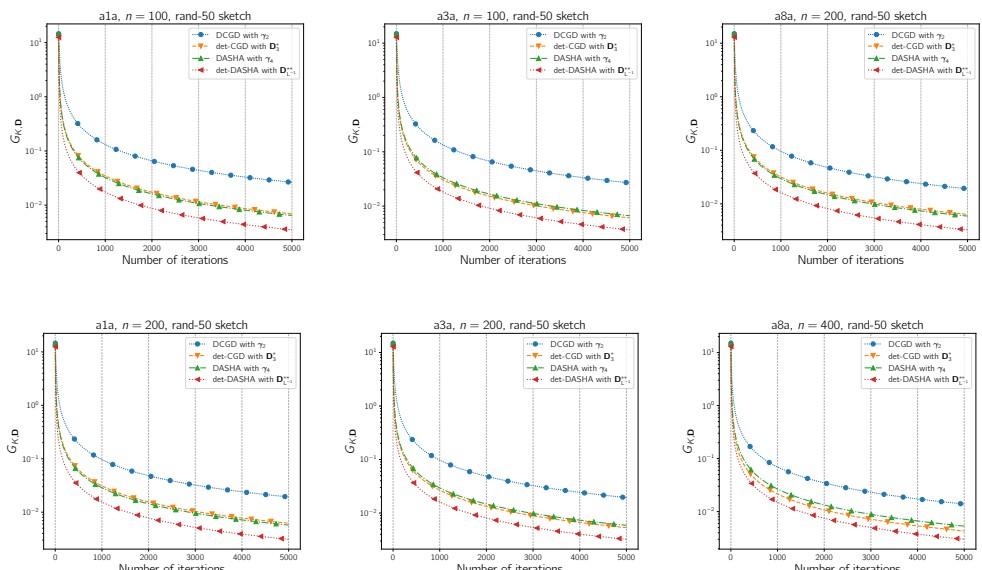

Figure 9: Comparison of DCGD with optimal scalar stepsize $\gamma_2$, det-CGD with optimal diagonal stepsize $\boldsymbol{D}_3^*$, DASHA with optimal scalar stepsize $\gamma_1$, and det-DASHA with optimal stepsize $\boldsymbol{D}_{\boldsymbol{L}^{-1}}^{**}$ with respect to $\boldsymbol{W} = \boldsymbol{L}^{-1}$. $\lambda = 0.9$ is fixed throughout the experiment. The notation $n$ in the title indicates the number of clients in each case. Rand-$\tau$ sketch with $\tau = 50$ are used in all four algorithms.

complexity as well as communication complexity than DASHA given that the same sketch is used for the two algorithm.

## I.9 COMPARISON OF DCGD, DET-CGD, DASHA AND DET-DASHA

In this experiment, we consider the comparison between the two non variance reduced methods DCGD, det-CGD and the two variance reduced method DASHA, det-DASHA. The stepsize choices for DCGD and det-CGD have already been discussed in the previous sections, (for DCGD we use $\gamma_2$ and for det-CGD we use $\boldsymbol{D}_3^*$) ,for DASHA and det-DASHA, we use the stepsize choices of Appendix I.8. Note that $\varepsilon^2$ is set as $0.01$, and $\lambda$ is fixed at $0.9$ here. Throughout this experiment, we consider the case where Rand-$\tau$ sketch is used in the four algorithms.

It is easy to observe that in each case of Figure 9, det-DASHA outperforms the rest of the algorithms. It is expected that det-DASHA outperforms DASHA, as it is also illustrated by Figure 8, which is a consequence of using matrix stepsize instead of a scalar stepsize. We also see that det-DASHA and DASHA outperform det-CGD and DCGD respectively, which demonstrate the advantages of the variance reduction technique. Note that in this case, all four algorithms are using the same sketch, which means that the number of bits transferred in each iteration is the same for the four algorithms, as a result, compared to the other algorithms, det-DASHA is better in terms of both iteration complexity and communication complexity.

## I.10 COMPARISON OF DET-DASHA AND DET-CGD WITH DIFFERENT STEPSIZES

In this experiment, we try to compare det-DASHA and det-CGD with different matrix stepsizes. Throughout this experiment, we will fix $\varepsilon^2 = 0.01$ and $\lambda = 0.9$. The same Rand-$\tau$ sketch is used for the two algorithms. For det-CGD, we use the stepsize $\boldsymbol{D}_1 = \gamma_{\boldsymbol{I}_d} \cdot \boldsymbol{I}_d, \boldsymbol{D}_2 = \gamma_{\mathrm{diag}^{-1}(\boldsymbol{L})} \cdot \mathrm{diag}^{-1}(\boldsymbol{L})$ and $\boldsymbol{D}_3 = \gamma_{\boldsymbol{L}^{-1}} \cdot \boldsymbol{L}^{-1}$, for det-DASHA we use the stepsize $\boldsymbol{D}_{\boldsymbol{L}^{-1}}^{**}$.

It can be observed that in all cases of Figure 10, det-DASHA outperforms det-CGD with different stepsizes. This further corroborates our theory that det-DASHA is variance reduced and thus is better in terms of both iteration complexity, and communication complexity (because in this case

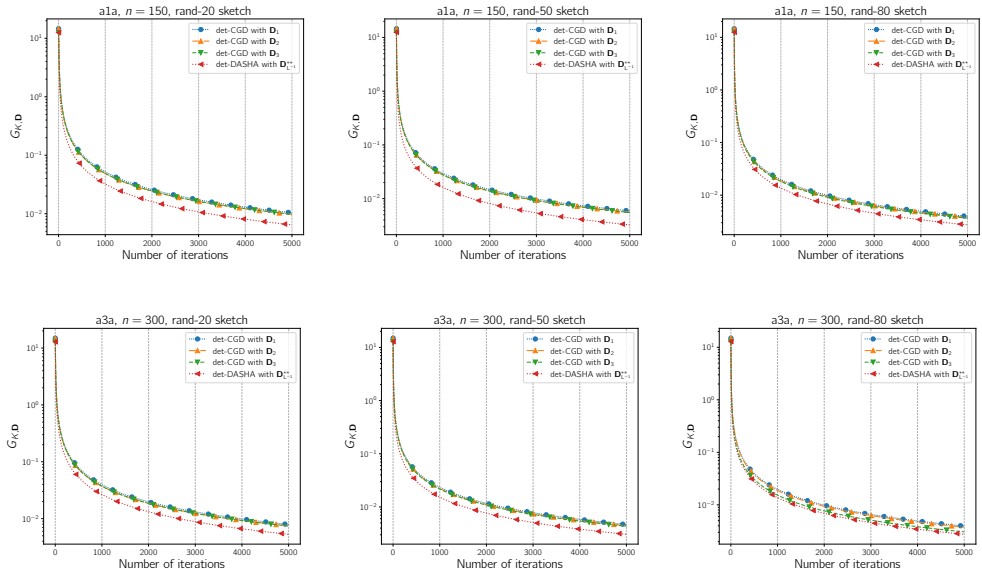

Figure 10: Comparison of det-DASHA with stepsize $\boldsymbol{D}^{**}_{\boldsymbol{L}^{-1}}$ and det-CGD with three different stepsizes $\boldsymbol{D}_1$, $\boldsymbol{D}_2$ and $\boldsymbol{D}_3$. Throughout the experiment, $\lambda$ is fixed at $0.9$, $\varepsilon^2$ is fixed at $0.01$, Rand-$\tau$ sketch is used for all the algorithms with $\tau$ selected from $\{20, 50, 80\}$. The notation $n$ denotes the number of clients in each setting.

the same number of bits are transmitted in each iteration due to the fact that the sketch used is the same).

### I.11  COMPARISON OF DIFFERENT STEPSIZES OF DET-DASHA

In this experiment, we try to compare det-DASHA with different matrix stepsizes. Specifically, we fix matrix $\boldsymbol{W}$ to be three different matrices, $\boldsymbol{I}_d$, $\mathrm{diag}^{-1}(\boldsymbol{L})$ and $\boldsymbol{L}^{-1}$. We denote the optimal stepsizes as $\boldsymbol{D}^{**}_{\boldsymbol{I}_d}$, $\boldsymbol{D}^{**}_{\mathrm{diag}^{-1}(\boldsymbol{L})}$ and $\boldsymbol{D}^{**}_{\boldsymbol{L}^{-1}}$, respectively. For $\boldsymbol{D}^{**}_{\boldsymbol{L}^{-1}}$, it is already given in (68), for $\boldsymbol{D}^{**}_{\boldsymbol{I}_d}$ and $\boldsymbol{D}^{**}_{\mathrm{diag}^{-1}(\boldsymbol{L})}$, we use Corollary 2 to compute them. As a result,

$$\boldsymbol{D}^{**}_{\boldsymbol{I}_d} = \frac{2}{1 + \sqrt{1 + 16 \cdot \frac{\omega_{\boldsymbol{I}_d}\left(4\omega_{\boldsymbol{I}_d}+1\right)}{n} \cdot \frac{\lambda_{\min}(\boldsymbol{L})}{\lambda_{\max}(\boldsymbol{L})}}} \cdot \frac{\boldsymbol{I}_d}{\lambda_{\max}(\boldsymbol{L})}, \tag{69}$$

and

$$\boldsymbol{D}^{**}_{\mathrm{diag}^{-1}(\boldsymbol{L})} = \frac{2}{1 + \sqrt{1 + 16C_{\mathrm{diag}^{-1}(\boldsymbol{L})} \cdot \lambda_{\min}(\boldsymbol{L})}} \cdot \mathrm{diag}^{-1}(\boldsymbol{L}). \tag{70}$$

Throughout the experiment, $\lambda$ is fixed at $0.9$, Rand-$\tau$ sketch is used for all the algorithms.

We can observe from Figure 11, det-DASHA with $\boldsymbol{D}^{**}_{\boldsymbol{L}^{-1}}$ and $\boldsymbol{D}^{**}_{\mathrm{diag}^{-1}(\boldsymbol{L})}$ both outperform det-DASHA with $\boldsymbol{D}^{**}_{\boldsymbol{I}_d}$, which demonstrate the effectiveness of using a matrix stepsize instead of a scalar stepsize. However, depending on the parameters of the problem, it is hard to reach a general conclusion whether $\boldsymbol{D}^{**}_{\boldsymbol{L}^{-1}}$ is better than $\boldsymbol{D}^{**}_{\mathrm{diag}^{-1}(\boldsymbol{L})}$ or not.

### I.12  COMPARISON OF DET-MARINA AND DET-DASHA

In this section, we aim to provide a comparison of det-DASHA and det-MARINA. They are similar as they are both variance reduced version of det-CGD. However, the variance reduction techniques that are utilized are different. For det-MARINA, it is based on MARINA, and it requires synchronization from time to time depending on a probability parameter $p$, while for det-DASHA it utilizes

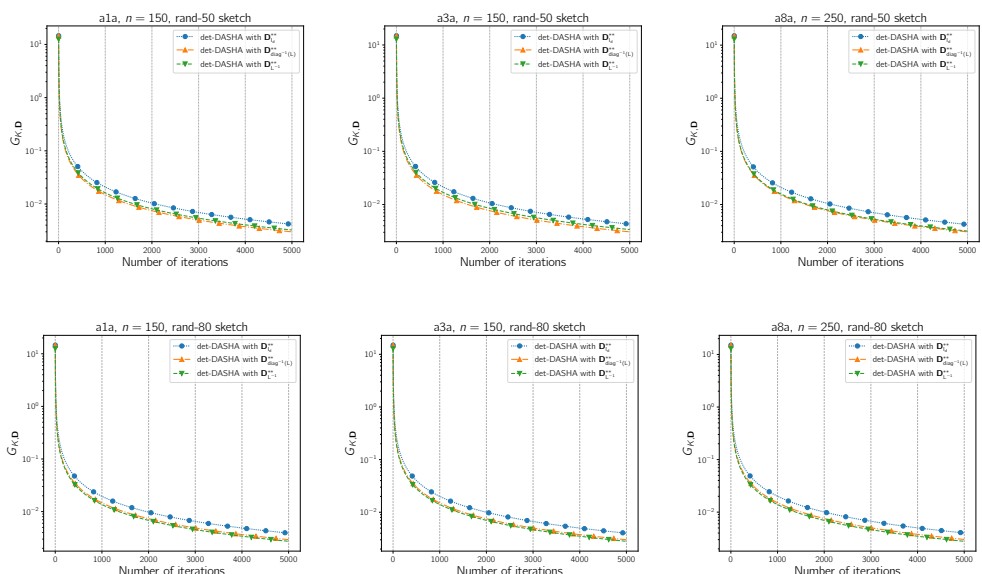

Figure 11: Comparison of det-DASHA three different stepsizes $\boldsymbol{D}^{**}_{\boldsymbol{L}^{-1}}$, $\boldsymbol{D}^{**}_{\mathrm{diag}^{-1}(\boldsymbol{L})}$ and $\boldsymbol{D}^{**}_{\boldsymbol{I}_d}$. The definition for those matrix stepsize notation are given in (68), (70) and (69) respectively. Throughout the experiment, $\lambda$ is fixed at $0.9$, Rand-$\tau$ sketch is used for all the algorithms. The notation $n$ denotes the number of clients in each setting.

the momentum variance reduction technique which was also presented in DASHA, it does not need any synchronization at all. Notice that for a fair comparison, we implement the two algorithms so that they use the same sketch. We mainly focus on the communication complexity, i.e. the convergence with respect to the number of bits transferred. Throughout the experiment, $\lambda = 0.9$ is fixed. For det-DASHA we pick 3 different kinds of stepsizes $\boldsymbol{D}^{**}_{\boldsymbol{I}_d}$, $\boldsymbol{D}^{**}_{\boldsymbol{L}^{-1}}$ and $\boldsymbol{D}^{**}_{\mathrm{diag}^{-1}(\boldsymbol{L})}$. For det-MARINA, we also pick three different kinds of stepsizes correspondingly $\boldsymbol{D}^{*}_{\boldsymbol{I}_d}$, $\boldsymbol{D}^{*}_{\boldsymbol{L}^{-1}}$ and $\boldsymbol{D}^{*}_{\mathrm{diag}^{-1}(\boldsymbol{L})}$. We use the same sketch for all of the algorithms we are trying to compare.

It is obvious from Figure 12 that det-DASHA always has a better communication complexity comparing to the det-MARINA counterpart. Notice that here since each algorithm is run for a fixed number of iterations, so $x$-axis actually records the total number of bytes transferred for each algorithm. For det-DASHA, $\boldsymbol{D}^{**}_{\boldsymbol{L}^{-1}}$ perform similarly to $\boldsymbol{D}^{**}_{\mathrm{diag}^{-1}(\boldsymbol{L})}$, and both are better than $\boldsymbol{D}^{**}_{\boldsymbol{I}_d}$. This is expected since the same sketch is used, and the number of bytes transferred in each iteration is the same for each variant of det-DASHA. The same relation also holds for det-MARINA.

## I.13 COMPARISON IN TERMS OF FUNCTION VALUES

In this section, we compare det-MARINA and det-DASHA in terms of function values. The starting points of the two algorithms are set to be the same, and we run the two algorithms for multiple times and we average the function values we obtained in each iteration. For the two algorithms, we use the same sketch, and since we are interested in the performance in terms of communication complexity, we use the number of bytes transferred in the training process as the $x$-axis. We run each of the algorithm for 20 times, and fix $\lambda = 0.9$. The starting point is fixed throughout the experiment. We pick $\boldsymbol{D}^{**}_{\boldsymbol{L}^{-1}}$ as the stepsize of det-DASHA, while $\boldsymbol{D}^{*}_{\boldsymbol{L}^{-1}}$ as the stepsize of det-MARINA.

Observing Figure 13, we can see that the function values continuously decrease as the algorithms progress through more iterations. However, the stability observed here differs from the case of the average (matrix) norm of gradients. Our theoretical framework, as presented in this paper, primarily addresses the average norm of gradients in the non-convex case. Despite this, the experiment reinforces the effectiveness of our algorithms, showcasing consistent decreases in function values.

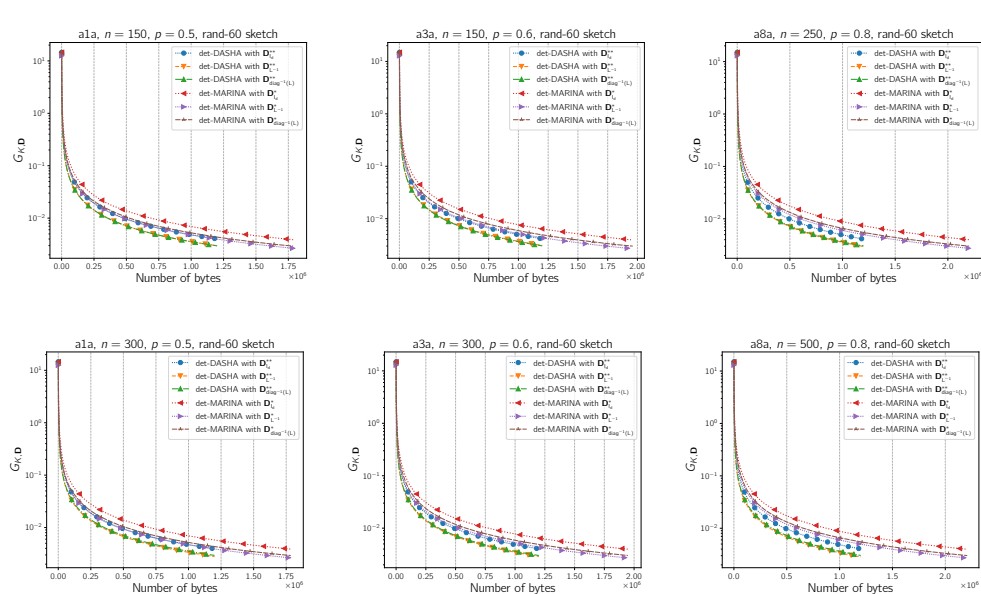

Figure 12: Comparison of det-DASHA with three different stepsizes $D^{**}_{I_d}$, $D^{**}_{L^{-1}}$ and $D^{**}_{\mathrm{diag}^{-1}(L)}$, and det-MARINA with $D^*_{I_d}$, $D^*_{L^{-1}}$ and $D^*_{\mathrm{diag}^{-1}(L)}$ in terms of communication complexity. Throughout the experiment, $\lambda$ is fixed at $0.9$, the same Rand-$\tau$ sketch is used for all the algorithms. The notation $n$ denotes the number of clients in each setting. Each algorithm is run for a fixed number of iteration $K = 5000$.

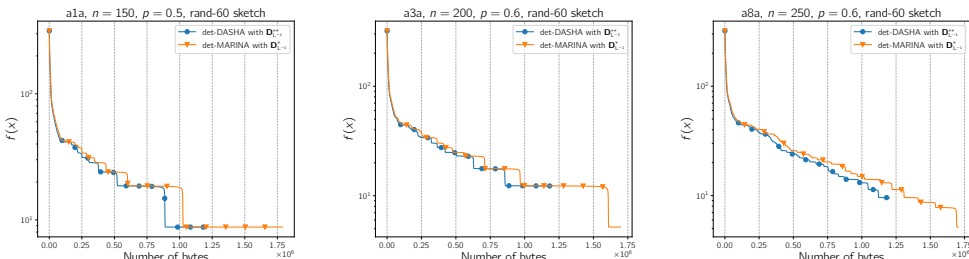

Figure 13: Comparing the performance of det-DASHA with $D^{**}L^{-1}$ and det-MARINA with $D^*L^{-1}$ in terms of the decreasing function values. The function values for each algorithm represent an average of 20 runs using different random seeds. Here, $\lambda = 0.9$ is fixed throughout the experiment, and the starting point for the two algorithms in different runs is the same. The notation $n$ stands for the number of clients, and $p$ represents the probability used in det-MARINA. The same Rand-$\tau$ sketch is employed for both algorithms.

