# OpenReview forum: "Variance Reduced Distributed Non-Convex Optimization Using Matrix Stepsizes"
_ICLR.cc/2025/Conference — Submitted to ICLR 2025_

### Official Review · Reviewer_3M4m · 2024-10-21

**Soundness:** 3
**Presentation:** 3
**Contribution:** 2
**Rating:** 5
**Confidence:** 3

**Summary:**

the paper considers matrix step size for federated learning, combined with some variance reduction methods to achieve exact convergence

**Strengths:**

variance reduced method together with matrix step size shows convergence

**Weaknesses:**

only considered nonconvex case

**Questions:**

1. What is the challenge in analysis of combining variance reduction with matrix step size?
2. How does the rate compare with existing literature?
3. Why one should take the trouble of using matrix step size?

---

> ### Author Response · Authors · 2024-11-17
> **Response to Reviewer 3M4m**
>
> > We thank the reviewer for taking time to review our paper.
>
> ---
> #### Weakness: "only considered nonconvex case".
>
> > In this paper, we focused on the general case where $f$ may be non-convex; however, this does not imply that $f$ must necessarily be non-convex. Our rationale for this approach is to establish a general result that remains valid for convex functions as well. The primary motivation behind this work is to address the limitations of det-CGD, which converges only to a neighborhood of the solution.
>
> ---
> #### Question $1$: "What is the challenge in analysis of combining variance reduction with matrix step size?"
>
> > The primary challenge lies in providing an optimal theoretical analysis. Making matrix step sizes work effectively with variance reduction in non-convex settings presents numerous difficulties, each requiring careful consideration. To date, all existing results on variance reduction are confined to scalar step sizes. To achieve a meaningful convergence guarantee, it is necessary to introduce the matrix Lipschitz condition, examine its relationship with smoothness, and assess its practical feasibility. In particular, the proper handling of various matrix norms to derive an optimal bound presents a significant challenge. For det-MARINA and det-DASHA, we also demonstrate that their complexity compares favorably to their scalar counterparts and to det-CGD, even without employing optimal step sizes. Significantly, our work represents the first rigorous effort to establish the effectiveness of variance reduction techniques with matrix step sizes.
>
> ---
> #### Question $2$: "How does the rate compare with existing literature?"
>
> > As discussed in Section 6, the det-MARINA and det-DASHA algorithms outperform their scalar counterparts, MARINA and DASHA, in terms of both iteration complexity and communication complexity from a theoretical perspective. Numerical evidence further supports the effectiveness of our methods. In comparison to det-CGD, which is limited to convergence within a neighborhood, both det-MARINA and det-DASHA achieve superior performance in reaching a specified accuracy level. These conclusions are rigorously supported by the proofs provided in Appendices D and E, with additional numerical evidence presented in Appendix I.
>
> ---
> #### Question $3$: "Why one should take the trouble of using matrix step size?"
>
> > The motivation for employing matrix step-sized methods lies in achieving faster convergence and reducing communication overhead, albeit at the cost of some additional computation. This rationale is analogous to the reasoning behind using second-order methods, such as Newton's method, instead of relying solely on first-order methods. Matrix step-sized methods can be viewed as an intermediate approach between first-order and second-order methods. While they require extra computational effort, they offer faster convergence rates, ultimately leading to improved complexities.

---

### Official Review · Reviewer_iSCf · 2024-10-27

**Soundness:** 3
**Presentation:** 2
**Contribution:** 2
**Rating:** 5
**Confidence:** 3

**Summary:**

The paper proposes matrix-stepsize versions, namely det-MARINA and det-DASHA, of distributed variance-reduction algorithms MARINA and DASHA. The main idea was that the scalar stepsize can be replaced by a matrix stepsize similar to second-order (Newton) methods. The authors derive convergence guarantees to stationarity in the sense that the gradient squared norm vanishes with rate $O(1/k)$ where $k$ is the algorithms' iterate. Numerical experiments show some improvement of the proposed methods compared to MARINA and DASHA.

**Strengths:**

The paper is generally well-written with sufficient background. The motivation is also clear and natural: matrix stepsize is better than scalar stepsize. Convergence analysis is provided.

**Weaknesses:**

Since the proposed methods are matrix-stepsize adaptations of MARINA and DASHA, we can expect them to retain the general properties observed in the original MARINA and DASHA schemes. To my understanding, both MARINA and DASHA also converge to stationarity and, unlike det-CGD, do not exhibit a restrictive convergence neighborhood. Thus, these new algorithms would naturally inherit this characteristic, which makes this advantage less novel in my view. From a theoretical standpoint, it would be helpful to directly compare Theorems 1 and 2 with the convergence properties of MARINA and DASHA (see specific questions below). Additionally, the Lipschitz matrix seems impractical in most cases, whereas the Lipschitz constant is typically more accessible; therefore, including the Lipschitz matrix may not be essential. While the experiments do show improvements, the gains appear relatively modest.

**Questions:**

(1) Regarding the optimal matrix stepsize in the optimization problem in line 304, the authors minimize the RHS of (5) with respect to D that aims to makes the bound as small as possible. However, the LHS of (5) also contains D. How could you tell that minimizing the RHS yields the optimal improvement?

(2) When comparing det-MARINA and and MARINA, in Corollary 3 for example, they say this iteration complexity of det-MARINA is better than MARINA. Please elaborate this claim more. For example, their gradient norm are not the same, how can we compare them?

---

> ### Author Response · Authors · 2024-11-17
> **Response to Reviewer iSCf**
>
> > We thank the reviewer for taking time to review our paper.
>
> ---
> #### Weakness $1$: "... Additionally, ... While the experiments do show improvements, the gains appear relatively modest."
>
> > We agree with the reviewer that the matrix Lipschitz assumption is indeed less practical than its scalar counterpart. We do not claim that this assumption is universally practical across all settings. However, it is worth noting that the assumption encompasses the scalar Lipschitz condition as a special case when ${\bf L} = L\cdot {\bf I}_d$ (this is to say that whenever scalar Lipschitz condition holds, matrix Lipschitz also holds in this special case). The key point is that, even under the scalar Lipschitz condition, utilizing matrix step sizes still leads to faster convergence. Our intention in presenting the matrix Lipschitz condition is similar to the motivation behind the matrix smoothness assumption introduced in [1] : to illustrate the extent of improvement that can be achieved under such conditions.
>
> > [1] Smoothness matrices beat smoothness constants: better communication compression techniques for distributed optimization. M. Safaryan, F. Hanzely and P. Richtarik.
>
> ---
> #### Question $1$: "However, the LHS of (5) also contains D. How could you tell that minimizing the RHS yields the optimal improvement?"
>
> > Good question. Yes, the left-hand side of (5) indeed involves $\bf{D}$ as the weight matrix of the matrix norm. However, the weight matrix is given by ${\bf{D}}/\det({\bf{D}})^{1/d}$ which has determinant $1$, which has a determinant of 1. In this sense, it is analogous to the standard Euclidean norm, which is a matrix norm with the identity matrix as its weight, also having a determinant of 1. This can be interpreted geometrically as the vector being contained within an ellipsoid of the same volume as a corresponding ball. Numerical evidence from the det-CGD paper further suggests that the matrix norm in this context is comparable to the standard Euclidean norm. Based on this observation, we temporarily set aside the left-hand side and focus solely on optimizing the right-hand side.
>
> ---
> #### Question $2$: "... Please elaborate this claim more. For example, their gradient norm are not the same, how can we compare them?"
>
> > Actually, this question relates to the first question you raised. While the gradient norm is not identical in the two cases, as previously noted, the matrix norm of  ${\bf{D}}/\det({\bf{D}})^{1/d}$ and the standard Euclidean norm are similar. For a given accuracy level $\varepsilon^2$, the distinction lies in the geometric interpretation: the matrix norm corresponds to an ellipsoid with the same volume as the ball represented by the standard Euclidean norm. The comparison here is grounded in the similarity between the two norms. While we agree that, through appropriate transformations of the matrix norm, direct comparisons can be achieved, such comparisons often depend on the specific choice of $\bf{D}$, which adds further complexity to the results.

---

> > ### Comment · Reviewer_iSCf · 2024-11-22
> >
> > Thank you for taking the time to address my comments. While I agree with the argument on the comparability of the two norms, other weaknesses remain. I believe that the paper has certain merits, but I am not able to recommend an acceptance because it feels like the idea is a bit less novel and technical (I did not mean the convergence analysis would be easy) and the empirical experiment does not show a clear advantage of the proposed matrix stepsize -- the method that already adds one layer of complexity into the classical scalar stepsize.
> >
> > Future suggestion: maybe to demonstrate the advantage of the matrix stepsize in practice, one can set up the problem where the Hessian of the objective function is very skewed. In such a case, the scalar stepsize would be very sensitive and need a very small stepsize to run, while your method can scale differently in different directions.

---

> > > ### Author Response · Authors · 2024-11-22
> > > **Response to Reviewer iSCf**
> > >
> > > We sincerely thank the reviewer for the valuable suggestions and the time dedicated to reviewing our paper.

---

### Official Review · Reviewer_H5fA · 2024-11-01

**Soundness:** 3
**Presentation:** 2
**Contribution:** 2
**Rating:** 3
**Confidence:** 3

**Summary:**

The work "Variance reduced distributed nonconvex optimization using matrix stepsizes" proposes decentralized variance reduced schemes with matrix step-sizes. The incorporation of variance reductions allows for convergence to arbitrary precision as opposed to the original work Li et. al (2024b). The authors provide theoretical guarantees for their algorithms and demonstrate their performance empirically.

**Strengths:**

1) The proposed schemes address an issue that makes the distributed version of det-CGD less desirable. This is achieved via the incorporation of a variance reduction mechanisms that aids in controlling the variance introduced by the compression.
2) The work is well written and the contribution is clear.

**Weaknesses:**

1) The novelty of this work is limited, as it seems combine a known algorithm (detCGD) with a given problem with two known solutions to the problem (proposed in MARINA and DASHA). It is unclear to this reviewer what is the main challenge when combining these. I suggest the authors highlight which technical challenges need to be addressed to incorporate variance reduction into detCGD, and why these significantly differ from the case in which the step-size is a scalar.

**Questions:**

See weaknesses.

---

> ### Author Response · Authors · 2024-11-17
> **Response to Reviewer H5fA**
>
> > We thank the reviewer for taking time to review our paper.
>
> ---
> #### Weakness: "The novelty of this work is limited, as it seems combine a known algorithm (detCGD) with a given problem with two known solutions to the problem (proposed in MARINA and DASHA). It is unclear to this reviewer what is the main challenge when combining these. I suggest the authors highlight which technical challenges need to be addressed to incorporate variance reduction into detCGD, and why these significantly differ from the case in which the step-size is a scalar."
>
> > Thank you for your feedback. Effectively implementing matrix step sizes with variance reduction in non-convex settings involves many challenges, each demanding attention. To date, all established results on variance reduction are restricted to scalar step sizes, and addressing the associated technicalities requires significant effort before reaching definitive conclusions.
>
> > The primary challenge lies in developing an optimal theoretical analysis for the matrix-based method. Achieving a meaningful convergence guarantee necessitates the introduction of the matrix Lipschitz condition, an exploration of its relationship with smoothness, and a thorough assessment of its practical applicability. A particularly intricate aspect is the proper handling of various matrix norms to derive an optimal bound. While there are multiple ways to achieve a similar, albeit sub-optimal, convergence guarantee for both det-DASHA and det-MARINA, the specific choice of matrix norm significantly influences the outcome, despite the general guidelines provided by their scalar counterparts.
>
> > For det-MARINA and det-DASHA, we demonstrate that their complexity is competitive with their scalar counterparts and with det-CGD, even without employing optimal step sizes. Reaching these conclusions required significant effort, as comparisons across different norms are inherently more challenging to address. Notably, our work represents the first rigorous attempt to establish the effectiveness of variance reduction techniques within the framework of matrix step sizes.

---

### Official Review · Reviewer_gJqi · 2024-11-05

**Soundness:** 3
**Presentation:** 3
**Contribution:** 2
**Rating:** 5
**Confidence:** 3

**Summary:**

This paper provided two novel federated learning optimization methods based on variance-reduced matrix stepsizes. The theoretical analysis shows the communication complexity of proposed methods are better than existing methods, which is also validated by numerical experiments.

**Strengths:**

The idea of using matrix stepsize and variance-reduction to improve the communication complexity in federated learning is interesting. The thermotical guarantees have been established.

**Weaknesses:**

Please see the part of questions.

**Questions:**

I have several questions about Section 4.2.

a) Line 310 says we can relax the problem to certain linear subspace and achieve Corollary 1. Is there any theoretical result to bound the approximation error of such relax?

b) In the expression of equation (6), we require achieving matrices ${\bf W}^{1/2}$, ${\bf L}^{-1}$, and several eigenvalues. Is the computation of these quantity expensive in practice?

c) Line 394 says we can estimate ${\bf L}_i$ by gradient method (Wang et al., 2022). I have not detailed checked the paper of Want et al. (2022). Can you summarize their main results and ideas (e.g., the complexity and the algorithm design)?

I have also some questions about the comparisons with related work.

d) I think more detailed introduction for MARINA and DASHA is necessary, which is useful to help the readers understand why the authors focus on the comparisons with these two methods.

e) The iteration with matrix stepsize looks more expensive than classical first-order methods. I think the comparison with classical first-order methods (e.g., distributed GD, SGD, SARAH...) should be involved, including the discussion on the computational cost.

---

> ### Author Response · Authors · 2024-11-17
> **Response to Reviewer gJqi**
>
> > We thank the reviewer for taking time to review our paper.
>
> ---
> #### Question (a):
>
> > Yes, in fact, due to the high complexity involved in determining the optimal matrix step size that satisfies the required conditions, we evaluate our algorithms, det-MARINA and det-DASHA, using suboptimal matrix step sizes obtained through this alternative (relaxing to linear subspace) approach. In Section 6, as well as Appendices D and E, we provide rigorous proofs demonstrating that these suboptimal matrix step sizes achieve better complexity than their scalar counterparts, provided that the matrix $\bf W$ is selected appropriately.
>
> ---
> #### Question (b):
>
> > Good question. In practice, the computation is often not overly expensive. This is because, rather than requiring individual eigenvalues of the matrices, we primarily need to compute the following: (i): $\lambda_{\max}({\bf W^{1/2}LW^{1/2}})$ and (ii): $\lambda_{\max}(\mathbb{E}[{\bf SWS}]-{\bf W})$. Choices of $\bf W$  that yield satisfactory performance often include performance are ofthen $L^{-1}$, $diag({\bf L})^{-1}$ and other options related to $\bf L$. For nearly all these cases, the computation of (i) is greatly simplified. For instance, in the two examples mentioned, this value reduces to $1$. For (ii), the eigenvalue computation depends on the specific sketch used. For example, is we are using rand-$1$ sketch, then we result in $\lambda_{\max}(d\cdot diag(W)-{\bf W})$, which is also straightforward to evaluate.
>
> > Moreover, these computations occur on the server side only once before the training start, where computational resources are typically assumed to be sufficient. Thus, the overall computational burden is manageable in most practical scenarios.
>
> ---
> #### Question (c):
>
> > The results of Wang et al. (2022) extend the findings of [1]. Under the matrix smoothness assumption, they generalize the linear compressors used in [1] to arbitrary unbiased compressors in the smooth convex setting, achieving significant communication savings. Their main algorithm generalizes the distributed compressed gradient descent algorithm by incorporating a specially designed smoothness-aware unbiased compressor. When combined with variance reduction techniques in the convex setting, their proposed algorithm achieves state-of-the-art communication complexity for convex problems.
>
> > Although our work primarily focuses on the non-convex setting, both approaches require estimating the smoothness matrix $\bf L$. This shared requirement is why we referenced their paper as a relevant and valuable contribution.
>
> > [1] Smoothness matrices beat smoothness constants: better communication compression techniques for distributed optimization. M. Safaryan, F. Hanzely and P. Richtarik.
>
> ---
> #### Question (d):
>
> > Thank you for your suggestion. We have expanded the discussion of MARINA and DASHA in Appendix A.2, where we also cover variance reduction methods such as L-SVRG, DIANA, and STORM. DASHA and MARINA are designed to address convergence to a neighborhood in non-convex settings. To further enhance the discussion on variance reduction, we have added the following to the related work section:
>
> > DASHA and MARINA are variance reduction methods specifically designed to eliminate the issue of convergence to a neighborhood in non-convex settings. In this paper, we provide extensive comparisons with these two algorithms, as our proposed det-DASHA and det-MARINA serve as their counterparts in the matrix setting. For readers seeking additional details on distributed CGD in either the convex or non-convex case, as well as the associated variance reduction techniques, we refer them to Appendix A.2.
>
> ---
> #### Question (e):
>
> > Thank you for the suggestion. We will incorporate the following discussion into the latest version of the paper:
>
> > Using a matrix step size introduces some additional computational overhead. However, in det-MARINA, det-DASHA, and det-CGD, the matrix-vector multiplications occur on the server side. Although matrix multiplications involving sketches take place on the client side, these sketches act as compressors and are typically very sparse, making the computational cost negligible. Notably, this computation also occurs in the standard CGD algorithm (which is also SGD but with certain sparse gradient estimators).
>
> > In this paper, we assume sufficient computational resources on the server, focusing on reducing communication overhead during training. Both det-DASHA and det-MARINA extend det-CGD, a generalization of standard CGD, with SGD recoverable as a special case when identity matrices are used as sketches. The main computational difference lies in matrix-vector multiplication on the server, which is not a bottleneck under our assumption. DASHA and MARINA are variance-reduced methods achieving optimal complexities for non-convex functions, while algorithms like DIANA and SARAH provide similar benefits for convex settings.

---

### Author Response · Authors · 2024-11-17
**Response to all reviewers**

We sincerely thank all the reviewers for taking the time to review our paper.

The reviewers raised several concerns, for which we have prepared a comprehensive, case-by-case response to each reviewer. We hope this detailed clarification addresses their questions and provides additional insight into our work.

---

### Meta-Review · Area_Chair_T5rh · 2024-12-18

**Metareview:**

This paper extends the det-CGD algorithm, which uses matrix stepsizes for non-convex optimization, by introducing two variance-reduced versions: det-MARINA and det-DASHA. These methods improve iteration and communication complexities over the existing methods, with both theoretical and empirical validation of their superior performance.

While the paper's theoretical approach is interesting, the main criticism raised by the reviewers was the lack of novelty in the proposed method. Specifically, the following weaknesses were highlighted:

1. One reviewer noted that the novelty of this work is limited, as it combines a known algorithm (det-CGD) with a given problem and two existing solutions (MARINA and DASHA). The reviewer found it unclear what specific challenges were addressed in combining these components.
2. Another reviewer expressed a similar concern, stating that the results of this paper are unsurprising when compared to MARINA and DASHA.

**Additional Comments On Reviewer Discussion:**

Four reviews were collected for this paper, but one was dropped by the AC due to being short and uninformative. The remaining reviewers unanimously recommended rejection, and the AC concurs with their concerns, supporting a rejection. However, the AC strongly encourages resubmission, recognizing the paper's potential importance and interest while noting the need for substantial revisions.

---

### Decision · Program_Chairs · 2025-01-22

Reject